# Closing the Computational-Statistical Gap in Best Arm Identification for Combinatorial Semi-bandits

**Ruo-Chun Tzeng**
EECS
KTH, Stockholm, Sweden
rctzeng@kth.se

**Po-An Wang**
EECS
KTH, Stockholm, Sweden
wang9@kth.se

**Alexandre Proutiere**
EECS and Digital Futures
KTH, Stockholm, Sweden
alepro@kth.se

**Chi-Jen Lu**
Institute of Information Science
Academia Sinica, Taiwan
cjlu@iis.sinica.edu.tw

## Abstract

We study the best arm identification problem in combinatorial semi-bandits in the fixed confidence setting. We present Perturbed Frank-Wolfe Sampling (`P-FWS`), an algorithm that (i) runs in polynomial time, (ii) achieves the instance-specific minimal sample complexity in the high confidence regime, and (iii) enjoys polynomial sample complexity guarantees in the moderate confidence regime. To the best of our knowledge, even for the vanilla bandit problems, no algorithm was able to achieve (ii) and (iii) simultaneously. With `P-FWS`, we close the computational-statistical gap in best arm identification in combinatorial semi-bandits. The design of `P-FWS` starts from the optimization problem that defines the information-theoretical and instance-specific sample complexity lower bound. `P-FWS` solves this problem in an online manner using, in each round, a single iteration of the Frank-Wolfe algorithm. Structural properties of the problem are leveraged to make the `P-FWS` successive updates computationally efficient. In turn, `P-FWS` only relies on a simple linear maximization oracle.

## 1 Introduction

An efficient method to design statistically optimal algorithms solving active learning tasks (e.g., regret minimization or pure exploration in bandits and reinforcement learning) consists in the following two-step procedure. The first step amounts to deriving, through change-of-measure arguments, tight information-theoretical fundamental limits satisfied by a wide class of learning algorithms. These limits are often expressed as the solution of an optimization problem, referred in this paper to as the *lower-bound problem*. Interestingly, this solution specifies the instance-specific optimal exploration process: it characterizes the limiting behavior of the adaptive sampling rule any statistically optimal algorithm should implement. In the second step, the learning algorithm is designed so that its exploration process approaches the solution of the lower-bound problem. This design yields statistically optimal algorithms, but typically requires to repeatedly solve the lower-bound problem. This method has worked remarkably well for simple learning tasks such as regret minimization or best-arm identification with fixed confidence in classical stochastic bandits [Lai87, GC11, GK16], but also in bandits whose arm-to-average reward function satisfies simple structural properties (e.g., Lipschitz, unimodal) [MCP14, WTP21].

The method also provides a natural way of studying the computational-statistical gap [KLLM22] for active learning tasks. Indeed, if solving the lower-bound problem in polynomial time is possible, one

37th Conference on Neural Information Processing Systems (NeurIPS 2023).

may hope to devise learning algorithms that are both statistically optimal and computationally efficient. As of now, however, the computational complexity of the lower-bound problem remains largely unexplored, except for simple learning tasks. For example, in the case of best policy identification in tabular Markov Decision Processes, the lower-bound problem is non-convex [AMP21] and its complexity and approximability are unclear.

In this paper, we leverage the aforementioned two-step procedure to assess the computational-statistical gap for the best arm identification in combinatorial semi-bandits in the fixed confidence setting. We establish that, essentially, this gap does not exist (a result that was conjectured in [JMKK21]). Specifically, we present an algorithm that enjoys the following three properties: (i) it runs in polynomial time, (ii) its sample complexity matches the fundamental limits asymptotically in the high confidence regime, and (iii) its sample complexity is at most polynomial in the moderate confidence regime. Next, after formally introducing combinatorial semi-bandits, we describe our contributions and techniques in detail.

**Best arm identification in combinatorial semi-bandits.** In combinatorial semi-bandits [CBL12, CTMSP$^+$15], the learner sequentially selects an action from a combinatorial set $\mathcal{X} \subset \{0,1\}^K$. When in round $t$, the action $\boldsymbol{x}(t) = (x_1(t), \ldots, x_K(t)) \in \mathcal{X}$ is chosen, the environment samples a $K$-dimensional vector $\boldsymbol{y}(t)$ whose distribution is assumed to be Gaussian $\mathcal{N}(\boldsymbol{\mu}, \mathbf{I})$. The learner then receives the detailed reward vector $\boldsymbol{x}(t) \odot \boldsymbol{y}(t)$ where $\odot$ denotes the element-wise product (in other words, the learner observes the individual reward $y_k(t)$ of the arm $k$ if and only if this arm is selected in round $t$, i.e., $x_k(t) = 1$). The parameter $\boldsymbol{\mu}$ characterizing the average rewards of the various arms is initially unknown. The goal of a learner is to identify the best action $\boldsymbol{i}^\star(\boldsymbol{\mu}) = \mathrm{argmax}_{\boldsymbol{x} \in \mathcal{X}} \langle \boldsymbol{x}, \boldsymbol{\mu} \rangle$ with a given level of confidence $1 - \delta$, for some $\delta > 0$ while minimizing the expected number of rounds needed. We assume that the best action is unique and denote by $\Lambda = \{\boldsymbol{\mu} \in \mathbb{R}^K : |\boldsymbol{i}^\star(\boldsymbol{\mu})| = 1\}$ the set of parameters satisfying this assumption. The learner strategy is defined by three components: (i) a sampling rule dictating the sequence of the selected actions; (ii) a stopping time $\tau$ defining the last round where the learner interacts with the environment; (iii) a decision rule specifying the action $\hat{\boldsymbol{i}} \in \mathcal{X}$ believed to be optimal based on the data gathered until $\tau$.

**The sample complexity lower-bound problem.** Consider the set of $\delta$-PAC algorithms such that for any $\boldsymbol{\mu} \in \Lambda$, the best action is identified correctly with probability at least $1 - \delta$. We wish to find a $\delta$-PAC algorithm with minimal expected sample complexity $\mathbb{E}_{\boldsymbol{\mu}}[\tau]$. To this aim, using classical change-of-measure arguments [GK16], we may derive a lower bound of the expected sample complexity satisfied by any $\delta$-PAC algorithm. This lower bound is given by[1] $\mathbb{E}_{\boldsymbol{\mu}}[\tau] \geq T^\star(\boldsymbol{\mu})\mathsf{kl}(\delta, 1 - \delta)$. The characteristic time $T^\star(\boldsymbol{\mu})$ is defined as the value of the following problem

$$T^\star(\boldsymbol{\mu})^{-1} = \sup_{\boldsymbol{\omega} \in \Sigma} \inf_{\boldsymbol{\lambda} \in \mathsf{Alt}(\boldsymbol{\mu})} \left\langle \boldsymbol{\omega}, \frac{(\boldsymbol{\mu} - \boldsymbol{\lambda})^2}{2} \right\rangle, \tag{1}$$

where[2] $\Sigma = \{\sum_{\boldsymbol{x} \in \mathcal{X}} w_{\boldsymbol{x}} \boldsymbol{x} : \boldsymbol{w} \in \Sigma_{|\mathcal{X}|}\}$, $\mathsf{kl}(a, b)$ is the KL-divergence between two Bernoulli distributions with respective means $a$ and $b$, and $\mathsf{Alt}(\boldsymbol{\mu}) = \{\boldsymbol{\lambda} \in \Lambda : \boldsymbol{i}^\star(\boldsymbol{\lambda}) \neq \boldsymbol{i}^\star(\boldsymbol{\mu})\}$ is the set of *confusing* parameters. As it turns out (see Lemma 1), $T^\star(\boldsymbol{\mu})$ is at most quadratic in $K$, and hence the sample complexity lower bound is polynomial. (1) is a concave program over $\Sigma$ [WTP21], and a point $\boldsymbol{\omega}^\star$ in its solution set corresponds to an optimal allocation of action draws: an algorithm sampling actions according to $\boldsymbol{\omega}^\star$ and equipped with an appropriate stopping rule would yield a sample complexity matching the lower bound. In this paper, we provide computationally efficient algorithms to solve (1) and show how these can be used to devise a $\delta$-PAC best action identification algorithm with minimal sample complexity and running in polynomial time. We only assume that we have access to a computationally efficient Oracle, referred to as the LM (Linear Maximization) Oracle, identifying the best action should $\boldsymbol{\mu}$ be known (but for any possible $\boldsymbol{\mu}$). This assumption, made in all previous work on combinatorial semi-bandits (see e.g. [JMKK21, PBVP20]), is crucial as indeed, if there is no computationally efficient algorithm solving the offline problem $\mathrm{argmax}_{\boldsymbol{x} \in \mathcal{X}} \langle \boldsymbol{x}, \boldsymbol{\mu} \rangle$ with known $\boldsymbol{\mu}$, there is no hope to solve its online version with unknown $\boldsymbol{\mu}$ in a computationally efficient manner. The assumption holds for a large array of combinatorial sets of actions [S$^+$03], including $m$-sets, matchings, (source–destination)-paths, spanning trees, matroids (refer to [CCG21b] for a thorough discussion).

---

[1] We present proof in Appendix K for completeness – see also [JMKK21].

[2] $\Sigma_N$ denotes the $(N - 1)$ dimensional simplex.

**The Most-Confusing-Parameter (**MCP**) algorithm.** The difficulty of solving (1) lies in the inner optimization problem, i.e., in evaluating the objective function:

$$F_{\boldsymbol{\mu}}(\boldsymbol{\omega}) = \inf_{\boldsymbol{\lambda} \in \mathsf{Alt}(\boldsymbol{\mu})} \left\langle \boldsymbol{\omega}, \frac{(\boldsymbol{\mu} - \boldsymbol{\lambda})^2}{2} \right\rangle = \min_{\boldsymbol{x} \neq \boldsymbol{i}^{\star}(\boldsymbol{\mu})} f_{\boldsymbol{x}}(\boldsymbol{\omega}, \boldsymbol{\mu}) \tag{2}$$

where $f_{\boldsymbol{x}}(\boldsymbol{\omega}, \boldsymbol{\mu}) = \inf_{\boldsymbol{\lambda} \in \mathcal{C}_{\boldsymbol{x}}} \langle \boldsymbol{\omega}, \frac{(\boldsymbol{\mu} - \boldsymbol{\lambda})^2}{2} \rangle$ and $\mathcal{C}_{\boldsymbol{x}} = \{ \boldsymbol{\lambda} \in \mathbb{R}^K : \langle \boldsymbol{\lambda}, \boldsymbol{i}^{\star}(\boldsymbol{\mu}) - \boldsymbol{x} \rangle < 0 \}$. Evaluating $F_{\boldsymbol{\mu}}(\boldsymbol{\omega})$ is required to solve (1), but also in the design of an efficient stopping rule. Our first contribution is MCP (Most-Confusing-Parameter), a polynomial time algorithm able to approximate $F_{\boldsymbol{\mu}}(\boldsymbol{\omega})$ for any given $\boldsymbol{\mu}$ and $\boldsymbol{\omega}$. The algorithm's name refers to the fact that by computing $F_{\boldsymbol{\mu}}(\boldsymbol{\omega})$, we implicitly identify the *most confusing parameter* $\boldsymbol{\lambda}^{\star} \in \arg \inf_{\boldsymbol{\lambda} \in \mathsf{Alt}(\boldsymbol{\mu})} \langle \boldsymbol{\omega}, \frac{(\boldsymbol{\mu} - \boldsymbol{\lambda})^2}{2} \rangle$. The design of MCP leverages a Lagrangian relaxation of the optimization problem defining $f_{\boldsymbol{x}}(\boldsymbol{\omega}, \boldsymbol{\mu})$ and exploits the fact that the Lagrange dual function linearly depends on $\boldsymbol{x}$. In turn, this linearity allows us to make use of the LM Oracle. From these observations, we show that computing $F_{\boldsymbol{\mu}}(\boldsymbol{\omega})$ boils down to solving a two-player game, for which one of the players can simply update her strategy using the LM Oracle. We prove the following informally stated theorem quantifying the performance of the MCP algorithm (see Theorem 3 for a more precise statement).

**Theorem 1.** *For any* $(\boldsymbol{\omega}, \boldsymbol{\mu})$*, the* MCP *algorithm with precision* $\epsilon$ *and certainty parameter* $\theta$ *returns* $\hat{F}$ *and* $\hat{\boldsymbol{x}}$ *satisfying* $\mathbb{P}_{\boldsymbol{\mu}}[F_{\boldsymbol{\mu}}(\boldsymbol{\omega}) \leq \hat{F} \leq (1 + \epsilon)F_{\boldsymbol{\mu}}(\boldsymbol{\omega})] \geq 1 - \theta$ *and* $\hat{F} = f_{\hat{\boldsymbol{x}}}(\boldsymbol{\omega}, \boldsymbol{\mu})$*. The number of calls to the* LM *Oracle is, almost surely, at most polynomial in* $K$*,* $\epsilon^{-1}$*, and* $\ln \theta^{-1}$*.*

**The Perturbed Frank-Wolfe Sampling (**P-FWS**) algorithm.** The MCP algorithm allows us to solve the lower-bound problem (1) for any given $\boldsymbol{\mu}$. The latter is initially unknown, but could be estimated. A Track-and-Stop algorithm [GK16] solving (1) with this plug-in estimator in each round would yield asymptotically minimal sample complexity. It would however be computationally expensive. To circumvent this difficulty, as in [WTP21], our algorithm, P-FWS, performs a single iteration of the Frank-Wolfe algorithm for the program (1) instantiated with an estimator of $\boldsymbol{\mu}$. To apply the Frank-Wolfe algorithm, P-FWS uses stochastic smoothing techniques to approximate the non-differentiable objective function $F_{\boldsymbol{\mu}}$ by a smooth function. To estimate the gradient of the latter, P-FWS leverages both the LM Oracle and the MCP algorithm (more specifically its second output $\hat{\boldsymbol{x}}$). Finally, P-FWS stopping rule takes the form of a classical Generalized Likelihood Ratio Test (GLRT) where the estimated objective function is compared to a time-dependent threshold. Hence the stopping rule also requires the MCP algorithm. We analyze the sample and computational complexities of P-FWS. Our main results are summarized in the following theorem (refer to Theorem 4 for details).

**Theorem 2.** *For any* $\delta \in (0, 1)$*,* P-FWS *is* $\delta$*-PAC, and for any* $(\epsilon, \tilde{\epsilon}) \in (0, 1)$ *small enough, its sample complexity satisfies:*

$$\mathbb{E}_{\boldsymbol{\mu}}[\tau] \leq \frac{(1 + \tilde{\epsilon})^2}{T^{\star}(\boldsymbol{\mu})^{-1} - \epsilon} \times H\left(\frac{1}{\delta} \cdot \frac{c(1 + \tilde{\epsilon})^2}{T^{\star}(\boldsymbol{\mu})^{-1} - \epsilon}\right) + \Psi(\epsilon, \tilde{\epsilon}),$$

*where* $H(x) = \ln(x) + \ln\ln(x)$*,* $c > 0$ *is a universal constant, and* $\Psi(\epsilon, \tilde{\epsilon})$ *is polynomial in* $\epsilon^{-1}$*,* $\tilde{\epsilon}^{-1}$*,* $K$*,* $\|\boldsymbol{\mu}\|_{\infty}$*, and* $\triangle_{\min}^{-1}$*, where* $\triangle_{\min} = \min_{\boldsymbol{x} \neq \boldsymbol{i}^{\star}(\boldsymbol{\mu})} \langle \boldsymbol{i}^{\star}(\boldsymbol{\mu}) - \boldsymbol{x}, \boldsymbol{\mu} \rangle$*. Under* P-FWS*, the number of* LM *Oracle calls per round is at most polynomial in* $\ln \delta^{-1}$ *and* $K$*. The total expected number of these calls is also polynomial.*

To the best of our knowledge, P-FWS is the first polynomial time best action identification algorithm with minimal sample complexity in the high confidence regime (when $\delta$ tends to 0). Its sample complexity is also polynomial in $K$ in the moderate confidence regime.

## 2 Preliminaries

We start by introducing some notation. We use bold lowercase letters (e.g., $\boldsymbol{x}$) for vectors, and bold uppercase letter (e.g., $\mathbf{A}$) for matrices. $\odot$ (resp. $\oplus$) denotes the element-wise product (resp. sum over $\mathbb{Z}_2$). For $\boldsymbol{x} \in \mathbb{R}^K$, $i \in \mathbb{N}$, $\boldsymbol{x}^i = (x_k^i)_{k \in [K]}$ is the $i$-th element-wise power of $\boldsymbol{x}$. $D = \max_{\boldsymbol{x} \in \mathcal{X}} \|\boldsymbol{x}\|_1$ denotes the maximum number of arms part of an action. For any $\boldsymbol{\mu} \in \Lambda$, we define the sub-optimality gap of $\boldsymbol{x} \in \mathcal{X}$ as $\triangle_{\boldsymbol{x}}(\boldsymbol{\mu}) = \langle \boldsymbol{i}^{\star}(\boldsymbol{\mu}) - \boldsymbol{x}, \boldsymbol{\mu} \rangle$, and the minimal gap as $\triangle_{\min}(\boldsymbol{\mu}) = \min_{\boldsymbol{x} \neq \boldsymbol{i}^{\star}(\boldsymbol{\mu})} \triangle_{\boldsymbol{x}}(\boldsymbol{\mu})$. $\mathbb{P}_{\boldsymbol{\mu}}$ (resp. $\mathbb{E}_{\boldsymbol{\mu}}$) denotes the probability measure (resp. expectation) when the arm rewards are parametrized by $\boldsymbol{\mu}$. Whenever it is clear from the context, we will drop $\boldsymbol{\mu}$ for simplicity, e.g. $\boldsymbol{i}^{\star} = \boldsymbol{i}^{\star}(\boldsymbol{\mu})$, $\triangle_{\boldsymbol{x}} = \triangle_{\boldsymbol{x}}(\boldsymbol{\mu})$, and $\triangle_{\min} = \min_{\boldsymbol{x} \neq \boldsymbol{i}^{\star}} \triangle_{\boldsymbol{x}}$. Refer to Appendix A for an exhaustive table of notation.

## 2.1 The lower-bound problem

Classical change-of-measure arguments lead to the asymptotic sample complexity lower bound $\mathbb{E}_{\boldsymbol{\mu}}[\tau] \geq T^{\star}(\boldsymbol{\mu})\mathsf{kl}(\delta, 1 - \delta)$ where the characteristic time is defined in (1). To have a chance to develop a computationally efficient best action identification algorithm, we need that the sample complexity lower bound grows at most polynomially in $K$. This is indeed the case as stated in the following lemma, whose proof is provided in Appendix K.

**Lemma 1.** *For any $\boldsymbol{\mu} \in \Lambda$, $T^{\star}(\boldsymbol{\mu}) \leq 4KD\triangle_{\min}(\boldsymbol{\mu})^{-2}$.*

We will use first-order methods to solve the lower-bound problem, and to this aim, we will need to evaluate the gradient w.r.t. $\boldsymbol{\omega}$ of $f_{\boldsymbol{x}}(\boldsymbol{\omega}, \boldsymbol{\mu})$. We can apply the envelop theorem [WTP21] to show that for $(\boldsymbol{\omega}, \boldsymbol{\mu}) \in \Sigma_+ \times \Lambda$,

$$\nabla_{\boldsymbol{\omega}} f_{\boldsymbol{x}}(\boldsymbol{\omega}, \boldsymbol{\mu}) = \frac{(\boldsymbol{\mu} - \boldsymbol{\lambda}_{\boldsymbol{\omega},\boldsymbol{\mu}}^{\star}(\boldsymbol{x}))^2}{2},$$

where $\Sigma_+ = \Sigma \cap \mathbb{R}_{>0}^K$, $\boldsymbol{\lambda}_{\boldsymbol{\omega},\boldsymbol{\mu}}^{\star}(\boldsymbol{x}) = \operatorname{argmin}_{\boldsymbol{\lambda} \in \mathsf{cl}(\mathcal{C}_{\boldsymbol{x}})} \langle \boldsymbol{\omega}, \frac{(\boldsymbol{\mu}-\boldsymbol{\lambda})^2}{2} \rangle$ and $\mathsf{cl}(\mathcal{C}_{\boldsymbol{x}})$ is the closure of $\mathcal{C}_{\boldsymbol{x}}$ (refer to Lemma 19 in Appendix G.2).

## 2.2 The Linear Maximization Oracle

As mentioned earlier, we assume that we have access to a computationally efficient Oracle, referred to as the LM (Linear Maximization) Oracle, identifying the best action if $\boldsymbol{\mu}$ is known. More precisely, as in existing works in combinatorial semi-bandits [KWA+14, PPV19, PBVP20], we make the following assumption.

**Assumption 1.** *(i) There exists a polynomial-time algorithm identifying $\boldsymbol{i}^{\star}(\boldsymbol{v})$ for any $\boldsymbol{v} \in \mathbb{R}^K$; (ii) $\mathcal{X}$ is inclusion-wise maximal, i.e., there is no $\boldsymbol{x}, \boldsymbol{x}' \in \mathcal{X}$ s.t. $\boldsymbol{x} < \boldsymbol{x}'$; (iii) for each $k \in [K]$, there exists $\boldsymbol{x} \in \mathcal{X}$ such that $x_k = 1$; (iv) $|\mathcal{X}| \geq 2$.*

Assumption 1 holds for combinatorial sets including $m$-sets, spanning forests, bipartite matching, $s$-$t$ paths. For completeness, we verify the assumption for these action sets in Appendix J. In the design of our MCP algorithm, we will actually need to solve for some $\boldsymbol{v} \in \mathbb{R}^K$ the linear maximization problem $\max\langle \boldsymbol{x}, \boldsymbol{v} \rangle$ over $\mathcal{X} \setminus \{\boldsymbol{i}^{\star}(\boldsymbol{\mu})\}$; in other words, we will probably need to identify the second best action. Fortunately, this can be done in a computationally efficient manner under Assumption 1. The following lemma formalizes this observation. Its proof, presented in Appendix J, is inspired by Lawler's $m$-best algorithm [Law72].

**Lemma 2.** *Let $\boldsymbol{v} \in \mathbb{R}^K$ and $\boldsymbol{x} \in \mathcal{X}$. Under Assumption 1, there exists an algorithm that solves $\max_{\boldsymbol{x}' \in \mathcal{X}: \boldsymbol{x}' \neq \boldsymbol{x}} \langle \boldsymbol{v}, \boldsymbol{x}' \rangle$ by only making at most $D$ queries to the LM Oracle.*

# 3 Solving the lower bound problem: the MCP algorithm

Solving the lower bound problem first requires to evaluate its objective function $F_{\boldsymbol{\mu}}(\boldsymbol{\omega})$. A naive approach, enumerating $f_{\boldsymbol{x}}(\boldsymbol{\omega}, \boldsymbol{\mu})$ for all $\boldsymbol{x} \in \mathcal{X} \setminus \{\boldsymbol{i}^{\star}\}$, would be computationally infeasible. In this section, we present and analyze MCP, an algorithm that approximates $F_{\boldsymbol{\mu}}(\boldsymbol{\omega})$ by calling the LM Oracle a number of times growing at most polynomially in $K$.

## 3.1 Lagrangian relaxation

The first step towards the design of MCP consists in considering the Lagrangian relaxation of the optimization problem defining $f_{\boldsymbol{x}}(\boldsymbol{\omega}, \boldsymbol{\mu}) = \inf_{\boldsymbol{\lambda} \in \mathcal{C}_{\boldsymbol{x}}} \langle \boldsymbol{\omega}, \frac{(\boldsymbol{\mu}-\boldsymbol{\lambda})^2}{2} \rangle$ (see e.g., [BV04, Vis21]). For any $(\boldsymbol{\omega}, \boldsymbol{\mu}) \in \Sigma_+ \times \Lambda$ and $\boldsymbol{x} \neq \boldsymbol{i}^{\star}$, the Lagrangian $\mathcal{L}_{\boldsymbol{\omega},\boldsymbol{\mu}}$ and Lagrange dual function $g_{\boldsymbol{\omega},\boldsymbol{\mu}}$ of this problem are defined as, $\forall \alpha \geq 0$,

$$\mathcal{L}_{\boldsymbol{\omega},\boldsymbol{\mu}}(\boldsymbol{\lambda}, \boldsymbol{x}, \alpha) = \left\langle \boldsymbol{\omega}, \frac{(\boldsymbol{\mu} - \boldsymbol{\lambda})^2}{2} \right\rangle + \alpha \langle \boldsymbol{i}^{\star} - \boldsymbol{x}, \boldsymbol{\lambda} \rangle \quad \text{and} \quad g_{\boldsymbol{\omega},\boldsymbol{\mu}}(\boldsymbol{x}, \alpha) = \inf_{\boldsymbol{\lambda} \in \mathbb{R}^K} \mathcal{L}_{\boldsymbol{\omega},\boldsymbol{\mu}}(\boldsymbol{\lambda}, \boldsymbol{x}, \alpha),$$

respectively. The following proposition, proved in Appendix C.1, provides useful properties of $g_{\boldsymbol{\omega},\boldsymbol{\mu}}$:

**Proposition 1.** *Let $(\boldsymbol{\omega}, \boldsymbol{\mu}) \in \Sigma_+ \times \Lambda$ and $\boldsymbol{x} \in \mathcal{X} \setminus \{\boldsymbol{i}^{\star}(\boldsymbol{\mu})\}$.*
*(a) The Lagrange dual function is linear in $\boldsymbol{x}$. More precisely, $g_{\boldsymbol{\omega},\boldsymbol{\mu}}(\boldsymbol{x}, \alpha) = c_{\boldsymbol{\omega},\boldsymbol{\mu}}(\alpha) + \langle \boldsymbol{\ell}_{\boldsymbol{\omega},\boldsymbol{\mu}}(\alpha), \boldsymbol{x} \rangle$*

where $c_{\boldsymbol{\omega},\boldsymbol{\mu}}(\alpha) = \alpha \left\langle \boldsymbol{\mu} - \frac{\alpha}{2}\boldsymbol{\omega}^{-1}, \boldsymbol{i}^\star(\boldsymbol{\mu}) \right\rangle$ and $\boldsymbol{\ell}_{\boldsymbol{\omega},\boldsymbol{\mu}}(\alpha) = -\alpha \left( \boldsymbol{\mu} + \frac{\alpha}{2}\boldsymbol{\omega}^{-1} \odot (\mathbf{1}_K - 2\boldsymbol{i}^\star(\boldsymbol{\mu})) \right)$.

*(b) $g_{\boldsymbol{\omega},\boldsymbol{\mu}}(\boldsymbol{x}, \cdot)$ is strictly concave (for any fixed $\boldsymbol{x}$).*

*(c) $f_{\boldsymbol{x}}(\boldsymbol{\omega}, \boldsymbol{\mu}) = \max_{\alpha \geq 0} g_{\boldsymbol{\omega},\boldsymbol{\mu}}(\boldsymbol{x}, \alpha)$ is attained by $\alpha_{\boldsymbol{x}}^\star = \frac{\triangle_{\boldsymbol{x}}(\boldsymbol{\mu})}{\langle \boldsymbol{x} \oplus \boldsymbol{i}^\star(\boldsymbol{\mu}), \boldsymbol{\omega}^{-1}\rangle}$.*

*(d) $\|\boldsymbol{\ell}_{\boldsymbol{\omega},\boldsymbol{\mu}}(\alpha_{\boldsymbol{x}}^\star)\|_1 \leq L_{\boldsymbol{\omega},\boldsymbol{\mu}} = 4D^2 K \|\boldsymbol{\mu}\|_\infty^2 \|\boldsymbol{\omega}^{-1}\|_\infty$.*

From Proposition 1 (c), strong duality holds for the program defining $f_{\boldsymbol{x}}(\boldsymbol{\omega}, \boldsymbol{\mu})$, and we conclude:

$$F_{\boldsymbol{\mu}}(\boldsymbol{\omega}) = \min_{\boldsymbol{x} \neq \boldsymbol{i}^\star} \max_{\alpha \geq 0} g_{\boldsymbol{\omega},\boldsymbol{\mu}}(\boldsymbol{x}, \alpha). \tag{3}$$

$F_{\boldsymbol{\mu}}(\boldsymbol{\omega})$ can hence be seen as the value in a two-player game. The aforementioned properties of the Lagrange dual function will help to compute this value.

### 3.2 Solving the two-player game with no regret

There is a rich and growing literature on solving zero-sum games using no-regret algorithms, see for example [RS13, ALLW18, DFG21, ZODS21]. Our game has the particularity that the $\boldsymbol{x}$-player has a discrete combinatorial action set whereas the $\alpha$-player has a convex action set. Importantly, for this game, we wish not only to estimate its value $F_{\boldsymbol{\mu}}(\boldsymbol{\omega})$ but also an *equilibrium* action $\boldsymbol{x}_e$ such that $F_{\boldsymbol{\mu}}(\boldsymbol{\omega}) = \max_{\alpha \geq 0} g_{\boldsymbol{\omega},\boldsymbol{\mu}}(\boldsymbol{x}_e, \alpha)$. Indeed, an estimate of $\boldsymbol{x}_e$ will be needed when implementing the Frank-Wolfe algorithm and more specifically when estimating the gradient of $F_{\boldsymbol{\mu}}(\boldsymbol{\omega})$. To return such an estimate, one could think of leveraging results from the recent literature on last-iterate convergence, see e.g. [DP19, GPDO20, LNP+21, WLZL21, APFS22, AAS+23]. However, most of these results concern saddle-point problems only, and are not applicable in our setting. Here, we adopt a much simpler solution, and take advantage of the properties of the Lagrange dual function $g_{\boldsymbol{\omega},\boldsymbol{\mu}}(\boldsymbol{x}, \alpha)$ to design an iterative procedure directly leading to estimates of $(F_{\boldsymbol{\mu}}(\boldsymbol{\omega}), \boldsymbol{x}_e)$. In this procedure, the two players successively update their actions until a stopping criterion is met, say up to the $N$-th iterations. The procedure generates a sequence $\{(\boldsymbol{x}^{(n)}, \alpha^{(n)})\}_{1 \leq n \leq N}$, and from this sequence, estimates $(\hat{F}, \hat{\boldsymbol{x}})$ of $(F_{\boldsymbol{\mu}}(\boldsymbol{\omega}), \boldsymbol{x}_e)$. The details of the resulting `MCP` algorithm are presented in Algorithm 1.

**$\boldsymbol{x}$-player.** We use a variant of the Follow-the-Perturbed-Leader (FTPL) algorithm [Han57, KV05]. The $\boldsymbol{x}$-player updates her action as follows:

$$\boldsymbol{x}^{(n)} \in \underset{\boldsymbol{x} \neq \boldsymbol{i}^\star}{\operatorname{argmin}} \left( \sum_{m=1}^{n-1} g_{\boldsymbol{\omega},\boldsymbol{\mu}}(\boldsymbol{x}, \alpha^{(m)}) + \left\langle \frac{\boldsymbol{\mathcal{Z}}_n}{\eta_n}, \boldsymbol{x} \right\rangle \right) = \underset{\boldsymbol{x} \neq \boldsymbol{i}^\star}{\operatorname{argmin}} \left( \left\langle \sum_{m=1}^{n-1} \boldsymbol{\ell}_{\boldsymbol{\omega},\boldsymbol{\mu}}(\alpha^{(m)}) + \frac{\boldsymbol{\mathcal{Z}}_n}{\eta_n}, \boldsymbol{x} \right\rangle \right),$$

where $\boldsymbol{\mathcal{Z}}_n$ is a random vector, exponentially distributed and with unit mean ($\{\boldsymbol{\mathcal{Z}}_n\}_{n \geq 1}$ are i.i.d.). Compared to the standard FTPL algorithm, we vary learning rate $\eta_n$ over time to get *anytime* guarantees (as we do not know a priori when the iterative procedure will stop). This kind of time-varying learning rate was also used in [Neu15] with a similar motivation. Note that thanks to the linearity of $g_{\boldsymbol{\omega},\boldsymbol{\mu}}$ and Lemma 2, the $\boldsymbol{x}$-player update can be computed using at most $D$ calls to the `LM` Oracle.

**$\alpha$-player and `MCP` outputs.** From Proposition 1, $f_{\boldsymbol{x}}(\boldsymbol{\omega}, \boldsymbol{\mu}) = \max_{\alpha \geq 0} g_{\boldsymbol{\omega},\boldsymbol{\mu}}(\boldsymbol{x}, \alpha)$. This suggests that the $\alpha$-player can just implement a best-response strategy: after the $\boldsymbol{x}$-player action $\boldsymbol{x}^{(n)}$ is selected, the $\alpha$-player chooses $\alpha^{(n)} = \alpha_{\boldsymbol{x}^{(n)}}^\star = \frac{\triangle_{\boldsymbol{x}^{(n)}}(\boldsymbol{\mu})}{\langle \boldsymbol{x}^{(n)} \oplus \boldsymbol{i}^\star(\boldsymbol{\mu}), \boldsymbol{\omega}^{-1}\rangle}$. This choice ensures that $f_{\boldsymbol{x}^{(n)}}(\boldsymbol{\omega}, \boldsymbol{\mu}) = g_{\boldsymbol{\omega},\boldsymbol{\mu}}(\boldsymbol{x}^{(n)}, \alpha^{(n)})$, and suggests natural outputs for `MCP`: should it stops after $N$ iterations, it can return $\hat{F} = \min_{n \in [N]} g_{\boldsymbol{\omega},\boldsymbol{\mu}}(\boldsymbol{x}^{(n)}, \alpha^{(n)})$ and $\hat{\boldsymbol{x}} \in \operatorname{argmin}_{n \in [N]} g_{\boldsymbol{\omega},\boldsymbol{\mu}}(\boldsymbol{x}^{(n)}, \alpha^{(n)})$.

**Stopping criterion.** The design of the `MCP` stopping criterion relies on the convergence analysis and regret from the $\boldsymbol{x}$-player perspective of the above iterative procedure, which we present in the next subsection. This convergence will be controlled by $\boldsymbol{\ell}_{\boldsymbol{\omega},\boldsymbol{\mu}}(\alpha_{\boldsymbol{x}}^\star)$ and its upper bound $L_{\boldsymbol{\omega},\boldsymbol{\mu}}$ derived in Proposition 1. Introducing $c_\theta = L_{\boldsymbol{\omega},\boldsymbol{\mu}}(4\sqrt{K(\ln K + 1)} + \sqrt{\ln(\theta^{-1})/2})$, the `MCP` stopping criterion is: $\sqrt{n} > c_\theta(1 + \epsilon)/(\epsilon\hat{F})$. Since $\sqrt{n}$ strictly increases with $n$ and since $\hat{F} \geq F_{\boldsymbol{\mu}}(\boldsymbol{\omega})$, this criterion ensures that the algorithm terminates in a finite number of iterates. Moreover, as shown in the next subsection, it also ensures that $\hat{F}$ returned by `MCP` is an $(1 + \epsilon)$-approximation of $F_{\boldsymbol{\mu}}(\boldsymbol{\omega})$ with probability at least $1 - \theta$.

**Algorithm 1:** $(\epsilon, \theta)$-MCP$(\boldsymbol{\omega}, \boldsymbol{\mu})$

---

**initialization:** $n = 1, \hat{F} = \infty, c_\theta = L_{\boldsymbol{\omega}, \boldsymbol{\mu}} \left( 4\sqrt{K(\ln K + 1)} + \sqrt{\ln(\theta^{-1})/2} \right)$;

**while** $(n = 1)$ **or** $(n > 1$ **and** $\sqrt{n} \leq c_\theta(1 + \epsilon)/(\epsilon\hat{F}))$ **do**

    Sample $\boldsymbol{\mathcal{Z}}_n \sim \exp(1)^K$ and set $\eta_n = \sqrt{K(\ln K + 1)/(4nL_{\boldsymbol{\omega}, \boldsymbol{\mu}}^2)}$;

    $\boldsymbol{x}^{(n)} \leftarrow \operatorname{argmin}_{\boldsymbol{x} \neq i^\star(\boldsymbol{\mu})} \left( \sum_{m=1}^{n-1} g_{\boldsymbol{\omega}, \boldsymbol{\mu}}(\boldsymbol{x}, \alpha^{(m)}) + \langle \boldsymbol{\mathcal{Z}}_n, \boldsymbol{x} \rangle / \eta_n \right)$ *(ties broken arbitrarily)*;

    $\alpha^{(n)} \leftarrow \operatorname{argmax}_{\alpha \geq 0} g_{\boldsymbol{\omega}, \boldsymbol{\mu}}(\boldsymbol{x}^{(n)}, \alpha)$ *(uniqueness ensured by Proposition 1 (c))*;

    **if** $g_{\boldsymbol{\omega}, \boldsymbol{\mu}}(\boldsymbol{x}^{(n)}, \alpha^{(n)}) < \hat{F}$ **then** $(\hat{F}, \hat{\boldsymbol{x}}) \leftarrow (g_{\boldsymbol{\omega}, \boldsymbol{\mu}}(\boldsymbol{x}^{(n)}, \alpha^{(n)}), \boldsymbol{x}^{(n)})$ ;

    $n \leftarrow n + 1$;

**end**

**return** $(\hat{F}, \hat{\boldsymbol{x}})$;

---

### 3.3 Performance analysis of the MCP algorithm

We start the analysis by quantifying the regret from the $\boldsymbol{x}$-player perspective of MCP before its stops. The following lemma is proved in Appendix C.3.

**Lemma 3.** *Let* $N \in \mathbb{N}$. *Under* $(\epsilon, \theta)$-MCP$(\boldsymbol{\omega}, \boldsymbol{\mu})$,

$$\mathbb{P}\left[ \frac{1}{N} \sum_{n=1}^{N} g_{\boldsymbol{\omega}, \boldsymbol{\mu}}(\boldsymbol{x}^{(n)}, \alpha^{(n)}) - \frac{1}{N} \min_{\boldsymbol{x} \neq i^\star} \sum_{n=1}^{N} g_{\boldsymbol{\omega}, \boldsymbol{\mu}}(\boldsymbol{x}, \alpha^{(n)}) \leq \frac{c_\theta}{\sqrt{N}} \right] \geq 1 - \theta.$$

Observe that on the one hand,

$$\frac{1}{N} \sum_{n=1}^{N} g_{\boldsymbol{\omega}, \boldsymbol{\mu}}(\boldsymbol{x}^{(n)}, \alpha^{(n)}) \geq \min_{n \in [N]} g_{\boldsymbol{\omega}, \boldsymbol{\mu}}(\boldsymbol{x}^{(n)}, \alpha^{(n)}) = \hat{F} \tag{4}$$

always holds. On the other hand, if $\boldsymbol{x}_e \in \operatorname{argmin}_{\boldsymbol{x} \neq i^\star} \max_{\alpha \geq 0} g_{\boldsymbol{\omega}, \boldsymbol{\mu}}(\boldsymbol{x}, \alpha)$, then we have:

$$\frac{1}{N} \min_{\boldsymbol{x} \neq i^\star} \sum_{n=1}^{N} g_{\boldsymbol{\omega}, \boldsymbol{\mu}}(\boldsymbol{x}, \alpha^{(n)}) \leq \frac{1}{N} \sum_{n=1}^{N} g_{\boldsymbol{\omega}, \boldsymbol{\mu}}(\boldsymbol{x}_e, \alpha^{(n)}) \leq \max_{\alpha \geq 0} g_{\boldsymbol{\omega}, \boldsymbol{\mu}}(\boldsymbol{x}_e, \alpha) = F_{\boldsymbol{\mu}}(\boldsymbol{\omega}). \tag{5}$$

We conclude that for $N$ such that $\sqrt{N} \geq \frac{c_\theta(1+\epsilon)}{\epsilon\hat{F}}$, Lemma 3 together with the inequalities (4) and (5) imply that $\hat{F} - F_{\boldsymbol{\mu}}(\boldsymbol{\omega}) \leq \frac{c_\theta}{\sqrt{N}} \leq \frac{\epsilon\hat{F}}{1+\epsilon}$ holds with probability at least $1 - \theta$. Hence $\mathbb{P}\left[ \hat{F} \leq (1+\epsilon)F_{\boldsymbol{\mu}}(\boldsymbol{\omega}) \right] \geq 1 - \theta$. From this observation, we essentially deduce the following theorem, whose complete proof is given in Appendix C.2.

**Theorem 3.** *Let* $\epsilon, \theta \in (0, 1)$. *Under Assumption 1, for any* $(\boldsymbol{\omega}, \boldsymbol{\mu}) \in \Sigma_+ \times \Lambda$, *the* $(\epsilon, \theta)$-MCP$(\boldsymbol{\omega}, \boldsymbol{\mu})$ *algorithm outputs* $(\hat{F}, \hat{\boldsymbol{x}})$ *satisfying* $\mathbb{P}\left[ F_{\boldsymbol{\mu}}(\boldsymbol{\omega}) \leq \hat{F} \leq (1+\epsilon)F_{\boldsymbol{\mu}}(\boldsymbol{\omega}) \right] \geq 1 - \theta$ *and* $\hat{F} = \max_{\alpha \geq 0} g_{\boldsymbol{\omega}, \boldsymbol{\mu}}(\hat{\boldsymbol{x}}, \alpha)$. *Moreover, the number of* LM *Oracle calls the algorithm does is almost surely at most* $\left\lceil \frac{c_\theta^2(1+\epsilon)^2}{\epsilon^2 F_{\boldsymbol{\mu}}(\boldsymbol{\omega})^2} \right\rceil = \mathcal{O}\left( \frac{\|\boldsymbol{\mu}\|_\infty^4 \|\boldsymbol{\omega}^{-1}\|_\infty^2 K^3 D^5 \ln K \ln \theta^{-1}}{\epsilon^2 F_{\boldsymbol{\mu}}(\boldsymbol{\omega})^2} \right)$.

## 4 The Perturbed Frank-Wolfe Sampling (P-FWS) algorithm

To identify an optimal sampling strategy, rather than solving the lower-bound problem in each round as a Track-and-Stop algorithm would [GK16], we devise P-FWS, an algorithm that performs a single iteration of the Frank-Wolfe algorithm for the lower-bound problem instantiated with an estimator of $\boldsymbol{\mu}$. This requires us to first smooth the objective function $F_{\boldsymbol{\mu}}(\boldsymbol{\omega}) = \min_{\boldsymbol{x} \neq i^\star} f_{\boldsymbol{x}}(\boldsymbol{\omega}, \boldsymbol{\mu})$ (the latter is not differentiable at points $\boldsymbol{\omega}$ where the min is achieved for several sub-optimal actions $\boldsymbol{x}$). To this aim, we cannot leverage the same technique as in [WTP21], where $r$-subdifferential subspaces are built from gradients of $f_{\boldsymbol{x}}(\boldsymbol{\omega}, \boldsymbol{\mu})$. These subspaces could indeed be generated by a number of vectors (here gradients) exponentially growing with $K$. Instead, to cope with the combinatorial

decision sets, `P-FWS` applies more standard stochastic smoothing techniques as described in the next subsection. All the ingredients of `P-FWS` are gathered in §4.2. By design, the algorithm just leverages the `MCP` algorithm as a subroutine, and hence only requires the `LM` Oracle. In §4.3, we analyze the performance of `P-FWS`.

## 4.1 Smoothing the objective function $F_{\boldsymbol{\mu}}$

Here, we present and analyze a standard stochastic technique to smooth a function $\Phi$. In `P-FWS`, this technique will be applied to the objective function $\Phi = F_{\boldsymbol{\mu}}$. Let $\Phi : \mathbb{R}_{>0}^K \mapsto \mathbb{R}$ be a concave and $\ell$-Lipschitz function. Assume that the set of points where $\Phi$ is not differentiable is of Lebesgue-measure zero. To smooth $\Phi$, we can take its average value in a neighborhood of the point considered, see e.g. [FKM05]. Formally, we define the *stochastic smoothed* approximate of $\Phi$ as:

$$\bar{\Phi}_\eta(\boldsymbol{\omega}) = \mathbb{E}_{\boldsymbol{\mathcal{Z}} \sim \text{Uniform}(B_2)}[\Phi(\boldsymbol{\omega} + \eta \boldsymbol{\mathcal{Z}})], \tag{6}$$

where $B_2 = \{\boldsymbol{v} \in \mathbb{R}^K : \|\boldsymbol{v}\|_2 \leq 1\}$ and $\eta \in (0, \min_{k \in [K]} \omega_k)$. The following proposition lists several properties of this smoothed function, and gathers together some of the results from [DBW12], see Appendix H for more details.

**Proposition 2.** *For any $\boldsymbol{\omega} \in \Sigma_+$ and $\eta \in (0, \min_{k \in [K]} \omega_k)$, $\bar{\Phi}_\eta$ satisfies: (i) $\Phi(\boldsymbol{\omega}) - \eta\ell \leq \bar{\Phi}_\eta(\boldsymbol{\omega}) \leq \Phi(\boldsymbol{\omega})$; (ii) $\nabla \bar{\Phi}_{\boldsymbol{\mu},\eta}(\boldsymbol{\omega}) = \mathbb{E}_{\boldsymbol{\mathcal{Z}} \sim Uniform(B_2)}[\nabla \Phi_{\boldsymbol{\mu}}(\boldsymbol{\omega} + \eta \boldsymbol{\mathcal{Z}})]$; (iii) $\bar{\Phi}_\eta$ is $\frac{\ell K}{\eta}$-smooth; (iv) if $\eta > \eta' > 0$, then $\bar{\Phi}_{\eta'}(\boldsymbol{\omega}) \geq \bar{\Phi}_\eta(\boldsymbol{\omega})$.*

Note that with (i), we may control the approximation error between $\bar{\Phi}_\eta$ and $\Phi$ by $\eta$. (ii) and (iii) ensure the differentiability and smoothness of $\bar{\Phi}_\eta$ respectively. (iii) is equivalent to $\bar{\Phi}_\eta(\boldsymbol{\omega}') \leq \bar{\Phi}_\eta(\boldsymbol{\omega}) + \langle \nabla \bar{\Phi}_\eta(\boldsymbol{\omega}), \boldsymbol{\omega}' - \boldsymbol{\omega} \rangle + \frac{\ell K}{2\eta} \|\boldsymbol{\omega} - \boldsymbol{\omega}'\|_2^2$, $\forall \boldsymbol{\omega}, \boldsymbol{\omega}' \in \Sigma_+$. Finally, (iv) stems from the concavity of $\Phi$, and implies that the value $\bar{\Phi}_\eta(\boldsymbol{\omega})$ monotonously increases while $\eta$ decreases, and it is upper bounded by $\Phi(\boldsymbol{\omega})$ thanks to (i). The above results hold for $\Phi = F_{\boldsymbol{\mu}}$. Indeed, first it is clear that the definition (2) of $F_{\boldsymbol{\mu}}$ can be extended to $\mathbb{R}^K$; then, it can be shown that $F_{\boldsymbol{\mu}}$ is Lipschitz-continuous and almost-everywhere differentiable – refer to Appendices I and H for formal proofs.

## 4.2 The algorithm

Before presenting `P-FWS`, we need to introduce the following notation. For $t \geq 1$, $k \in [K]$, we define $N_k(t) = \sum_{s=1}^t \mathbb{1}\{x_k(s) = 1\}$, $\hat{\omega}_k(t) = N_k(t)/t$, and $\hat{\mu}_k(t) = \sum_{s=1}^t y_k(s) \mathbb{1}\{x_k(s) = 1\} / N_k(t)$ when $N_k(t) > 0$.

**Sampling rule.** The design of the sampling rule is driven by the following objectives: (i) the empirical allocation should converge to the solution of the lower-bound problem (1), and (ii) the number of calls to the `LM` Oracle should be controlled. To meet the first objective, we need in the Frank-Wolfe updates to plug an accurate estimator of $\boldsymbol{\mu}$ in. The accuracy of our estimator will be guaranteed by alternating between *forced exploration* and *FW update* sampling phases. Now for the second objective, we also use forced exploration phases when in a Frank-Wolfe update, the required number of calls to the `LM` Oracle predicted by the upper bound presented in Theorem 3 is too large. In view of Lemma 1 and Theorem 3, this happens in round $t$ if $\|\hat{\boldsymbol{\mu}}(t-1)\|_\infty$ or $\triangle_{\min}(\hat{\boldsymbol{\mu}}(t-1))^{-1}$ is too large. Next, we describe the forced exploration and Frank-Wolfe update phases in detail.

*Forced exploration.* Initially, `P-FWS` applies the `LM` Oracle to compute the *forced exploration set* $\mathcal{X}_0 = \{\boldsymbol{i}^\star(\boldsymbol{e}_k) : k \in [K]\}$, where $\boldsymbol{e}_k$ is the $K$-dimensional vector whose $k$-th component is equal to one and zero elsewhere. `P-FWS` then selects each action in $\mathcal{X}_0$ once. Note that Assumption 1 (iii) ensures that the $k$-th component of $\boldsymbol{i}^\star(\boldsymbol{e}_k)$ is equal to one. In turn, this ensures that $\mathcal{X}_0$ is a $[K]$-covering set, and that playing actions from $\mathcal{X}_0$ is enough to estimate $\boldsymbol{\mu}$. `P-FWS` starts an exploration phase at rounds $t$ such that $\sqrt{t/|\mathcal{X}_0|}$ is an integer or such that the maximum of $\triangle_{\min}(\hat{\boldsymbol{\mu}}(t-1))^{-1}$ and $\|\hat{\boldsymbol{\mu}}(t-1)\|_\infty$ is larger than $\sqrt{t-1}$. Whenever this happens, `P-FWS` pulls each $\boldsymbol{x} \in \mathcal{X}_0$ once.

*Frank-Wolfe updates.* When in round $t$, the algorithm is not in a forced exploration phase, it implements an iteration of the Frank-Wolfe algorithm applied to maximize the smoothed function $\bar{F}_{\hat{\boldsymbol{\mu}}(t-1),\eta_t}(\hat{\boldsymbol{\omega}}(t-1)) = \mathbb{E}_{\boldsymbol{\mathcal{Z}} \sim \text{Uniform}(B_2)}[F_{\hat{\boldsymbol{\mu}}(t-1)}(\hat{\boldsymbol{\omega}}(t-1) + \eta_t \boldsymbol{\mathcal{Z}})]$. The sequence of parameters $\{\eta_t\}_{t \geq 1}$ is chosen to ensure that $\eta_t$ chosen in $(0, \min_k \hat{\omega}_k(t))$, and hence $\hat{\boldsymbol{\omega}}(t-1) + \eta_t \boldsymbol{\mathcal{Z}} \in \mathbb{R}_{>0}^K$. Also note that in a round $t$ where the algorithm is not in a forced exploration phase,

**Algorithm 2:** P-FWS $(\{(\epsilon_t, \eta_t, n_t, \rho_t, \theta_t)\}_t)$

---

**initialization:**
> **for** $k = 1, \ldots, K$ **do**
>> $\mathcal{X}_0 \leftarrow \operatorname{argmax}_{\boldsymbol{x} \in \mathcal{X}} \langle \boldsymbol{e}_k, \boldsymbol{x} \rangle$ (tie broken arbitrarily)
>
> **end**
> Sample $\boldsymbol{x} \in \mathcal{X}_0$ in a round-robin manner for $4|\mathcal{X}_0|$ rounds; update $\hat{\boldsymbol{\mu}}(4|\mathcal{X}_0|)$ and $\hat{\boldsymbol{\omega}}(4|\mathcal{X}_0|)$;

**for** $t = 4|\mathcal{X}_0| + 1, \cdots$ **do**
> **if** $\sqrt{t/|\mathcal{X}_0|} \in \mathbb{N}$ **or** $\max\{\triangle_{\min}(\hat{\boldsymbol{\mu}}(t-1))^{-1}, \|\hat{\boldsymbol{\mu}}(t-1)\|_\infty\} > \sqrt{t-1}$ **then**
>> Sample each $\boldsymbol{x} \in \mathcal{X}_0$ once, update $\hat{\boldsymbol{\mu}}(t)$ and $\hat{\boldsymbol{\omega}}(t)$, and $t \leftarrow t + |\mathcal{X}_0| - 1$;
>
> **else**
>> Compute $\nabla \tilde{F}_{\hat{\boldsymbol{\mu}}(t-1), \eta_t, n_t}(\hat{\boldsymbol{\omega}}(t-1))$ by $(\rho_t, \theta_t)$-MCP algorithm;
>> $\boldsymbol{x}(t) \leftarrow \boldsymbol{i}^\star\left(\nabla \tilde{F}_{\hat{\boldsymbol{\mu}}(t-1), \eta_t, n_t}(\hat{\boldsymbol{\omega}}(t-1))\right)$;
>> Sample $\boldsymbol{x}(t)$ and update $\hat{\boldsymbol{\mu}}(t)$ and $\hat{\boldsymbol{\omega}}(t)$;
>
> **end**
> **if** $\max\{\triangle_{\min}(\hat{\boldsymbol{\mu}}(t))^{-1}, \|\hat{\boldsymbol{\mu}}(t)\|_\infty\} \leq \sqrt{t}$ **then**
>> $\hat{F}_t \leftarrow \left(\epsilon_t, \delta/t^2\right)$ -MCP$(\hat{\boldsymbol{\omega}}(t), \hat{\boldsymbol{\mu}}(t))$;
>> **if** $t\hat{F}_t > (1 + \epsilon_t)\,\beta\left(t, \left(1 - \frac{1}{4|\mathcal{X}_0|}\right)\delta\right)$ **then** break;

**end**
**return** $\hat{\boldsymbol{i}} = \boldsymbol{i}^\star(\hat{\boldsymbol{\mu}}(t))$;

---

by definition $\triangle_{\min}(\hat{\boldsymbol{\mu}}(t-1)) > 0$. This implies that $\hat{\boldsymbol{\mu}}(t-1) \in \Lambda$ and that $F_{\hat{\boldsymbol{\mu}}(t-1)}$ and $\bar{F}_{\hat{\boldsymbol{\mu}}(t-1), \eta_t}(\hat{\boldsymbol{\omega}}(t-1))$ are well-defined. Now an ideal FW update would consist in playing an action $\boldsymbol{i}^\star(\nabla \bar{F}_{\hat{\boldsymbol{\mu}}(t-1), \eta_t}(\hat{\boldsymbol{\omega}}(t-1))) = \operatorname{argmax}_{\boldsymbol{x} \in \mathcal{X}} \left\langle \nabla \bar{F}_{\hat{\boldsymbol{\mu}}(t-1), \eta_t}(\hat{\boldsymbol{\omega}}(t-1)), \boldsymbol{x} \right\rangle$, see e.g. [Jag13]. Unfortunately, we do not have access to $\nabla \bar{F}_{\hat{\boldsymbol{\mu}}(t-1), \eta_t}(\hat{\boldsymbol{\omega}}(t-1))$. But the latter can be approximated, as suggested in Proposition 2 (ii), by $\nabla \tilde{F}_{\hat{\boldsymbol{\mu}}(t-1), \eta_t, n_t}(\hat{\boldsymbol{\omega}}(t-1)) = \frac{1}{n_t} \sum_{m=1}^{n_t} \nabla f_{\hat{\boldsymbol{x}}_m}(\hat{\boldsymbol{\omega}}(t-1) + \eta_t \boldsymbol{\mathcal{Z}}_m, \hat{\boldsymbol{\mu}}(t-1))$, where $\boldsymbol{\mathcal{Z}}_1, \cdots, \boldsymbol{\mathcal{Z}}_{n_t} \overset{i.i.d.}{\sim}$ Uniform$(B_2)$, $\hat{\boldsymbol{x}}_m$ is the action return by $(\rho_t, \theta_t)$-MCP$(\hat{\boldsymbol{\omega}}(t-1) + \eta_t \boldsymbol{\mathcal{Z}}_m, \hat{\boldsymbol{\mu}}(t-1))$. P-FWS uses this approximation and the LM Oracle to select the action: $\boldsymbol{x}(t) \in \boldsymbol{i}^\star\left(\nabla \tilde{F}_{\hat{\boldsymbol{\mu}}(t-1), \eta_t, n_t}(\hat{\boldsymbol{\omega}}(t-1))\right)$. The choices of the parameters $\eta_t$, $n_t$, $\rho_t$ and $\theta_t$ do matter. $\eta_t$ impacts the sample complexity and should converge to 0 as $t \to \infty$ so that $\bar{F}_{\boldsymbol{\mu}, \eta_t}(\boldsymbol{\omega}) \to F_{\boldsymbol{\mu}}(\boldsymbol{\omega})$ at any point $\boldsymbol{\omega} \in \Sigma_+$ (this is a consequence of Proposition 2 (i)(iv)). $\eta_t$ should not decay too fast however as it would alter the smoothness of $\bar{F}_{\boldsymbol{\mu}, \eta_t}$. We will show that $\eta_t$ should actually decay as $1/\sqrt{t}$. $(n_t, \rho_t, \theta_t)$ impact the trade-off between the sample complexity and the computational complexity of the algorithm. We let $n_t \to \infty$ and $(\rho_t, \theta_t) \to 0$ as $t \to \infty$ so that $\left\langle \nabla \tilde{F}_{\boldsymbol{\mu}, \eta_t, n_t}(\boldsymbol{\omega}) - \nabla \bar{F}_{\boldsymbol{\mu}, \eta_t}(\boldsymbol{\omega}), \boldsymbol{x} \right\rangle \to 0$ for any $(\boldsymbol{\omega}, \boldsymbol{x}) \in \Sigma_+ \times \mathcal{X}$.

**Stopping and decision rule.** As often in best arm identification algorithms, the P-FWS stopping rule takes the form of a GLRT:

$$\tau = \inf\left\{ t > 4|\mathcal{X}_0| : \frac{t\hat{F}_t}{1 + \epsilon_t} > \beta\left(t, \left(1 - \frac{1}{4|\mathcal{X}_0|}\right)\delta\right), \max\left\{\triangle_{\min}(\hat{\boldsymbol{\mu}}(t))^{-1}, \|\hat{\boldsymbol{\mu}}(t)\|_\infty\right\} \leq \sqrt{t} \right\},$$
$$(7)$$

where $\epsilon_t \in \mathbb{R}_{>0}$, $\hat{F}_t$ is returned by the $(\epsilon_t, \delta/t^2)$-MCP$(\hat{\boldsymbol{\omega}}(t), \hat{\boldsymbol{\mu}}(t))$ algorithm. The function $\beta$ satisfies

$$\forall t \geq 1, \quad \left(tF_{\hat{\boldsymbol{\mu}}(t)}(\boldsymbol{\omega}(t)) \geq \beta(t, \delta)\right) \implies \left(\mathbb{P}_{\boldsymbol{\mu}}[\boldsymbol{i}^\star(\hat{\boldsymbol{\mu}}(t)) \neq \boldsymbol{i}^\star(\boldsymbol{\mu})] \leq \delta\right), \tag{8}$$

$$\exists c_1, c_2 > 0 : \quad \forall t \geq c_1, \beta(t, \delta) \leq \ln\left(\frac{c_2 t}{\delta}\right). \tag{9}$$

Examples of function $\beta$ satisfying the above conditions can be found in [GK16, JP20, KK21]. The condition (8) will ensure the $\delta$-correctness of P-FWS, whereas (9) will control its sample complexity. Finally, the action returned by P-FWS is simply defined as $\hat{\boldsymbol{i}} = \boldsymbol{i}^\star(\hat{\boldsymbol{\mu}}(\tau))$. The complete pseudo-code of P-FWS is presented in Algorithm 2.[3]

---

[3] Our Julia implementation could be found at https://github.com/rctzeng/NeurIPS2023-PerturbedFWS.

## 4.3 Non-asymptotic performance analysis of `P-FWS`

The following theorem provides an upper bound of the sample complexity of `P-FWS` valid for any confidence level $\delta$, as well as the computational complexity of the algorithm.

**Theorem 4.** *Let $\boldsymbol{\mu} \in \Lambda$ and $\delta \in (0,1)$. If `P-FWS` is parametrized using*

$$(\epsilon_t, \eta_t, n_t, \rho_t, \theta_t) = \left( t^{-\frac{1}{9}}, \frac{1}{4\sqrt{t|\mathcal{X}_0|}}, \left\lceil t^{\frac{1}{4}} \right\rceil, \frac{1}{16tD^2|\mathcal{X}_0|}, \frac{1}{t^{\frac{1}{4}}e^{\sqrt{t}}} \right), \tag{10}$$

*then (i) the algorithm finishes in finite time almost surely and $\mathbb{P}_{\boldsymbol{\mu}}[\hat{\imath} \neq \boldsymbol{i}^{\star}(\boldsymbol{\mu})] \leq \delta$; (ii) its sample complexity satisfies $\mathbb{P}_{\boldsymbol{\mu}}\left[\limsup_{\delta \to 0} \frac{\tau}{\ln \delta^{-1}} \leq T^{\star}(\boldsymbol{\mu})\right] = 1$ and for any $\epsilon, \tilde{\epsilon} \in (0,1)$ with $\epsilon < \min\{1, \frac{2D^2\triangle_{\min}^2}{K}, \frac{D^2\|\boldsymbol{\mu}\|_{\infty}^2}{3}\}$,*

$$\mathbb{E}_{\boldsymbol{\mu}}[\tau] \leq \frac{(1+\tilde{\epsilon})^2}{T^{\star}(\boldsymbol{\mu})^{-1} - 6\epsilon} \times H\left( \frac{1}{\delta} \cdot \frac{4c_2}{3} \cdot \frac{(1+\tilde{\epsilon})^2}{T^{\star}(\boldsymbol{\mu})^{-1} - 6\epsilon} \right) + \Psi(\epsilon, \tilde{\epsilon}),$$

*where $H(x) = \ln x + \ln\ln x + 1$ and $\Psi(\epsilon, \tilde{\epsilon})$ (refer to (34) for a detailed expression) is polynomial in $\epsilon^{-1}, \tilde{\epsilon}^{-1}, K, \|\boldsymbol{\mu}\|_{\infty}$, and $\triangle_{\min}(\boldsymbol{\mu})^{-1}$; (iii) the expected number of `LM` Oracle calls is upper bounded by a polynomial in $\ln \delta^{-1}, K, \|\boldsymbol{\mu}\|_{\infty}$, and $\triangle_{\min}(\boldsymbol{\mu})^{-1}$.*

The above theorem establishes the statistical asymptotic optimality of `P-FWS` since it implies that $\limsup_{\delta \to 0} \mathbb{E}_{\boldsymbol{\mu}}[\tau] / \ln(1/\delta) \leq (1+\tilde{\epsilon})^2/(T^{\star}(\boldsymbol{\mu})^{-1} - 6\epsilon)$. This upper bound matches the sample complexity lower bound (1) when $\epsilon \to 0$ and $\tilde{\epsilon} \to 0$.

**Proof sketch.** The complete proof of Theorem 4 is presented in Appendix D.

*(i) Correctness.* To establish the $\delta$-correctness of the algorithm, we introduce the event $\mathcal{G}$ under which $\hat{F}_t$, computed by $(\epsilon_t, \delta/t^2)$-`MCP`$(\hat{\boldsymbol{\omega}}(t), \hat{\boldsymbol{\mu}}(t))$, is an $(1+\epsilon_t)$-approximation of $F_{\hat{\boldsymbol{\mu}}(t)}(\hat{\boldsymbol{\omega}}(t))$ in each round $t \geq 4|\mathcal{X}_0| + 1$. From Theorem 3, we deduce that $\mathbb{P}_{\boldsymbol{\mu}}[\mathcal{G}^c] \leq \sum_{t=4|\mathcal{X}_0|+1}^{\infty} \delta/t^2 \leq \delta/4|\mathcal{X}_0|$. In view of (8), this implies that $\mathbb{P}_{\boldsymbol{\mu}}[\hat{\imath} \neq \boldsymbol{i}^{\star}(\boldsymbol{\mu})] \leq \delta$.

*(ii) Non-asymptotic sample complexity upper bound.*

Step 1. (Concentration and certainty equivalence) We first define two *good* events, $\mathcal{E}_t^{(1)}$ and $\mathcal{E}_t^{(2)}$. $\mathcal{E}_t^{(2)}$ corresponds to the event where $\hat{\boldsymbol{\mu}}(t)$ is close to $\boldsymbol{\mu}$, and its occurrence probability can be controlled using the forced exploration rounds and concentration inequalities. Under $\mathcal{E}_t^{(1)}$, the selected action $\boldsymbol{x}(t)$ is close to the ideal FW update. Again using concentration results and the performance guarantees of `MCP` given in Theorem 3, we can control the occurrence probability of $\mathcal{E}_t^{(2)}$. Overall, we show that $\sum_{t=1}^{\infty} \mathbb{P}_{\boldsymbol{\mu}}[(\mathcal{E}_t^{(1)} \cap \mathcal{E}_t^{(2)})^c] < \infty$. To this aim, we derive several important continuity results presented in Appendix G. These results essentially allow us to study the convergence of the smoothed FW updates as if the certainty equivalence principle held, i.e., as if $\hat{\boldsymbol{\mu}}(t) = \boldsymbol{\mu}$.

Step 2. (Convergence of the smoothed FW updates) We study the convergence assuming that $(\mathcal{E}_t^{(1)} \cap \mathcal{E}_t^{(2)})$ holds. We first show that $\bar{F}_{\boldsymbol{\mu}, \eta_t}$ is $\ell$-Lipschitz and smooth for $\ell = 2D^2 \|\boldsymbol{\mu}\|_{\infty}^2$, see Appendices H and I. Then, in Appendix E, we establish that the dynamics of $\phi_t = \max_{\boldsymbol{\omega} \in \Sigma} F_{\boldsymbol{\mu}}(\boldsymbol{\omega}) - F_{\boldsymbol{\mu}}(\hat{\boldsymbol{\omega}}(t))$ satisfy $t\phi_t \leq (t-1)\phi_{t-1} + \ell\left(\eta_{t-1} + \frac{K^2}{2t\eta_t}\right)$. Observe that, as mentioned earlier, $1/\sqrt{t}$ is indeed the optimal scaling choice for $\eta_t$. We deduce that after a certain finite number $T_1$ of rounds, $\phi_t$ is sufficiently small and $\max\{\triangle_{\min}(\hat{\boldsymbol{\mu}}(t))^{-1}, \|\hat{\boldsymbol{\mu}}(t)\|_{\infty}\} \leq \sqrt{t}$.

Step 3. Finally, we observe that $\mathbb{E}_{\boldsymbol{\mu}}[\tau] \leq T_1 + \sum_{t=T_1}^{\infty} \mathbb{P}_{\boldsymbol{\mu}}\left[t\hat{F}_t \leq (1+\epsilon_t)\beta\left(t, (1 - \frac{1}{4|\mathcal{X}_0|})\delta\right)\right] + \sum_{t=T_1+1}^{\infty} \mathbb{P}_{\boldsymbol{\mu}}\left[(\mathcal{E}_t^{(1)} \cap \mathcal{E}_t^{(2)})^c\right]$, and show that the second term in the r.h.s. in this inequality is equivalent to $T^{\star}(\boldsymbol{\mu}) \ln(1/\delta)$ as $\delta \to 0$ using the property of the function $\beta$ defining the stopping threshold and similar arguments as those used in [GK16, WTP21].

*(iii) Expected number of `LM` Oracle calls.* The `MCP` algorithm is called to compute $\hat{F}_t$ and to perform the FW update only in rounds $t$ such that $\max\{\triangle_{\min}(\hat{\boldsymbol{\mu}}(t))^{-1}, \|\hat{\boldsymbol{\mu}}(t)\|_{\infty}\} \leq \sqrt{t}$. Thus, from Theorem 3 and Lemma 1, the number of `LM` Oracle calls per-round is a polynomial in $t$ and $K$. As the $\mathbb{E}_{\boldsymbol{\mu}}[\tau]$ is polynomial (in $\ln \delta^{-1}, K, \|\boldsymbol{\mu}\|_{\infty}$ and $\triangle_{\min}^{-1}$), the expected number of `LM` Oracle calls is also polynomial in the same variables.

## 5    Related Work

We provide an exhaustive survey of the related literature in Appendix B. To summarize, to the best of our knowledge, CombGame [JMKK21] is the state-of-the-art algorithm for BAI in combinatorial semi-bandits in the high confidence regimes. A complete comparison to `P-FWS` is presented in Appendix B. CombGame was initially introduced in [DKM19] for classical bandit problems. There, the lower-bound problem is casted as a two-player game and the authors propose to use no-regret algorithms for each player to solve it. [JMKK21] adapts the algorithm for combinatorial semi-bandits, and provides a non-asymptotic sample complexity upper bound matching (1) asymptotically. However, the resulting algorithm requires to call an oracle solving the Most-Confusing-Parameter problem as our `MCP` algorithm. The authors of [JMKK21] conjectured the existence of such a computationally efficient oracle, and we establish this result here.

## 6    Conclusion

In this paper, we have presented `P-FWS`, the first computationally efficient and statistically optimal algorithm for the best arm identification problem in combinatorial semi-bandits. For this problem, we have studied the computational-statistical trade-off through the analysis of the optimization problem leading to instance-specific sample complexity lower bounds. This approach can be extended to study the computational-statistical gap in other learning tasks. Of particular interest are problems with an underlying structure (e.g. linear bandits [DMSV20, JP20], or RL in linear / low rank MDPs [AKKS20]). Most results on these problems are concerned with statistical efficiency, and ignore computational issues.

## Acknowledgments and Disclosure of Funding

We thank Aristides Gionis and the anonymous reviewers for their valuable feedback. The research is funded by ERC Advanced Grant REBOUND (834862), the Wallenberg AI, Autonomous Systems and Software Program (WASP), and Digital Futures.

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
