# Contents

# A  Table of Notation

| **Problem setting** | |
|---|---|
| $K$ | Number of arms |
| $[m]$ for any $m \in \mathbb{N}$ | The set $\{1, 2 \ldots, m\}$ |
| $\delta$ | Required uncertainty |
| $\boldsymbol{\mu} \in \mathbb{R}^K$ | Vector of the expected rewards of the various arms |
| $\mathbb{E}_{\boldsymbol{\mu}}$ and $\mathbb{P}_{\boldsymbol{\mu}}$ | The expectation and probability measure corresponding to $\boldsymbol{\mu}$ |
| $\Lambda$ | $\{\boldsymbol{\mu} \in \mathbb{R}^K : |\boldsymbol{i}^\star(\boldsymbol{\mu})| = 1\}$ |
| $\boldsymbol{i}^\star(\boldsymbol{\mu})$ | Best arm under parameter $\boldsymbol{\mu}$ |
| $\mathcal{X}$ | Set of actions in $\{0, 1\}^K$ |
| $D$ | $\max_{\boldsymbol{x} \in \mathcal{X}} \|\boldsymbol{x}\|_1$ |
| $\triangle_{\boldsymbol{x}}(\boldsymbol{\mu})$ | $\langle \boldsymbol{i}^\star(\boldsymbol{\mu}) - \boldsymbol{x}, \boldsymbol{\mu} \rangle$ |
| $\triangle_{\min}(\boldsymbol{\mu})$ | $\min_{\boldsymbol{x} \neq \boldsymbol{i}^\star(\boldsymbol{\mu})} \triangle_{\boldsymbol{x}}(\boldsymbol{\mu})$ |

| **Notation related to a given algorithm** | |
|---|---|
| $N_k(t)$ | Number of pulls of arm $k$ up to time $t$ |
| $\hat{\omega}_k(t)$ | $N_k(t)/t$ |
| $\boldsymbol{x}(t)$ | The action taken in time $t$ |
| $y_k(t)$ | Random reward received if $x_k(t) = 1$ |
| $\hat{\mu}_k(t)$ | $\sum_{s=1}^t y_k(s) \mathbb{1}\{x_k(s) = 1\}/N_k(t)$ |
| $\tau$ | Stopping time |
| $\hat{\boldsymbol{i}}$ | Recommended action |

| **Notation used for sets and vectors** | |
|---|---|
| $\odot$ | Elementwise product |
| $\oplus$ | Elementwise sum over $\mathbb{Z}_2$ |
| $\boldsymbol{x}^i$ | The $i$-th elementwise power of $\boldsymbol{x} \in \mathbb{R}^K$, i.e., $(x_k^i)_{k \in [K]}$ |
| $\mathsf{cl}(\mathcal{S})$ | The closure of set $\mathcal{S}$ |
| $\boldsymbol{e}_k$ | the $K$-dimensional vector whose $k$-th component is equal to one and zero elsewhere |

| **Properties for lower bound** | |
|---|---|
| $d(\mu, \mu')$ | KL divergence between the distributions parametrized by $\mu$ and $\mu'$ |
| $\mathsf{kl}(a, b)$ | KL divergence between two Bernoulli distributions of means $a$ and $b$ |
| $\mathsf{Alt}(\boldsymbol{\mu})$ | $\{\boldsymbol{\lambda} \in \Lambda : \boldsymbol{i}^\star(\boldsymbol{\lambda}) \neq \boldsymbol{i}^\star(\boldsymbol{\mu})\}$ |
| $\Sigma$ | $\{\sum_{\boldsymbol{x} \in \mathcal{X}} w_{\boldsymbol{x}} \boldsymbol{x} : \boldsymbol{w} \in \Sigma_{|\mathcal{X}|}\}$ where $\Sigma_N$ is a $(N-1)$-dimensional simplex |
| $\Sigma_+$ | $\Sigma \cap \mathbb{R}_{>0}^K$ |

| **Notation for `MCP`** | |
|---|---|
| $F_{\boldsymbol{\mu}}(\boldsymbol{\omega})$ | $\min_{\boldsymbol{x} \neq \boldsymbol{i}^\star(\boldsymbol{\mu})} f_{\boldsymbol{x}}(\boldsymbol{\omega}, \boldsymbol{\mu})$ |
| $f_{\boldsymbol{x}}(\boldsymbol{\omega}, \boldsymbol{\mu})$ | $\inf_{\boldsymbol{\lambda} \in \mathcal{C}_{\boldsymbol{x}}} \langle \boldsymbol{\omega}, \frac{(\boldsymbol{\mu} - \boldsymbol{\lambda})^2}{2} \rangle$, where $\mathcal{C}_{\boldsymbol{x}} = \{\boldsymbol{\lambda} \in \mathbb{R}^K : \langle \boldsymbol{\lambda}, \boldsymbol{i}^\star(\boldsymbol{\mu}) - \boldsymbol{x} \rangle < 0\}$ |
| $\mathcal{L}_{\boldsymbol{\omega}, \boldsymbol{\mu}}(\boldsymbol{\lambda}, \boldsymbol{x}, \alpha)$ | $\langle \boldsymbol{\omega}, \frac{(\boldsymbol{\mu} - \boldsymbol{\lambda})^2}{2} \rangle + \alpha \langle \boldsymbol{i}^\star(\boldsymbol{\mu}) - \boldsymbol{x}, \boldsymbol{\lambda} \rangle$ |
| $g_{\boldsymbol{\omega}, \boldsymbol{\mu}}(\boldsymbol{x}, \alpha)$ | $\inf_{\boldsymbol{\lambda} \in \mathbb{R}^K} \mathcal{L}_{\boldsymbol{\omega}, \boldsymbol{\mu}}(\boldsymbol{\lambda}, \boldsymbol{x}, \alpha)$ |

| **Notation for `P-FWS`** | |
|---|---|
| $\mathcal{X}_0$ | A $[K]$-covering set |
| $\hat{F}_t$ | `MCP`-approximated value of $F_{\hat{\boldsymbol{\mu}}(t)}(\hat{\boldsymbol{\omega}}(t))$ for stopping rule |
| $\bar{F}_{\boldsymbol{\mu}, \eta}(\cdot)$ | $\mathbb{E}_{\boldsymbol{\mathcal{Z}} \sim \text{Uniform}(B_2)}[\nabla F_{\boldsymbol{\mu}}(\cdot + \eta \boldsymbol{\mathcal{Z}})]$ where $B_2 = \{\boldsymbol{v} \in \mathbb{R}^K : \|\boldsymbol{v}\|_2 \leq 1\}$ |
| $\bar{F}_{\boldsymbol{\mu}, \eta, n}$ | The empirical $n$-sample estimate of $\bar{F}_{\boldsymbol{\mu}, \eta}$ |
| $\ell$ | Lipschitz constant of $F_{\boldsymbol{\mu}}$ |

# B Further related work

Combinatorial semi-bandits [CBL12] have found numerous applications including online ranking [DKC21], network routing [CLK+14, KWA+14], loan assignment [KWA+14], path planning problem [JMKK21], and influence marketing [Per22]). We do not discuss these applications here, but rather focus the literature that is the most relevant to our analysis and results.

**Solving the lower-bound problem in combinatorial semi-bandits.** We are not aware of any computationally efficient algorithm to solve the lower-bound problem, or to compute its objective function. To the best of our knowledge, MCP is the first algorithm to do so. A work closed to ours is [CCG21a] for combinatorial semi-bandits but in the regret minimization. Regret minimization yields a different lower-bound problem. There exits a statistically optimal algorithm [CMP17], called OSSB, that matches the regret lower bound by [CTMSP+15]. OSSB requires to solve the lower-bound problem in each round, and the authors [CCG21a] are the first to investigate whether this is at all possible in a computationally efficient way. They establish that if budgeted-linear maximization (BLM) [RG96, BBGS11] can be solved within an $\varepsilon$-approximation factor for the combinatorial set $\mathcal{X}$, then the lower-bound problem can be approximately solved with a precision depending on $\varepsilon$. As a consequence, the approach leads to an algorithm with asymptotically minimal regret only if one has access to an exact BLM solver. This is the case for $m$-sets and $s$-$t$ paths but this is not the case for spanning trees and perfect matchings. For the latter case, as mentioned [CCG21a], an algorithm using an approximately correct BLM solver would not be statistically optimal.

**Best arm identification in combinatorial semi-bandits.** Many tasks related to combinatorial best arm identification are formulated in the *transductive* setting [JMKK21], where the set $\mathcal{A} \subseteq \{0,1\}^K$ available for exploration is not necessarily the same as the set $\mathcal{X} \subseteq \{0,1\}^K$ for decision. The minimal sample complexity in the transductive setting is exactly (1) with $\Sigma$ replaced with $\{\sum_{\boldsymbol{x} \in \mathcal{A}} w_{\boldsymbol{x}} \boldsymbol{x} : \boldsymbol{\omega} \in \Sigma_{|\mathcal{A}|}\}$ - see (58) in Appendix L for details. Two most studied tasks are combinatorial multi-arm bandit (C-MB) where $\mathcal{A} = \{e_k\}_{k \in [K]}$ and the best action identification (C-BAI) where $\mathcal{A} = \mathcal{X}$. The former is arguably simpler than the latter if we compare the corresponding minimal sample complexities (note that $\Sigma_K \supseteq \Sigma$). We note that our results for C-BAI can be easily generalized to the transductive setting (see Appendix L).

Prior works mainly focus on the C-MB task. UCB-based [KTAS12, CLK+14] and elimination-based [CGL16, CGL+17, KSJJ+20] approaches are popular. Among these, EfficientGapElim [CGL+17] achieves the lowest sample complexity $\mathcal{O}(T^\star(\boldsymbol{\mu})(\ln \delta^{-1} + \ln^2 \triangle_{\min}^{-1}(\ln \ln \triangle_{\min}^{-1} + \ln|\mathcal{X}|))$ with high probability[4], but its computational complexity is hard to analyze. Peace [KSJJ+20], another elimination-based approach by experimental design, requires with high probability a polynomial number of the LM Oracle calls in total. The sample complexity of Peace has a $\delta$-dependent term (scaling as $KT^\star(\boldsymbol{\mu}) \ln \delta^{-1}$) worse than EfficientGapElim. Overall, none of these are statistically optimal when $\delta \to 0$. Note that algorithms for linear best-arm identification [DMSV20, WTP21] are applicable to C-MB but not to C-BAI and the general transductive setting.

For the task of C-BAI, we are only aware of two works: GCB-PE [DKC21] and CombGame [JMKK21]. GCB-PE is a UCB-based algorithm with guarantees on the sample complexity and computational complexity valid with high probability only. CombGame [JMKK21] is proposed for the transductive setting, and its design inherits from [DKM19] that interprets the lower-bound problem and more precisely $T^\star(\boldsymbol{\mu})^{-1}$ as the value of a two-player game (a $\boldsymbol{\omega}$-player and a $\boldsymbol{\lambda}$-player)[5]. Assuming that an MCP oracle is available, CombGame leverages Frank-Wolfe algorithms, namely OFW [HK12] and LLOO [GH16], for the $\boldsymbol{\omega}$-player and the MCP algorithm for the $\boldsymbol{\lambda}$-player. [JMKK21] leaves the existence of such an oracle running in polynomial time as an open problem. Our MCP algorithm resolves this issue. CombGame is statistically optimal in the high confidence regime but has no clear guarantees in the moderate regime [BGK22].

We wish to finally mention an algorithm that has inspired the design of P-FWS. This algorithm is referred to as Frank-Wolfe Sampling (FWS) [WTP21]. FWS is optimal in high confidence regime

---

[4] In Section 4.5 in [CGL+17], the authors provide a lemma stating that: if *parallel simulation* is additionally allowed, then any high-probability sample complexity upper bound can be converted to an upper bound in expectation.

[5] Note that this two-player game is different than the two-player game involved in our algorithm MCP.

but is not computationally efficient for combinatorial semi-bandits. For example, to deal with the non-smoothness issue of the objective function $F_{\boldsymbol{\mu}}$, FWS needs to construct the so-called $r$-subdifferentiable spaces and to optimize a linear function on these spaces. Unfortunately, these spaces can be generated by a number of vectors exponentially increasing with $K$ in combinatorial semi-bandits. Moreover, in moderate confidence regime, the sample complexity upper bound derived in [WTP21] has an exponential dependence in $K$.

All the relevant algorithms, their sample complexity guarantees and computational complexity are summarized in Table 1.

Table 1: Algorithms for best-arm identification in combinatorial semi-bandits with fixed confidence and their performance.

| Algorithm | Task | Instance-specific Sample Complexity | | Computational Complexity | |
|---|---|---|---|---|---|
| | | Non-asympt. | Asympt. Opt. | Needed (Provided) | Total LM oracle calls |
| Peace | C-MB | poly $\left(K, \triangle_{\min}^{-1}, \ln \delta^{-1}\right)$ w.h.p. | ✗ | LP solver (✔) | poly $\left(K, \triangle_{\min}^{-1}, \delta^{-1}\right)$ w.h.p. |
| GCB-PE | C-BAI | poly $\left(K, \triangle_{\min}^{-1}, \ln \delta^{-1}\right)$ w.h.p. | ✗ | - | poly $\left(K, \triangle_{\min}^{-1}, \ln \delta^{-1}\right)$ w.h.p. |
| CombGame | Trans. | ✗ (incomparable) | ✔ | MCP (✗) | ✗ |
| P-FWS | Trans. | poly $\left(K, \triangle_{\min}^{-1}, \ln \delta^{-1}\right)$ | ✔ | MCP (✔) | poly $\left(K, \triangle_{\min}^{-1}, \ln \delta^{-1}\right)$ |

# C Results related to our $(\epsilon, \theta)$-MCP algorithm

## C.1 Properties of Lagrangian dual of $f_x$

**Proposition 1.** *Let $(\omega, \mu) \in \Sigma_+ \times \Lambda$ and $x \in \mathcal{X} \setminus \{i^\star(\mu)\}$.*
*(a) The Lagrange dual function is linear in $x$. More precisely, $g_{\omega,\mu}(x, \alpha) = c_{\omega,\mu}(\alpha) + \langle \ell_{\omega,\mu}(\alpha), x \rangle$*
*where $c_{\omega,\mu}(\alpha) = \alpha \langle \mu - \frac{\alpha}{2}\omega^{-1}, i^\star(\mu) \rangle$ and $\ell_{\omega,\mu}(\alpha) = -\alpha \left(\mu + \frac{\alpha}{2}\omega^{-1} \odot (\mathbf{1}_K - 2i^\star(\mu))\right)$.*
*(b) $g_{\omega,\mu}(x, \cdot)$ is strictly concave (for any fixed $x$).*
*(c) $f_x(\omega, \mu) = \max_{\alpha \geq 0} g_{\omega,\mu}(x, \alpha)$ is attained by $\alpha_x^\star = \frac{\triangle_x(\mu)}{\langle x \oplus i^\star(\mu), \omega^{-1} \rangle}$.*
*(d) $\|\ell_{\omega,\mu}(\alpha_x^\star)\|_1 \leq L_{\omega,\mu} = 4D^2 K \|\mu\|_\infty^2 \|\omega^{-1}\|_\infty$.*

**Proof** Fix any $(\omega, \mu) \in \Sigma_+ \times \Lambda$ and let $i^\star = i^\star(\mu)$ for short. For convenience, the definition of $\mathcal{L}_{\omega,\mu}$ and $g_{\omega,\mu}$ are restated:

$$\mathcal{L}_{\omega,\mu}(\lambda, x, \alpha) = \left\langle \omega, \frac{(\mu - \lambda)^2}{2} \right\rangle + \alpha \langle i^\star - x, \lambda \rangle \quad \text{and} \quad g_{\omega,\mu}(x, \alpha) = \inf_{\lambda \in \mathbb{R}^K} \mathcal{L}_{\omega,\mu}(\lambda, x, \alpha).$$

Proof of (a): linearity of $g_{\omega,\mu}(\cdot, \alpha)$: Let $\lambda_{\omega,\mu}^\star(x, \alpha) \in \arg\inf_{\lambda \in \mathbb{R}^K} \mathcal{L}_{\omega,\mu}(\lambda, x, \alpha)$. The first-order condition implies that $\mathbf{0}_K = \nabla_\lambda \mathcal{L}_{\omega,\mu}(\lambda_{\omega,\mu}^\star(x, \alpha), x, \alpha) = \omega \odot (\lambda_{\omega,\mu}^\star(x, \alpha) - \mu) + \alpha(i^\star - x)$, which directly yields (as $\omega > \mathbf{0}_K$)

$$\lambda_{\omega,\mu}^\star(x, \alpha) = \mu + \alpha \omega^{-1} \odot (x - i^\star). \tag{11}$$

We plug (11) into $\mathcal{L}_{\omega,\mu}(\lambda_{\omega,\mu}^\star(x, \alpha), x, \alpha)$ and directly obtain that

$$g_{\omega,\mu}(x, \alpha) = \left\langle \omega, \frac{\alpha^2}{2}\omega^{-2} \odot (x - i^\star)^2 \right\rangle + \alpha \langle \mu, i^\star - x \rangle - \alpha^2 \langle \omega^{-1}, (x - i^\star)^2 \rangle$$

$$= \alpha \langle \mu, i^\star - x \rangle - \frac{\alpha^2}{2} \langle \omega^{-1}, (x - i^\star)^2 \rangle \tag{12}$$

$$= c_{\omega,\mu}(\alpha) + \langle \ell_{\omega,\mu}(\alpha), x \rangle, \tag{13}$$

where (13) follows from a fact that $(x - i^\star)^2 = i^\star - 2x \odot i^\star + x = i^\star + x \odot (\mathbf{1}_K - 2i^\star)$.

Proof of (b): strict concavity of $g_{\omega,\mu}(x, \cdot)$: This is trivial from (12).

Proof of (c): $f_x(\omega, \mu) = \max_{\alpha \geq 0} g_{\omega,\mu}(x, \alpha)$ is attained by $\alpha_x^\star = \frac{\triangle_x(\mu)}{\langle x \oplus i^\star, \omega^{-1} \rangle}$: For a fixed $x \neq i^\star$, by the first-order condition of (12), we find that the maximum of $g_{\omega,\mu}(x, \cdot)$ is reached at

$$\alpha_x^\star = \frac{\triangle_x(\mu)}{\langle \omega^{-1}, (x - i^\star)^2 \rangle} = \frac{\triangle_x(\mu)}{\langle x \oplus i^\star, \omega^{-1} \rangle}, \tag{14}$$

where for the second equality, we use the assumption that $i^\star$ and $x$ are binary vectors and hence $(x - i^\star)^2 = x \oplus i^\star$. We now verify that $(\alpha_x^\star, \lambda_x^\star)$, where $\lambda_x^\star = \lambda_{\omega,\mu}^\star(x, \alpha_x^\star)$ (see (11)), satisfies KKT conditions, which is equivalent to strong duality (refer to [Vis21, BV04]) under Slater's condition (there exists a $\lambda \in \mathbb{R}^K$ such that the constraint is strict). Since $x$ is a suboptimal action, $\triangle_x(\mu)$ is positive, so is $\alpha_x^\star$ (dual feasibility). To verify $\langle \lambda_x^\star, i^\star - x \rangle \leq 0$ (primal feasibility), the definition of $\lambda_{\omega,\mu}^\star(\cdot, \cdot)$, (11), yields

$$\langle \lambda_x^\star, i^\star - x \rangle = \triangle_x(\mu) + \alpha_x^\star \langle \omega^{-1} \odot (x - i^\star), (i^\star - x) \rangle$$
$$= \triangle_x(\mu) - \alpha_x^\star \langle \omega^{-1}, x \oplus i^\star \rangle = 0,$$

which implies that $\alpha_x^\star \langle i^\star - x, \lambda_x^\star \rangle = 0$ (complementary slackness). Finally, stationarity holds automatically as $\nabla_\lambda \mathcal{L}_{\omega,\mu}(\lambda_x^\star, x, \alpha) = 0$ for all $\alpha$.

Proof of (d): $\|\ell_{\omega,\mu}(\alpha_x^\star)\|_1 \leq L_{\omega,\mu} = 4D^2 K \|\mu\|_\infty^2 \|\omega^{-1}\|_\infty$: Following from the expression of $\ell_{\omega,\mu}(\alpha)$, we have $\ell_{\omega,\mu}(\alpha_x^\star) = -\alpha_x^\star \mu + \frac{\alpha_x^{\star 2}}{2}\omega^{-1} \odot (\mathbf{1}_K - 2i^\star)$. Observe that $\|\mu\|_1 \leq K \|\mu\|_\infty \leq$

$K \left\| \boldsymbol{\omega}^{-1} \right\|_\infty \left\| \boldsymbol{\mu} \right\|_\infty$ (as $\boldsymbol{\omega} \in \Sigma_+$) and the coordinate of $\mathbf{1}_K - 2\boldsymbol{i}^\star$ is either 1 or $-1$, a simple application of triangle inequality leads to

$$\left\| \boldsymbol{\ell}_{\boldsymbol{\omega},\boldsymbol{\mu}}(\alpha_{\boldsymbol{x}}^\star) \right\|_1 \leq K \left\| \boldsymbol{\omega}^{-1} \right\|_\infty \left( \left\| \boldsymbol{\mu} \right\|_\infty + \frac{\alpha_{\boldsymbol{x}}^\star}{2} \right) \alpha_{\boldsymbol{x}}^\star.$$

As for $\alpha_{\boldsymbol{x}}^\star$ (see (14)), $\triangle_{\boldsymbol{x}}(\boldsymbol{\mu}) \leq 2D \left\| \boldsymbol{\mu} \right\|_\infty$ and $\left\langle \boldsymbol{\omega}^{-1}, \boldsymbol{x} \oplus \boldsymbol{i}^\star \right\rangle \geq \min_k \omega_k^{-1} \geq 1$, hence we conclude that $\left\| \boldsymbol{\ell}_{\boldsymbol{\omega},\boldsymbol{\mu}}(\alpha_{\boldsymbol{x}}^\star) \right\|_1 \leq 2D(D+1)K \left\| \boldsymbol{\mu} \right\|_\infty^2 \left\| \boldsymbol{\omega}^{-1} \right\|_\infty \leq L_{\boldsymbol{\omega},\boldsymbol{\mu}}$. $\qquad\square$

## C.2 Analysis of MCP

**Theorem 3.** *Let $\epsilon, \theta \in (0,1)$. Under Assumption 1, for any $(\boldsymbol{\omega}, \boldsymbol{\mu}) \in \Sigma_+ \times \Lambda$, the $(\epsilon, \theta)$-MCP$(\boldsymbol{\omega}, \boldsymbol{\mu})$ algorithm outputs $(\hat{F}, \hat{\boldsymbol{x}})$ satisfying*

$$\mathbb{P}\left[ F_{\boldsymbol{\mu}}(\boldsymbol{\omega}) \leq \hat{F} \leq (1+\epsilon)F_{\boldsymbol{\mu}}(\boldsymbol{\omega}) \right] \geq 1 - \theta \quad and \quad \hat{F} = \max_{\alpha \geq 0} g_{\boldsymbol{\omega},\boldsymbol{\mu}}(\hat{\boldsymbol{x}}, \alpha).$$

*Moreover, the number of LM Oracle calls the algorithm does is almost surely at most*

$$\left\lceil \frac{c_\theta^2(1+\epsilon)^2}{\epsilon^2 F_{\boldsymbol{\mu}}(\boldsymbol{\omega})^2} \right\rceil = \mathcal{O}\left( \frac{\left\| \boldsymbol{\mu} \right\|_\infty^4 \left\| \boldsymbol{\omega}^{-1} \right\|_\infty^2 K^3 D^5 \ln K \ln \theta^{-1}}{\epsilon^2 F_{\boldsymbol{\mu}}(\boldsymbol{\omega})^2} \right).$$

**Proof** Fix any $(\boldsymbol{\omega}, \boldsymbol{\mu}) \in \Sigma_+ \times \Lambda$ and denote by $\boldsymbol{i}^\star = \boldsymbol{i}^\star(\boldsymbol{\mu})$. Suppose Algorithm 1 reaches the stopping criterion at the $N$-th iteration.

Guarantees on the outputs of MCP: By Proposition 1 (a),

$$\sum_{n=1}^N g_{\boldsymbol{\omega},\boldsymbol{\mu}}(\boldsymbol{x}^{(n)}, \alpha^{(n)}) - \min_{\boldsymbol{x} \neq \boldsymbol{i}^\star} \sum_{n=1}^N g_{\boldsymbol{\omega},\boldsymbol{\mu}}(\boldsymbol{x}, \alpha^{(n)}) = \sum_{n=1}^N \left\langle \boldsymbol{\ell}_{\boldsymbol{\omega},\boldsymbol{\mu}}(\alpha^{(n)}), \boldsymbol{x}^{(n)} \right\rangle - \min_{\boldsymbol{x} \neq \boldsymbol{i}^\star} \sum_{n=1}^N \left\langle \boldsymbol{\ell}_{\boldsymbol{\omega},\boldsymbol{\mu}}(\alpha^{(n)}), \boldsymbol{x} \right\rangle.$$

The regret of $\boldsymbol{x}$-player can be bounded by applying Lemma 3, resulting in:

$$\mathbb{P}\left[ \sum_{n=1}^N g_{\boldsymbol{\omega},\boldsymbol{\mu}}(\boldsymbol{x}^{(n)}, \alpha^{(n)}) - \min_{\boldsymbol{x} \neq \boldsymbol{i}^\star} \sum_{n=1}^N g_{\boldsymbol{\omega},\boldsymbol{\mu}}(\boldsymbol{x}, \alpha^{(n)}) \leq c_\theta \sqrt{N} \right] \geq 1 - \theta. \tag{15}$$

To relate $F_{\boldsymbol{\mu}}(\boldsymbol{\omega})$ with (15), let $\boldsymbol{x}_e$ be the minimizer attaining $F_{\boldsymbol{\mu}}(\boldsymbol{\omega}) = f_{\boldsymbol{x}_e}(\boldsymbol{\omega}, \boldsymbol{\mu})$. Then,

$$\min_{\boldsymbol{x} \neq \boldsymbol{i}^\star} \sum_{n=1}^N g_{\boldsymbol{\omega},\boldsymbol{\mu}}(\boldsymbol{x}, \alpha^{(n)}) \leq \sum_{n=1}^N g_{\boldsymbol{\omega},\boldsymbol{\mu}}(\boldsymbol{x}_e, \alpha^{(n)}) \leq N \max_{\alpha \geq 0} g_{\boldsymbol{\omega},\boldsymbol{\mu}}(\boldsymbol{x}_e, \alpha) = N F_{\boldsymbol{\mu}}(\boldsymbol{\omega}). \tag{16}$$

Recall that $\alpha^{(n)}$ is chosen as the best response $\max_{\alpha \geq 0} g_{\boldsymbol{\omega},\boldsymbol{\mu}}(\boldsymbol{x}^{(n)}, \alpha) = g_{\boldsymbol{\omega},\boldsymbol{\mu}}(\boldsymbol{x}^{(n)}, \alpha^{(n)})$ and that $\hat{F} = \min_{n \in [N]} g_{\boldsymbol{\omega},\boldsymbol{\mu}}(\boldsymbol{x}^{(n)}, \alpha^{(n)})$. These together with (16) imply that

$$N(\hat{F} - F_{\boldsymbol{\mu}}(\boldsymbol{\omega})) \leq \sum_{n=1}^N g_{\boldsymbol{\omega},\boldsymbol{\mu}}(\boldsymbol{x}^{(n)}, \alpha^{(n)}) - \min_{\boldsymbol{x} \neq \boldsymbol{i}^\star} \sum_{n=1}^N g_{\boldsymbol{\omega},\boldsymbol{\mu}}(\boldsymbol{x}, \alpha^{(n)}). \tag{17}$$

A simple rearrangement on (15) and (17) implies that: with probability at least $1 - \theta$,

$$\hat{F} - F_{\boldsymbol{\mu}}(\boldsymbol{\omega}) \leq \frac{c_\theta}{\sqrt{N}} \leq \epsilon(\hat{F} - \frac{c_\theta}{\sqrt{N}}) \leq \epsilon F_{\boldsymbol{\mu}}(\boldsymbol{\omega}),$$

where the second inequality follows from the stopping criterion that $\sqrt{N} > c_\theta(1+\epsilon)/\epsilon\hat{F}$, and the last inequality simply comes from the rearrangement of the first inequality.

Computational cost: From the stopping criterion of MCP, we know that

$$N = \left\lceil \frac{c_\theta^2(1+\epsilon)^2}{\epsilon^2 \hat{F}^2} \right\rceil \leq \left\lceil \frac{c_\theta^2(1+\epsilon^{-1})^2}{F_{\boldsymbol{\mu}}(\boldsymbol{\omega})^2} \right\rceil = \mathcal{O}\left( \frac{L_{\boldsymbol{\omega},\boldsymbol{\mu}}^2 \left( \sqrt{K \ln K} + \sqrt{\ln \theta^{-1}} \right)^2}{\epsilon^2 F_{\boldsymbol{\mu}}(\boldsymbol{\omega})^2} \right)$$

since $\hat{F} \geq F_{\boldsymbol{\mu}}(\boldsymbol{\omega})$ and $c_\theta = L_{\boldsymbol{\omega},\boldsymbol{\mu}}\left(4\sqrt{K(\ln K + 1)} + \sqrt{\ln(\theta^{-1})/2}\right)$. Finally, as computing each $\boldsymbol{x}^{(n)}$ takes at most $D$ calls to LM Oracle, the total number of LM Oracle calls is

$$\mathcal{O}\left(\frac{L_{\boldsymbol{\omega},\boldsymbol{\mu}}^2 D\left(\sqrt{K \ln K} + \sqrt{\ln \theta^{-1}}\right)^2}{\epsilon^2 F_{\boldsymbol{\mu}}(\boldsymbol{\omega})^2}\right) = \mathcal{O}\left(\frac{\|\boldsymbol{\mu}\|_\infty^4 \|\boldsymbol{\omega}^{-1}\|_\infty^2 K^3 D^5 \ln K \ln \theta^{-1}}{\epsilon^2 F_{\boldsymbol{\mu}}(\boldsymbol{\omega})^2}\right)$$

by recalling $L_{\boldsymbol{\omega},\boldsymbol{\mu}} = 4D^2 K \|\boldsymbol{\mu}\|_\infty^2 \|\boldsymbol{\omega}^{-1}\|_\infty$ from Proposition 1 (d) and $(\sqrt{\ln K} + \sqrt{\ln \theta^{-1}})^2 = \mathcal{O}(\ln K \ln \theta^{-1})$. $\qquad \square$

### C.3 Regret analysis of Follow-the-Perturbed-Leader

In this subsection, we aim at proving Lemma 3, which is a direct consequence of Lemma 4. One can find similar proofs in e.g. [KV05, Neu15, SN20]. However, the parameter $\eta_n$ in our MCP algorithm is varying and carefully chosen (without the knowledge of the last round), which makes the proof slightly more complicated.

**Lemma 3.** *Let $N \in \mathbb{N}$. Under $(\epsilon, \theta)$-MCP$(\boldsymbol{\omega}, \boldsymbol{\mu})$, then*

$$\mathbb{P}\left[\frac{1}{N}\sum_{n=1}^N g_{\boldsymbol{\omega},\boldsymbol{\mu}}(\boldsymbol{x}^{(n)}, \alpha^{(n)}) - \frac{1}{N}\min_{\boldsymbol{x} \neq i^\star}\sum_{n=1}^N g_{\boldsymbol{\omega},\boldsymbol{\mu}}(\boldsymbol{x}, \alpha^{(n)}) \leq \frac{c_\theta}{\sqrt{N}}\right] \geq 1 - \theta.$$

**Lemma 4.** *Let $\theta \in (0,1)$ and $\mathcal{M} \subseteq \{0,1\}^K$. Given an arbitrary sequence $\{\boldsymbol{\ell}_n\}_{n\geq 1}$ of vectors in $\mathbb{R}^K$ whose length $\|\boldsymbol{\ell}_n\|_1$ is bounded by $L > 0$ for all $n \in \mathbb{N}$. Suppose $\{\boldsymbol{x}^{(n)}\}_{n \geq 1}$ is generated by*

$$\boldsymbol{x}^{(n)} \in \operatorname*{argmin}_{\boldsymbol{x} \in \mathcal{M}}\left(\sum_{m=1}^{n-1}\langle \boldsymbol{\ell}_m, \boldsymbol{x}\rangle + \left\langle \frac{\boldsymbol{\mathcal{Z}}_n}{\eta_n}, \boldsymbol{x}\right\rangle\right),$$

*where $\boldsymbol{\mathcal{Z}}_n = (\mathcal{Z}_{1,n}, \cdots, \mathcal{Z}_{K,n})$ is a random vector with uncorrelated exponentially distributed (with unit mean) components, and $\eta_n = \sqrt{K(\ln K + 1)/(4nL^2)}$. Then, for any $N \in \mathbb{N}$,*

$$\mathbb{P}\left[\sum_{n=1}^N \left\langle \boldsymbol{\ell}_n, \boldsymbol{x}^{(n)}\right\rangle - \min_{\boldsymbol{x} \in \mathcal{M}}\sum_{n=1}^N \langle \boldsymbol{\ell}_n, \boldsymbol{x}\rangle \leq L\sqrt{N}\left(4\sqrt{K(\ln K + 1)} + \sqrt{\frac{\ln \theta^{-1}}{2}}\right)\right] \geq 1 - \theta.$$

**Proof of Lemma 4:** We will prove this lemma as if $\{\boldsymbol{\ell}_n\}_n$ is chosen in advance since there exists a standard technique for extending regret against oblivious player to the one against nonoblivious one (see Lemma 4.1 in [CBL06]). For convenience, we introduce the following notation. Let $\boldsymbol{m}^\star(\cdot) = \operatorname{argmin}_{\boldsymbol{x} \in \mathcal{M}}\langle \cdot, \boldsymbol{x}\rangle$. Finally, further define global minimizer $\boldsymbol{x}_\star = \boldsymbol{m}^\star\left(\sum_{n=1}^N \boldsymbol{\ell}_n\right)$ and an auxiliary vector $\boldsymbol{b}^{(n)} = \boldsymbol{m}^\star\left(\sum_{m=1}^n \boldsymbol{\ell}_m + \boldsymbol{\mathcal{Z}}_1/\eta_n\right)$.

It suffices to show the expected regret bound (18).

$$\mathbb{E}\left[\sum_{n=1}^N \left\langle \boldsymbol{\ell}_n, \boldsymbol{x}^{(n)}\right\rangle\right] - \min_{\boldsymbol{x} \in \mathcal{M}}\sum_{n=1}^N \langle \boldsymbol{\ell}_n, \boldsymbol{x}\rangle \leq 4L\sqrt{NK(\ln K + 1)}. \tag{18}$$

This is because $\{\langle \boldsymbol{\ell}_n, \boldsymbol{x}^{(n)}\rangle - \mathbb{E}[\langle \boldsymbol{\ell}_n, \boldsymbol{x}^{(n)}\rangle]\}_n$ forms a sequence of bounded martingale difference, so an application of a concentration inequality (Lemma 6) with $V_n = \langle \boldsymbol{\ell}_n, \boldsymbol{x}^{(n)}\rangle - \mathbb{E}[\langle \boldsymbol{\ell}_n, \boldsymbol{x}^{(n)}\rangle]$, $r_n = L$ for $n \in [N]$, and $s = L\sqrt{N \ln \theta^{-1}/2}$ gives that

$$\mathbb{P}\left[\sum_{n=1}^N \left\langle \boldsymbol{\ell}_n, \boldsymbol{x}^{(n)}\right\rangle - \mathbb{E}\left[\sum_{n=1}^N \left\langle \boldsymbol{\ell}_n, \boldsymbol{x}^{(n)}\right\rangle\right] > L\sqrt{\frac{N \ln \theta^{-1}}{2}}\right] \leq \theta$$

and combining this with (18) completes the proof.

Proof of (18): We decompose the regret into two terms:

$$\sum_{n=1}^{N} \left\langle \boldsymbol{\ell}_n, \boldsymbol{x}^{(n)} - \boldsymbol{x}_\star \right\rangle = \sum_{n=1}^{N} \left\langle \boldsymbol{b}^{(n)} - \boldsymbol{x}_\star, \boldsymbol{\ell}_n \right\rangle + \sum_{n=1}^{N} \left\langle \boldsymbol{x}^{(n)} - \boldsymbol{b}^{(n)}, \boldsymbol{\ell}_n \right\rangle.$$

**(i). We show that** $\mathbb{E}\left[ \sum_{n=1}^{N} \left\langle \boldsymbol{b}^{(n)} - \boldsymbol{x}_\star, \boldsymbol{\ell}_n \right\rangle \right] \leq \frac{K(\ln K + 1)}{\eta_N}$. Invoking Lemma 5 with $\boldsymbol{x} = \boldsymbol{x}_\star$ results in

$$\mathbb{E}\left[ \sum_{n=1}^{N} \left\langle \boldsymbol{b}^{(n)} - \boldsymbol{x}_\star, \boldsymbol{\ell}_n \right\rangle \right] \leq \mathbb{E}\left[ \left\langle \frac{\boldsymbol{x}_\star}{\eta_N} - \left( \frac{\boldsymbol{b}^{(1)}}{\eta_1} + \sum_{n=2}^{N} \left( \frac{1}{\eta_n} - \frac{1}{\eta_{n-1}} \right) \boldsymbol{b}^{(n)} \right), \boldsymbol{\mathcal{Z}}_1 \right\rangle \right]$$

$$\leq \mathbb{E}\left[ \| \boldsymbol{\mathcal{Z}}_1 \|_\infty \left\| \frac{\boldsymbol{x}_\star}{\eta_N} - \left( \frac{\boldsymbol{b}^{(1)}}{\eta_1} + \sum_{n=2}^{N} \left( \frac{1}{\eta_n} - \frac{1}{\eta_{n-1}} \right) \boldsymbol{b}^{(n)} \right) \right\|_1 \right],$$

where the last inequality uses Hölder's inequality. As all the components of $\boldsymbol{x}_\star$ and $\frac{\eta_N \boldsymbol{b}^{(1)}}{\eta_1} + \sum_{n=2}^{N} \eta_N \left( \frac{1}{\eta_n} - \frac{1}{\eta_{n-1}} \right) \boldsymbol{b}^{(n)}$ are nonnegative and bounded by 1, the 1-norm of their difference is bounded by $K$. It remains to show

$$\mathbb{E}[\| \boldsymbol{\mathcal{Z}}_1 \|_\infty] = \int_0^\infty \mathbb{P}\left[ \max_i \mathcal{Z}_{1,i} \geq x \right] dx \leq \int_0^{\ln K} \mathbb{P}\left[ \max_i \mathcal{Z}_{1,i} \geq x \right] dx + \int_{\ln K}^\infty K e^{-x} dx \leq \ln K + 1.$$

**(ii). We show that** $\mathbb{E}\left[ \sum_{n=1}^{N} \left\langle \boldsymbol{x}^{(n)} - \boldsymbol{b}^{(n)}, \boldsymbol{\ell}_n \right\rangle \right] \leq 2L^2 \sum_{n=1}^{N} \eta_n$. Let the pdf of $\exp(1)$ be $\pi(\cdot) = e^{-\|\cdot\|_1}$.

$$\mathbb{E}\left[ \left\langle \boldsymbol{b}^{(n)}, \boldsymbol{\ell}_n \right\rangle \right] = \int_{\boldsymbol{z} \in \mathbb{R}^K} \left\langle \boldsymbol{m}^\star \left( \eta_n \sum_{m=1}^{n} \boldsymbol{\ell}_m + \boldsymbol{z} \right), \boldsymbol{\ell}_n \right\rangle d\pi(\boldsymbol{z})$$

$$= \int_{\boldsymbol{y} \in \mathbb{R}^K} \left\langle \boldsymbol{m}^\star \left( \eta_n \sum_{m=1}^{n-1} \boldsymbol{\ell}_m + \boldsymbol{y} \right), \boldsymbol{\ell}_n \right\rangle d\pi(\boldsymbol{y} - \eta_n \boldsymbol{\ell}_n)$$

$$= \int_{\boldsymbol{y} \in \mathbb{R}^K} \left\langle \boldsymbol{m}^\star \left( \eta_n \sum_{m=1}^{n-1} \boldsymbol{\ell}_m + \boldsymbol{y} \right), \boldsymbol{\ell}_n \right\rangle e^{-\|\boldsymbol{y} - \eta_n \boldsymbol{\ell}_n\|_1 + \|\boldsymbol{y}\|_1} d\pi(\boldsymbol{y}).$$

Notice that the triangular inequality implies $-\|\boldsymbol{y} - \eta_n \boldsymbol{\ell}_n\|_1 + \|\boldsymbol{y}\|_1 \leq \|\eta_n \boldsymbol{\ell}_n\|_1 \leq \eta_n L$ and $e^x \leq 1 + 2x$ for all $x \in (0,1)$ (Taylor expansion), so recalling $\boldsymbol{x}^{(n)} = \boldsymbol{m}^\star \left( \eta_n \sum_{m=1}^{n-1} \boldsymbol{\ell}_m + \boldsymbol{\mathcal{Z}}_1 \right)$, we deduce that

$$\sum_{n=1}^{N} \mathbb{E}\left[ \left\langle \boldsymbol{x}^{(n)} - \boldsymbol{b}^{(n)}, \boldsymbol{\ell}_n \right\rangle \right] \leq \sum_{n=1}^{N} 2\eta_n L \int_{\boldsymbol{z} \in \mathbb{R}^K} \left\langle \boldsymbol{m}^\star \left( \eta_n \sum_{m=1}^{n} \boldsymbol{\ell}_m + \boldsymbol{z} \right), \boldsymbol{\ell}_n \right\rangle d\pi(\boldsymbol{z})$$

$$\leq \sum_{n=1}^{N} 2\eta_n L \int_{\boldsymbol{z} \in \mathbb{R}^K} \left\| \boldsymbol{m}^\star \left( \eta_n \sum_{m=1}^{n} \boldsymbol{\ell}_m + \boldsymbol{z} \right) \right\|_\infty \|\boldsymbol{\ell}_n\|_1 \, d\pi(\boldsymbol{z}) \leq 2L^2 \sum_{n=1}^{N} \eta_n.$$

Finally, plugging $\eta_n = \sqrt{\frac{K(\ln K + 1)}{4nL^2}}$ into (i). and (ii). directly concludes the proof. $\qquad \square$

The following lemma is a result that can be found in [CBL06, H+16], we rewrite it here for completeness.

**Lemma 5.** *According to* $\boldsymbol{b}^{(n)} = \boldsymbol{m}^\star \left( \eta_n \sum_{m=1}^{n} \boldsymbol{\ell}_m + \boldsymbol{\mathcal{Z}}_1 \right)$, *we can have*

$$\forall \boldsymbol{x} \in \mathcal{M}, \quad \sum_{n=1}^{N} \left\langle \boldsymbol{b}^{(n)} - \boldsymbol{x}, \boldsymbol{\ell}_n \right\rangle \leq \left\langle \frac{\boldsymbol{x}}{\eta_N}, \boldsymbol{\mathcal{Z}}_1 \right\rangle - \left\langle \frac{\boldsymbol{b}^{(1)}}{\eta_1} + \sum_{n=2}^{N} \left( \frac{1}{\eta_n} - \frac{1}{\eta_{n-1}} \right) \boldsymbol{b}^{(n)}, \boldsymbol{\mathcal{Z}}_1 \right\rangle. \quad (19)$$

**Proof** This is done by induction. For the base case, $N = 1$, as $\boldsymbol{b}^{(1)} = \boldsymbol{m}^\star \left( \boldsymbol{\ell}_1 + \boldsymbol{\mathcal{Z}}_1/\eta_1 \right)$

$$\left\langle \boldsymbol{b}^{(1)}, \boldsymbol{\ell}_1 + \boldsymbol{\mathcal{Z}}_1/\eta_1 \right\rangle \le \left\langle \boldsymbol{x}, \boldsymbol{\ell}_1 + \boldsymbol{\mathcal{Z}}_1/\eta_1 \right\rangle$$

for any $\boldsymbol{x} \in \mathcal{M}$. A simple rearrangement yields (19). While considering $N + 1$, we suppose (19) holds for all integers smaller than $N + 1$. For an arbitrary $\boldsymbol{x} \in \mathcal{M}$, the fact $\boldsymbol{b}^{(N+1)} = \boldsymbol{m}^\star \left( \sum_{n=1}^{N+1} \boldsymbol{\ell}_n + \boldsymbol{\mathcal{Z}}_1/\eta_{N+1} \right)$ directly implies that

$$
\begin{aligned}
\left\langle \boldsymbol{x}, \sum_{n=1}^{N+1} \boldsymbol{\ell}_n + \frac{\boldsymbol{\mathcal{Z}}_1}{\eta_{N+1}} \right\rangle &\ge \left\langle \boldsymbol{b}^{(N+1)}, \sum_{n=1}^{N+1} \boldsymbol{\ell}_n + \frac{\boldsymbol{\mathcal{Z}}_1}{\eta_{N+1}} \right\rangle \\
&= \left\langle \boldsymbol{b}^{(N+1)}, \boldsymbol{\ell}_{N+1} + \left( \frac{1}{\eta_{N+1}} - \frac{1}{\eta_N} \right) \boldsymbol{\mathcal{Z}}_1 \right\rangle + \left\langle \boldsymbol{b}^{(N+1)}, \sum_{n=1}^{N} \boldsymbol{\ell}_n + \frac{\boldsymbol{\mathcal{Z}}_1}{\eta_N} \right\rangle \\
&\ge \sum_{n=1}^{N+1} \left\langle \boldsymbol{b}^{(n)}, \boldsymbol{\ell}_n \right\rangle + \left\langle \frac{\boldsymbol{b}^{(1)}}{\eta_1} + \sum_{n=2}^{N+1} \left( \frac{1}{\eta_n} - \frac{1}{\eta_{n-1}} \right) \boldsymbol{b}^{(n)}, \boldsymbol{\mathcal{Z}}_1 \right\rangle,
\end{aligned}
$$

where the last inequality comes from applying the hypothesis (19) with $\boldsymbol{x} = \boldsymbol{b}^{(N+1)}$ on the second inner product. Rearrange the above inequality, our induction is completed. $\square$

**Lemma 6** (Hoeffding-Azuma)**.** *Let* $N \in \mathbb{N}$, $V_1, V_2, \cdots, V_N$ *be a bounded martingale difference sequence w.r.t.* $X_1, X_2, \cdots, X_N$ *such that for any* $n \in [N]$ $V_n \in [A_n, A_n + r_n]$ *for some random variable* $A_n$, *measurable w.r.t.* $X_1, \cdots, X_{n-1}$ *and a positive constant* $r_n$. *Then, for any* $s > 0$,

$$\mathbb{P}\left[ \sum_{n\in[N]} V_n > s \right] \le \exp\left( -\frac{2s^2}{\sum_{n\in[N]} r_n^2} \right) \quad and \quad \mathbb{P}\left[ \sum_{n\in[N]} V_n < -s \right] \le \exp\left( -\frac{2s^2}{\sum_{n\in[N]} c_n^2} \right).$$

# D  Analysis of `P-FWS`

In this appendix, we prove our main theorem.

**Theorem 4.** *Let $\boldsymbol{\mu} \in \Lambda$ and $\delta \in (0,1)$. If* `P-FWS` *is parametrized using*

$$(\epsilon_t, \eta_t, n_t, \rho_t, \theta_t) = \left( t^{-\frac{1}{9}}, \frac{1}{4\sqrt{t|\mathcal{X}_0|}}, \left\lceil t^{\frac{1}{4}} \right\rceil, \frac{1}{16tD^2|\mathcal{X}_0|}, \frac{1}{t^{\frac{1}{4}}e^{\sqrt{t}}} \right), \tag{10}$$

*then (i) the algorithm finishes in finite time almost surely and $\mathbb{P}_{\boldsymbol{\mu}}[\hat{\imath} \neq \boldsymbol{i}^\star(\boldsymbol{\mu})] \leq \delta$; (ii) its sample complexity satisfies $\mathbb{P}_{\boldsymbol{\mu}}\left[ \limsup_{\delta \to 0} \frac{\tau}{\ln \delta^{-1}} \leq T^\star(\boldsymbol{\mu}) \right] = 1$ and for any $\epsilon, \tilde{\epsilon} \in (0,1)$ with $\epsilon < \min\{1, \frac{2D^2\triangle_{\min}^2}{K}, \frac{D^2\|\boldsymbol{\mu}\|_\infty^2}{3}\}$,*

$$\mathbb{E}_{\boldsymbol{\mu}}[\tau] \leq \frac{(1+\tilde{\epsilon})^2}{T^\star(\boldsymbol{\mu})^{-1} - 6\epsilon} \times H\left( \frac{1}{\delta} \cdot \frac{4c_2}{3} \cdot \frac{(1+\tilde{\epsilon})^2}{T^\star(\boldsymbol{\mu})^{-1} - 6\epsilon} \right) + \Psi(\epsilon, \tilde{\epsilon}),$$

*where $H(x) = \ln x + \ln\ln x + 1$ and $\Psi(\epsilon, \tilde{\epsilon})$ (refer to (34) for a detailed expression) is polynomial in $\epsilon^{-1}, \tilde{\epsilon}^{-1}, K, \|\boldsymbol{\mu}\|_\infty$ and $\triangle_{\min}(\boldsymbol{\mu})^{-1}$; (iii) the expected number of* `LM` *Oracle calls is upper bounded by a polynomial in $\ln \delta^{-1}, K, \|\boldsymbol{\mu}\|_\infty$ and $\triangle_{\min}(\boldsymbol{\mu})^{-1}$.*

## D.1  $\delta$-correctness (Theorem 4 (i))

Recall that `P-FWS` stopping rule is:

$$\tau = \inf\left\{ t > 4|\mathcal{X}_0| : \frac{t\hat{F}_t}{1+\epsilon_t} > \beta\left( t, \left(1 - \frac{1}{4|\mathcal{X}_0|}\right)\delta \right), \max\left\{ \frac{1}{\triangle_{\min}(\hat{\boldsymbol{\mu}}(t))}, \|\hat{\boldsymbol{\mu}}(t)\|_\infty \right\} \leq \sqrt{t} \right\}, \tag{7}$$

where $\hat{F}_t$ is computed by $(\epsilon_t, \delta/t^2)$-`MCP`$(\hat{\boldsymbol{\omega}}(t), \hat{\boldsymbol{\mu}}(t))$. Let $\hat{\imath} = \boldsymbol{i}^\star(\hat{\boldsymbol{\mu}}(\tau))$ be the output of `P-FWS`. Define the good event $\mathcal{G} = \bigcap_{t=4|\mathcal{X}_0|+1}^\infty \{\hat{F}_t \leq (1+\epsilon_t)F_{\hat{\boldsymbol{\mu}}(t)}(\hat{\boldsymbol{\omega}}(t))\}$. Hence, it follows from the guarantee of $(\epsilon_t, \delta/t^2)$-`MCP` algorithm that

$$\mathbb{P}_{\boldsymbol{\mu}}[\mathcal{G}^c] \leq \delta \sum_{t=4|\mathcal{X}_0|+1}^\infty t^{-2} \leq \delta \int_{4|\mathcal{X}_0|}^\infty x^{-2}dx \leq \frac{\delta}{4|\mathcal{X}_0|}.$$

Besides, under the event $\mathcal{G}$,

$$(1+\epsilon_\tau)\tau F_{\hat{\boldsymbol{\mu}}(\tau)}(\hat{\boldsymbol{\omega}}(\tau)) \geq \tau \hat{F}_\tau \geq (1+\epsilon_\tau)\beta\left( \tau, \left(1 - \frac{1}{4|\mathcal{X}_0|}\right)\delta \right)$$

holds, implying that $\tau F_{\hat{\boldsymbol{\mu}}(\tau)}(\hat{\boldsymbol{\omega}}(\tau)) \geq \beta\left( \tau, \left(1 - \frac{1}{4|\mathcal{X}_0|}\right)\delta \right)$. So, by (8)-(9), $\hat{\imath} = \boldsymbol{i}^\star(\hat{\boldsymbol{\mu}}(t))$ satisfies:

$$\mathbb{P}_{\boldsymbol{\mu}}[\hat{\imath} \neq \boldsymbol{i}^\star(\boldsymbol{\mu}), \mathcal{G}] \leq \left(1 - \frac{1}{4|\mathcal{X}_0|}\right)\delta,$$

and thus $\mathbb{P}_{\boldsymbol{\mu}}[\hat{\imath} \neq \boldsymbol{i}^\star(\boldsymbol{\mu})] \leq \mathbb{P}_{\boldsymbol{\mu}}[\hat{\imath} \neq \boldsymbol{i}^\star(\boldsymbol{\mu}), \mathcal{G}] + \mathbb{P}_{\boldsymbol{\mu}}[\mathcal{G}^c] \leq \delta$.

## D.2  Almost-sure upper bound (Theorem 4 (ii))

In this section, we show Theorem 4 (ii) an almost-sure upper bound on the sample complexity for `P-FWS`. Our proof is based on the continuity of $F_{\boldsymbol{\mu}}$ in $\boldsymbol{\mu}$ (as in [GK16, WTP21]) and also on the following observations:

(a) $\{\hat{\boldsymbol{\mu}}(t) \overset{t \to \infty}{\longrightarrow} \boldsymbol{\mu}\}$ and $\{\nabla \tilde{F}_{\boldsymbol{\mu}, \eta_t, n_t}(\boldsymbol{\omega}) \overset{t \to \infty}{\longrightarrow} \nabla \bar{F}_{\boldsymbol{\mu}, \eta_t}(\boldsymbol{\omega}), \forall \boldsymbol{\omega} \in \Sigma_+\}$ happen almost surely,

(b) $\hat{F}_t \geq F_{\hat{\boldsymbol{\mu}}(t)}(\hat{\boldsymbol{\omega}}(t))$.

For (a), by the law of large numbers, $\hat{\boldsymbol{\mu}}(t) \overset{t \to \infty}{\longrightarrow} \boldsymbol{\mu}$ as $N_k(t) \overset{t \to \infty}{\longrightarrow} \infty$ for all $k \in [K]$ yielded by forced exploration rounds involved in `P-FWS` (Lemma 14 in Appendix F), $\nabla \tilde{F}_{\boldsymbol{\mu}, \eta_t, n_t}(\boldsymbol{\omega}) \overset{t \to \infty}{\longrightarrow}$

$\nabla \bar{F}_{\boldsymbol{\mu},\eta_t}(\boldsymbol{\omega})$, $\forall \boldsymbol{\omega} \in \Sigma_+$ is a direct consequence that $n_t \overset{t\to\infty}{\longrightarrow} \infty$. (b) is immediately derived from the definition of $\hat{F}_t$ as $\hat{F}_t = f_{\hat{\boldsymbol{x}}}(\hat{\boldsymbol{\omega}}(t), \hat{\boldsymbol{\mu}}(t))$ for some action $\hat{\boldsymbol{x}} \neq \boldsymbol{i}^\star(\hat{\boldsymbol{\mu}}(t))$ and $F_{\hat{\boldsymbol{\mu}}(t)}(\hat{\boldsymbol{\omega}}(t)) = \min_{\boldsymbol{x} \in \mathcal{X} \setminus \boldsymbol{i}^\star(\hat{\boldsymbol{\mu}}(t))} f_{\boldsymbol{x}}(\hat{\boldsymbol{\omega}}(t), \hat{\boldsymbol{\mu}}(t))$.

Introduce the event

$$\mathcal{E} = \left\{ F_{\boldsymbol{\mu}}(\hat{\boldsymbol{\omega}}(t)) \overset{t\to\infty}{\longrightarrow} \max_{\boldsymbol{\omega} \in \Sigma} F_{\boldsymbol{\mu}}(\boldsymbol{\omega}) \text{ and } \hat{\boldsymbol{\mu}}(t) \overset{t\to\infty}{\longrightarrow} \boldsymbol{\mu} \right\}.$$

Because of (a), Theorem 5 in Appendix E ensures that $\mathbb{P}_{\boldsymbol{\mu}}[\mathcal{E}] = 1$. Also, by the uniform continuity of $F_{\boldsymbol{\mu}}(\boldsymbol{\omega})$ in $\boldsymbol{\mu}$ for an arbitrary $\boldsymbol{\omega} \in \Sigma_+$ (Lemma 7 in D.3.3),

$$\max_{\boldsymbol{\omega} \in \Sigma_+} \left| F_{\hat{\boldsymbol{\mu}}(t)}(\boldsymbol{\omega}) - F_{\boldsymbol{\mu}}(\boldsymbol{\omega}) \right| \overset{t\to\infty}{\longrightarrow} 0$$

almost surely, and hence by the triangle inequality, this implies that

$$\mathbb{P}_{\boldsymbol{\mu}}\left[ F_{\hat{\boldsymbol{\mu}}(t)}(\hat{\boldsymbol{\omega}}(t)) \overset{t\to\infty}{\longrightarrow} \max_{\boldsymbol{\omega} \in \Sigma} F_{\boldsymbol{\mu}}(\boldsymbol{\omega}) \right] = 1.$$

For any $\epsilon \in (0,1)$, under $\mathcal{E}$, there exists a positive integer $T_\epsilon > \max\{c_1, 4|\mathcal{X}_0|\}$ such that for any $t \geq T_\epsilon$, we have

$$F_{\hat{\boldsymbol{\mu}}(t)}(\hat{\boldsymbol{\omega}}(t)) \geq (1-\epsilon)\max_{\boldsymbol{\omega} \in \Sigma} F_{\boldsymbol{\mu}}(\boldsymbol{\omega}), \ \max\left\{ \frac{1}{\triangle_{\min}(\hat{\boldsymbol{\mu}}(t))}, \|\hat{\boldsymbol{\mu}}(t)\|_\infty \right\} \leq \sqrt{t}, \text{ and } \epsilon_t \leq \epsilon, \quad (20)$$

where the second inequality is due to (a) and the third is because $\epsilon_t \to 0$. So, the stopping time (7) can be upper bounded by

$$\tau \leq T_\epsilon + \inf\left\{ t > T_\epsilon : t\hat{F}_t > (1+\epsilon)\beta\left(t, \frac{(4|\mathcal{X}_0|-1)\delta}{4|\mathcal{X}_0|}\right) \right\}$$

$$\leq T_\epsilon + \inf\left\{ t > T_\epsilon : tF_{\hat{\boldsymbol{\mu}}(t)}(\hat{\boldsymbol{\omega}}(t)) > (1+\epsilon)\beta\left(t, \frac{(4|\mathcal{X}_0|-1)\delta}{4|\mathcal{X}_0|}\right) \right\}$$

$$\leq T_\epsilon + \inf\left\{ t > T_\epsilon : t(1-\epsilon)\max_{\boldsymbol{\omega} \in \Sigma} F_{\boldsymbol{\mu}}(\boldsymbol{\omega}) > (1+\epsilon)\beta\left(t, \frac{(4|\mathcal{X}_0|-1)\delta}{4|\mathcal{X}_0|}\right) \right\}$$

$$\leq T_\epsilon + \inf\left\{ t > T_\epsilon : \frac{(1-\epsilon)t}{(1+\epsilon)T^\star(\boldsymbol{\mu})} > \ln\left( \frac{c_2 t}{\delta} \cdot \frac{4|\mathcal{X}_0|}{4|\mathcal{X}_0|-1} \right) \right\}$$

$$\leq 2T_\epsilon + \left(\frac{1+\epsilon}{1-\epsilon}\right)T^\star(\boldsymbol{\mu})H\left(\frac{1}{\delta} \cdot \frac{8c_2}{7}\left(\frac{1+\epsilon}{1-\epsilon}\right)T^\star(\boldsymbol{\mu})\right). \quad (21)$$

where the first inequality uses the last two inequalities of (20), the second inequality uses (b), the third inequality is based on the first inequality of (20), the fourth uses $T^\star(\boldsymbol{\mu})^{-1} = \max_{\boldsymbol{\omega} \in \Sigma} F_{\boldsymbol{\mu}}(\boldsymbol{\omega})$ and (9), and the last inequality results from $(4|\mathcal{X}_0|)/(4|\mathcal{X}_0|-1) \leq 8/7$ (as $|\mathcal{X}_0| \geq 2$), and an application of Lemma 9 with

$$\alpha = 1, \quad b_1 = \frac{1-\epsilon}{1+\epsilon} \cdot \frac{1}{T^\star(\boldsymbol{\mu})} \quad \text{and} \quad b_2 = \frac{8c_2}{7} \cdot \frac{1}{\delta}.$$

Finally, as $\epsilon \in (0,1)$ can be arbitrarily small, (21) implies that

$$\mathbb{P}_{\boldsymbol{\mu}}\left[ \limsup_{\delta \to 0} \frac{\tau}{\ln \delta^{-1}} \leq T^\star(\boldsymbol{\mu}) \right] = 1.$$

### D.3 Non-asymptotic sample complexity (Theorem 4 (ii))

We establish the following non-asymptotic upper bound on $\mathbb{E}_{\boldsymbol{\mu}}[\tau]$: for any $\tilde{\epsilon}, \epsilon \in (0,1)$ small enough,

$$\mathbb{E}_{\boldsymbol{\mu}}[\tau] \leq \frac{(1+\tilde{\epsilon})^2}{T^\star(\boldsymbol{\mu})^{-1} - 6\epsilon}H\left(\frac{1}{\delta} \cdot \frac{8c_2}{7} \cdot \frac{(1+\tilde{\epsilon})^2}{T^\star(\boldsymbol{\mu})^{-1} - 6\epsilon}\right) + \Psi(\epsilon, \tilde{\epsilon}),$$

where $H(x) = \ln x + \ln\ln x + 1$ and $\Psi(\epsilon, \tilde{\epsilon})$ is defined in (34).

Note that this directly implies the asymptotic optimality. Indeed, when $\delta \to 0$, we get:

$$\limsup_{\delta \to 0} \frac{\mathbb{E}_{\boldsymbol{\mu}}[\tau]}{\ln \delta^{-1}} \leq \frac{(1+\tilde{\epsilon})^2}{T^\star(\boldsymbol{\mu})^{-1} - 6\epsilon}.$$

As $\epsilon, \tilde{\epsilon}$ can be set arbitrarily small and $\mathrm{kl}(\delta, 1-\delta) \approx \ln \delta^{-1}$ as $\delta \to 0$, it matches the sample complexity lower bound (1) (Theorem 7 in Appendix K) asymptotically.

Throughout this section, we assume $\boldsymbol{\mu} \in \Lambda$ is given and take any $\epsilon \in (0,1)$ satisfying the following:

$$\epsilon < \min\left\{1, \frac{2D^2\triangle_{\min}^2}{K}, \frac{1}{6T^\star(\boldsymbol{\mu})}\right\} \leq \min\left\{1, \frac{2D^2\triangle_{\min}^2}{K}, \frac{D^2\|\boldsymbol{\mu}\|_\infty^2}{3}\right\}, \tag{22}$$

where the second inequality is because $T^\star(\boldsymbol{\mu})^{-1} \leq \ell = 2D^2\|\boldsymbol{\mu}\|_\infty^2$ by Lemma 22 in Appendix I. The assumption of $\epsilon < \min\{1, \frac{2D^2\triangle_{\min}^2}{K}, \frac{D^2\|\boldsymbol{\mu}\|_\infty^2}{3}\}$ is used to define the good events introduced in D.3.1 as well as to derive several necessary technical lemmas summarized in D.3.3.

### D.3.1 Good events

Since in early rounds, the estimation of $\hat{\boldsymbol{\mu}}(t)$ is noisy, we introduce two threshold functions, $\underline{h}$ and $\overline{h}$, on the round index $T$:

$$\begin{cases} \underline{h}(T) & = \min\{t \in \mathbb{N} : t \geq T^a, \sqrt{t/|\mathcal{X}_0|} \in \mathbb{N}\} \\ \overline{h}(T) & = \min\{t \in \mathbb{N} : t \geq T^b\underline{h}(T), \sqrt{t/|\mathcal{X}_0|} \in \mathbb{N}\} \end{cases}, \tag{23}$$

where $a, b \in (0,1)$ and $a + b < 1$ will be explained later in (27). Now, we define our good events:

$$\mathcal{E}_{1,\epsilon}(T) = \bigcap_{t=\underline{h}(T)}^{T} \mathcal{E}_{1,\epsilon}^{(t)} \quad \text{and} \quad \mathcal{E}_{2,\epsilon}(T) = \bigcap_{t=\underline{h}(T)}^{T} \mathcal{E}_{2,\epsilon}^{(t)}, \tag{24}$$

where $\mathcal{E}_{1,\epsilon}^{(t)} = \left\{\left\langle \nabla \bar{F}_{\hat{\boldsymbol{\mu}}(t-1),\eta_t}(\hat{\boldsymbol{\omega}}(t-1)), \boldsymbol{x}(t)\right\rangle \geq \max_{\boldsymbol{x} \in \mathcal{X}} \left\langle \nabla \bar{F}_{\hat{\boldsymbol{\mu}}(t-1),\eta_t}(\hat{\boldsymbol{\omega}}(t-1)), \boldsymbol{x}\right\rangle - \epsilon\right\}$ and $\mathcal{E}_{2,\epsilon}^{(t)} = \left\{\|\hat{\boldsymbol{\mu}}(t-1) - \boldsymbol{\mu}\|_\infty < \frac{\epsilon}{24D^3\|\boldsymbol{\mu}\|_\infty}\right\}$.

$\mathcal{E}_{1,\epsilon}^{(t)}$ is the event when the solution of FW update is bounded by at most $\epsilon$, and $\mathcal{E}_{2,\epsilon}^{(t)}$ is the event when the empirical estimate of $\boldsymbol{\mu}$ is sufficiently accurate. Under $\mathcal{E}_{2,\epsilon}^{(t)}$, the uniform continuity shown in Lemma 7 in D.3.3 ensures that:

$$|F_{\hat{\boldsymbol{\mu}}(t-1)}(\boldsymbol{\omega}) - F_{\boldsymbol{\mu}}(\boldsymbol{\omega})| < \epsilon, \quad \forall \boldsymbol{\omega} \in \Sigma_+,$$
$$\left|\left\langle \nabla \bar{F}_{\hat{\boldsymbol{\mu}}(t-1),\eta}(\boldsymbol{\omega}) - \nabla \bar{F}_{\boldsymbol{\mu},\eta}(\boldsymbol{\omega}), \boldsymbol{x} - \boldsymbol{\omega}\right\rangle\right| < \epsilon, \quad \forall (\boldsymbol{\omega}, \boldsymbol{x}) \in \Sigma_+ \times \mathcal{X}, \forall \eta \in (0, \min_{k \in [K]} \omega_k).$$

The second inequality enables the duality gap of FW algorithm to be controlled, leading to the convergence of P-FWS. Let

$$M = \max\left\{(4|\mathcal{X}_0|)^{\frac{1}{a}}, \left(\frac{4K^2}{\epsilon^2 D^2|\mathcal{X}_0|}\right)^{\frac{1}{a}}, \left(\frac{2}{\triangle_{\min}(\boldsymbol{\mu})}\right)^{\frac{2}{a}}, \left(\frac{3\|\boldsymbol{\mu}\|_\infty}{2}\right)^{\frac{2}{a}}\right\}$$
$$+ \max\left\{\left(\frac{\ell}{\epsilon}\right)^{\frac{1}{b}}, \left(\frac{5\ell K^2}{\epsilon\sqrt{|\mathcal{X}_0|}}\right)^{\frac{2}{a+b}}\right\}, \tag{25}$$

then overall, we have (Theorem 5 in D.3.3): for any $t \geq \overline{h}(M)$,

$$\max_{\boldsymbol{\omega} \in \Sigma} F_{\boldsymbol{\mu}}(\boldsymbol{\omega}) - F_{\boldsymbol{\mu}}(\hat{\boldsymbol{\omega}}(t)) \leq 5\epsilon, \ \triangle_{\min}(\hat{\boldsymbol{\mu}}(t)) \geq \frac{\triangle_{\min}(\boldsymbol{\mu})}{2}, \text{ and } \|\hat{\boldsymbol{\mu}}(t)\|_\infty \leq \frac{3\|\boldsymbol{\mu}\|_\infty}{2}. \tag{26}$$

Finally, the values of $a, b$ are set to the following:

$$a = \frac{7}{9} \quad \text{and} \quad b = \frac{1}{9}. \tag{27}$$

This choices will balance the leading order between $\epsilon^{-1}$ and $\tilde{\epsilon}^{-1}$ in the $\delta$-independent terms (34) of the non-asymptotic upper bound (which will be shown later).

### D.3.2 Proof of non-asymptotic sample complexity

Let $\delta \in (0, 1)$. We claim that:

$$\mathbb{E}_{\boldsymbol{\mu}}[\tau] \leq \sum_{T=1}^{\infty} \mathbb{P}_{\boldsymbol{\mu}}[\tau \geq T] \leq T_0(\delta) + \sum_{T=M+1}^{\infty} \mathbb{P}_{\boldsymbol{\mu}}[(\mathcal{E}_{1,\epsilon}(T) \cap \mathcal{E}_{2,\epsilon}(T))^c], \tag{28}$$

where $T_0(\delta) = \inf \left\{ T \geq M : \overline{h}(T) + \frac{(1+\epsilon_T)}{T^\star(\boldsymbol{\mu})^{-1} - 6\epsilon} \beta\left(T, \frac{(4|\mathcal{X}_0|-1)\delta}{4|\mathcal{X}_0|}\right) \leq T \right\}$. The proof is completed by bounding each term in the right-hand side of (28).

Proof of (28): Suppose $T \geq M$ and $\mathcal{E}_{1,\epsilon}(T) \cap \mathcal{E}_{2,\epsilon}(T)$ holds. Observe that

$$\min\{\tau, T\} \leq \overline{h}(T) + \sum_{t=\lceil \overline{h}(T) \rceil}^{T} \mathbb{1}\{\tau > t\}.$$

To derive an upper bound of $\sum_{t=\lceil \overline{h}(T)\rceil}^{T} \mathbb{1}\{\tau > t\}$, recall the stopping rule (7) that

$$\tau = \inf \left\{ t > 4|\mathcal{X}_0| : \frac{t\hat{F}_t}{1+\epsilon_t} > \beta\left(t, \frac{(4|\mathcal{X}_0|-1)\delta}{4|\mathcal{X}_0|}\right), \max\left\{ \frac{1}{\triangle_{\min}(\hat{\boldsymbol{\mu}}(t))}, \|\hat{\boldsymbol{\mu}}(t)\|_\infty \right\} \leq \sqrt{t} \right\}$$

$$\leq \inf \left\{ t \geq \overline{h}(M) : t\hat{F}_t > (1+\epsilon_t) \beta\left(t, \frac{(4|\mathcal{X}_0|-1)\delta}{4|\mathcal{X}_0|}\right) \right\}$$

$$\leq \inf \left\{ t \geq \overline{h}(M) : t(T^\star(\boldsymbol{\mu})^{-1} - 6\epsilon) > (1+\epsilon_t) \beta\left(t, \frac{(4|\mathcal{X}_0|-1)\delta}{4|\mathcal{X}_0|}\right) \right\},$$

where the first inequality uses (26), and the second follows from Lemma 7 and Theorem 5 in D.3.3:

$$|F_{\hat{\boldsymbol{\mu}}(t-1)}(\hat{\boldsymbol{\omega}}(t-1)) - F_{\boldsymbol{\mu}}(\hat{\boldsymbol{\omega}}(t-1))| < \epsilon \quad \text{and} \quad T^\star(\boldsymbol{\mu})^{-1} - F_{\boldsymbol{\mu}}(\hat{\boldsymbol{\omega}}(t)) \leq 5\epsilon, \tag{29}$$

and the fact that $\hat{F}_t \geq F_{\hat{\boldsymbol{\mu}}(t)}(\hat{\boldsymbol{\omega}}(t))$. Hence, $\sum_{t=\overline{h}(T)}^{T} \mathbb{1}\{\tau > t\}$ is upper bounded by

$$\sum_{t=\overline{h}(T)}^{T} \mathbb{1}\left\{ t(T^\star(\boldsymbol{\mu})^{-1} - 6\epsilon) \leq (1+\epsilon_t) \beta\left(t, \frac{(4|\mathcal{X}_0|-1)\delta}{4|\mathcal{X}_0|}\right) \right\} \leq \frac{(1+\epsilon_T)}{T^\star(\boldsymbol{\mu})^{-1} - 6\epsilon} \beta\left(T, \frac{(4|\mathcal{X}_0|-1)\delta}{4|\mathcal{X}_0|}\right).$$

By defining $T_0(\delta)$ as done in (28), we get (28), i.e.,

$$\mathbb{E}_{\boldsymbol{\mu}}[\tau] \leq \sum_{T=1}^{\infty} \mathbb{P}_{\boldsymbol{\mu}}[\tau \geq T] \leq T_0(\delta) + \sum_{T=M+1}^{\infty} \mathbb{P}_{\boldsymbol{\mu}}[(\mathcal{E}_{1,\epsilon}(T) \cap \mathcal{E}_{2,\epsilon}(T))^c]$$

because $\mathcal{E}_{1,\epsilon}(T) \cap \mathcal{E}_{2,\epsilon}(T) \subseteq \{\tau \leq T\}$ for any $T \geq T_0(\delta)$.

Now, we proceed with the proof by upper-bounding each term in the right-hand side of (28).

Bounding $T_0(\delta)$: Introduce $\tilde{\epsilon} \in (0, 1)$ that can be chosen arbitrarily small. Notice that

$$T - \overline{h}(T) = T - T^{a+b} \geq \frac{T}{1+\tilde{\epsilon}} \quad \text{when} \quad T \geq \left(1 + \frac{1}{\tilde{\epsilon}}\right)^{\frac{1}{1-(a+b)}}, \tag{30}$$

$$\epsilon_T = T^{-\frac{1}{9}} \leq \tilde{\epsilon} \quad \text{when} \quad T \geq \left(\frac{1}{\tilde{\epsilon}}\right)^9, \tag{31}$$

where the first inequality results from a simple rearrangement, and the second substitutes $\epsilon_t = t^{-1/9}$. Then, it follows from (9) that:

$$T_0(\delta) \leq \inf \left\{ T \geq \max\left\{ M, (1+\tilde{\epsilon}^{-1})^{\frac{1}{1-(a+b)}}, \tilde{\epsilon}^{-9} \right\} : \frac{(1+\tilde{\epsilon})\beta\left(T, \frac{3\delta}{4}\right)}{T^\star(\boldsymbol{\mu})^{-1} - 6\epsilon} \leq \frac{T}{1+\tilde{\epsilon}} \right\}$$

$$\leq \inf \left\{ T \geq \max\left\{ M, (1+\tilde{\epsilon}^{-1})^{\frac{1}{1-(a+b)}}, \tilde{\epsilon}^{-9}, c_1 \right\} : \ln\left(\frac{4c_2 T}{3\delta}\right) \leq \frac{T^\star(\boldsymbol{\mu})^{-1} - 6\epsilon}{(1+\tilde{\epsilon})^2} \cdot T \right\}$$

$$\leq \max\left\{ M, (1+\tilde{\epsilon}^{-1})^{\frac{1}{1-(a+b)}}, \tilde{\epsilon}^{-9}, c_1 \right\} + \frac{(1+\tilde{\epsilon})^2}{T^\star(\boldsymbol{\mu})^{-1} - 6\epsilon} \times H\left(\frac{4c_2}{3\delta} \cdot \frac{(1+\tilde{\epsilon})^2}{T^\star(\boldsymbol{\mu})^{-1} - 6\epsilon}\right),$$
$$\tag{32}$$

where the first inequality uses (30)-(31) and $\frac{4|\mathcal{X}_0|-1}{4|\mathcal{X}_0|} \geq \frac{3}{4}$ (as $|\mathcal{X}_0| \geq 2$ is shown in Lemma 23 in Appendix J), the second inequality is due to (9), and the last results from an application of Lemma 9 in Appendix D.3.3 with

$$\alpha = 1, \quad b_1 = \frac{T^\star(\boldsymbol{\mu})^{-1} - 6\epsilon}{(1+\tilde{\epsilon})^2}, \quad \text{and} \quad b_2 = \frac{4c_2}{3\delta}.$$

Bounding $\sum_{T=M+1}^{\infty} \mathbb{P}_{\boldsymbol{\mu}}[(\mathcal{E}_{1,\epsilon}(T) \cap \mathcal{E}_{2,\epsilon}(T))^c]$: By Lemma 8 in Appendix D.3.3, it is upper bounded by

$$2K \left( \frac{3}{\min\{1, \frac{\epsilon^2}{8\ell^2 K^3 D^2}\}^{2+\frac{2}{a}}} + 2 \left( \frac{2304 D^6 \|\boldsymbol{\mu}\|_\infty^2 \sqrt{|\mathcal{X}_0|}}{\epsilon^2} \right)^{2+\frac{2}{a}} \right) \Gamma \left( 2 + \frac{2}{a} \right). \qquad (33)$$

Putting things together: Finally, substituting $(a,b) = (\frac{7}{9}, \frac{1}{9})$ into (25)-(32)-(33) yields that:

- $T_0(\delta) \leq M + \left(1 + \frac{1}{\tilde{\epsilon}}\right)^9 + \left(\frac{1}{\tilde{\epsilon}}\right)^9 + c_1 + \frac{(1+\tilde{\epsilon})^2}{T^\star(\boldsymbol{\mu})^{-1}-6\epsilon} \times H\left(\frac{4c_2}{3\delta} \cdot \frac{(1+\tilde{\epsilon})^2}{T^\star(\boldsymbol{\mu})^{-1}-6\epsilon}\right)$

- $M \leq \max\{(4|\mathcal{X}_0|)^{\frac{9}{7}}, (\frac{4K^2}{\epsilon^2 D^2 |\mathcal{X}_0|})^{\frac{9}{7}}, (\frac{4}{\triangle_{\min}^2})^{\frac{9}{7}}, (\frac{9\|\boldsymbol{\mu}\|_\infty^2}{4})^{\frac{9}{7}}\} + \max\{(\frac{\ell}{\epsilon})^9, (\frac{5\ell K^2}{\epsilon\sqrt{|\mathcal{X}_0|}})^{2.25}\}$

- $\sum_{T=M}^{\infty} \mathbb{P}_{\boldsymbol{\mu}}[(\mathcal{E}_{1,\epsilon}(T) \cap \mathcal{E}_{2,\epsilon}(T))^c] < \frac{78K}{\epsilon^{10}} \left( 2^{15} K^{15} D^{10} \ell^{10} + 2^{41} 3^9 D^{30} \|\boldsymbol{\mu}\|_\infty^{10} |\mathcal{X}_0|^{2.5} \right)$

where simplifications are obtained remarking that $\Gamma(2 + \frac{2}{a}) \leq 13$ and $2 + \frac{2}{a} < 5$. Therefore, substituting $\ell = 2D^2 \|\boldsymbol{\mu}\|_\infty^2$ (defined in Appendix I) and $78 < 2^7$, $3^9 \leq 2^{15}$, and $4^{9/7} < 6$ lead to:

$$\mathbb{E}_{\boldsymbol{\mu}}[\tau] \leq \frac{(1+\tilde{\epsilon})^2}{T^\star(\boldsymbol{\mu})^{-1}-6\epsilon} \times H\left(\frac{4c_2}{3\delta} \cdot \frac{(1+\tilde{\epsilon})^2}{T^\star(\boldsymbol{\mu})^{-1}-6\epsilon}\right) + \Psi(\epsilon, \tilde{\epsilon}),$$

where

$$\Psi(\epsilon, \tilde{\epsilon}) = 6 \max\left\{|\mathcal{X}_0|, \frac{K^2}{\epsilon^2 D^2 |\mathcal{X}_0|}, \frac{1}{\triangle_{\min}^2}, \|\boldsymbol{\mu}\|_\infty^2\right\}^{\frac{9}{7}} + \max\left\{\frac{2^9 D^{18} \|\boldsymbol{\mu}\|_\infty^{18}}{\epsilon^9}, \frac{10^{2.25} D^{4.5} \|\boldsymbol{\mu}\|_\infty^{4.5} K^2}{\epsilon^{2.25}|\mathcal{X}_0|^{1.125}}\right\}$$

$$+ \left(1 + \frac{1}{\tilde{\epsilon}}\right)^9 + \left(\frac{1}{\tilde{\epsilon}}\right)^9 + c_1 + \frac{2^{32} K D^{30} \|\boldsymbol{\mu}\|_\infty^{10} \left(K^{15} \|\boldsymbol{\mu}\|_\infty^{10} + 2^{31}|\mathcal{X}_0|^{2.5}\right)}{\epsilon^{10}}. \qquad (34)$$

### D.3.3   Technical lemmas

The most important step in Theorem 4 is to bound the term $\sum_{T=M+1}^{\infty} \mathbb{P}_{\boldsymbol{\mu}}[(\mathcal{E}_{1,\epsilon}(T) \cap \mathcal{E}_{2,\epsilon}(T))^c]$ in (28) explicitly in terms of $K$, $\|\boldsymbol{\mu}\|_\infty$ and $\epsilon$. For this purpose, inspired by Assumption 3 in [WTP21], we developed Proposition 4 (see Appendix G.2 for the proof) and combine it with the mean-valued theorem to derive our main continuity results in Lemma 7 (see Appendix G for the proof). Throughout this section, we fix $\boldsymbol{\mu} \in \Lambda$ and denote $\triangle_{\min}(\boldsymbol{\mu})$ by $\triangle_{\min}$.

**Lemma 7.** Let $\epsilon \in (0, \frac{2D^2 \triangle_{\min}^2}{K})$. Then, any $\boldsymbol{\pi} \in \mathbb{R}^K$ with $\|\boldsymbol{\pi} - \boldsymbol{\mu}\|_\infty < \frac{\epsilon}{24D^3 \|\boldsymbol{\mu}\|_\infty}$ satisfies the following:

$$|F_{\boldsymbol{\mu}}(\boldsymbol{\omega}) - F_{\boldsymbol{\pi}}(\boldsymbol{\omega})| < \epsilon, \quad \forall \boldsymbol{\omega} \in \Sigma_+ \qquad (35)$$

$$|\langle \nabla \bar{F}_{\boldsymbol{\pi}, \eta}(\boldsymbol{\omega}) - \nabla \bar{F}_{\boldsymbol{\mu}, \eta}(\boldsymbol{\omega}), \boldsymbol{x} - \boldsymbol{\omega}\rangle| < \epsilon, \quad \forall (\boldsymbol{\omega}, \boldsymbol{x}) \in \Sigma_+ \times \mathcal{X}, \forall \eta \in (0, \min_{k \in [K]} \omega_k). \qquad (36)$$

Our main concentration result with error specified explicitly in terms of $\epsilon$ is (see Appendix F for the proof):

**Lemma 8.** Let $\epsilon \in (0, \frac{2D^2 \triangle_{\min}^2}{K})$ and $M$ be defined as in (25). Then,

$$\sum_{T=M+1}^{\infty} \mathbb{P}_{\boldsymbol{\mu}}[(\mathcal{E}_{1,\epsilon}(T) \cap \mathcal{E}_{2,\epsilon}(T))^c] < 2K \left( \frac{3}{A_1(\epsilon)^{2+\frac{2}{a}}} + \frac{2}{A_2(\epsilon)^{2+\frac{2}{a}}} \right) \Gamma\left(2 + \frac{2}{a}\right),$$

where $A_1(\epsilon) = \min\{1, \frac{\epsilon^2}{8\ell^2 K^3 D^2}\}$, $A_2(\epsilon) = \frac{\epsilon^2}{2304 D^6 \|\boldsymbol{\mu}\|_\infty^2 \sqrt{|\mathcal{X}_0|}}$, and $\Gamma$ denotes the gamma function $\Gamma(z) = \int_0^\infty t^{z-1} e^{-t} dt$ for any $z > 0$.

We remark that Lemma 8 sharpens a similar result, Lemma 2 in [WTP21], by a factor of $e^K$ after performing a more careful analysis.

Under good events $\mathcal{E}_{1,\epsilon}(T) \cap \mathcal{E}_{2,\epsilon}(T)$, we show Theorem 5, the convergence of P-FWS when $\hat{\boldsymbol{\mu}}(t)$ is replaced with $\boldsymbol{\mu}$. As shown in Appendix D.3.2, the extra error due to this replacement is controlled, thanks to Lemma 7.

**Theorem 5.** *Let $\epsilon \in (0, \min\{1, \frac{2D^2 \triangle_{\min}^2}{K}\})$ and $T$ be an integer at least larger than*

$$\max\left\{(4|\mathcal{X}_0|)^{\frac{1}{a}}, \left(\frac{4K^2}{\epsilon^2 D^2 |\mathcal{X}_0|}\right)^{\frac{1}{a}}, \left(\frac{2}{\triangle_{\min}}\right)^{\frac{2}{a}}, \left(\frac{3 \|\boldsymbol{\mu}\|_\infty}{2}\right)^{\frac{2}{a}}\right\} + \max\left\{\left(\frac{\ell}{\epsilon}\right)^{\frac{1}{b}}, \left(\frac{5\ell K^2}{\epsilon\sqrt{|\mathcal{X}_0|}}\right)^{\frac{2}{a+b}}\right\}.$$

*Under $\mathcal{E}_{1,\epsilon}(T) \cap \mathcal{E}_{2,\epsilon}(T)$, Algorithm 2 with (10) satisfies that: for any $t = \overline{h}(T), \overline{h}(T) + 1 \cdots, T$,*

*(i)* $\max_{\boldsymbol{\omega} \in \Sigma} F_{\boldsymbol{\mu}}(\boldsymbol{\omega}) - F_{\boldsymbol{\mu}}(\hat{\boldsymbol{\omega}}(t)) \leq 5\epsilon$, *(ii)* $\triangle_{\min}(\hat{\boldsymbol{\mu}}(t)) \geq \frac{\triangle_{\min}}{2}$, *and (iii)* $\|\hat{\boldsymbol{\mu}}(t)\|_\infty \leq \frac{3 \|\boldsymbol{\mu}\|_\infty}{2}$.

The proof of Theorem 5 is given in Appendix E.

Finally, the last ingredient is Lemma 9.

**Lemma 9** (Lemma 18 in [GK16]). *Let $\alpha \in [1, \frac{e}{2}]$ and $b_1, b_2 > 0$. Then,*

$$x = \frac{1}{b_1}\left(\ln\left(\frac{b_2 e}{b_1^\alpha}\right) + \ln\ln\left(\frac{b_2}{b_1^\alpha}\right)\right)$$

*satisfies $b_1 x \geq \ln(b_2 x^\alpha)$.*

### D.4 Computational complexity (Theorem 4 (iii))

In this section, we analyze the computational complexity of P-FWS running with (10) in terms of the number of calls to LM Oracle. We will show that the expected number of LM Oracle calls is upper bounded by a polynomial in $\ln \delta^{-1}$, $K$, $\|\boldsymbol{\mu}\|_\infty$ and $\triangle_{\min}(\boldsymbol{\mu})^{-1}$.

**Proof** The construction of $\mathcal{X}_0$ and computation of $\hat{\boldsymbol{\imath}}$ merely takes $\mathcal{O}(KD)$ calls to LM Oracle. The overall complexity is dominated by the LM Oracle calls performed from $4|\mathcal{X}_0| + 1$ to round $\tau$, analyzed as follows.

Per-round complexity: Fix $t \in \{4|\mathcal{X}_0| + 1, \cdots, \tau\}$. Recall from P-FWS that the FW update in round $t$ and the stopping rule in round $t - 1$ are computed only if:

$$\max\{\triangle_{\min}(\hat{\boldsymbol{\mu}}(t-1))^{-1}, \|\hat{\boldsymbol{\mu}}(t-1)\|_\infty\} \leq \sqrt{t-1}. \tag{37}$$

Otherwise, forced-exploration procedure is invoked. Verifying (37) takes at most $D + 1$ calls.[6] The computation of $\hat{F}_{t-1}$ and $\boldsymbol{i}^\star(\nabla \tilde{F}_{\hat{\boldsymbol{\mu}}(t-1), \eta_t, n_t}(\hat{\boldsymbol{\omega}}(t-1)))$ by Theorem 3 in Appendix C.2 takes at most

$$\mathcal{O}\left(D + \frac{\|\hat{\boldsymbol{\mu}}(t-1)\|_\infty^4 \|\hat{\boldsymbol{\omega}}(t-1)^{-1}\|_\infty^2 K^3 D^5 \ln K}{F_{\hat{\boldsymbol{\mu}}(t-1)}(\hat{\boldsymbol{\omega}}(t-1))^2}\left(\frac{\ln(t^2 \delta^{-1})}{\epsilon_t^2} + \frac{n_t \ln \theta_t^{-1}}{\rho_t^2}\right)\right) \tag{38}$$

calls to LM Oracle. To evaluate (38), we need a lower bound on $F_{\hat{\boldsymbol{\mu}}(t-1)}(\hat{\boldsymbol{\omega}}(t-1))$. By Proposition 1 (c) in Appendix C.1, one evaluates $F_{\hat{\boldsymbol{\mu}}(t-1)}(\hat{\boldsymbol{\omega}}(t-1))$ in closed-form:

$$F_{\hat{\boldsymbol{\mu}}(t-1)}(\hat{\boldsymbol{\omega}}(t-1)) = \min_{\boldsymbol{x} \neq \boldsymbol{i}^\star(\hat{\boldsymbol{\mu}}(t-1))} \frac{\triangle_{\boldsymbol{x}}(\hat{\boldsymbol{\mu}}(t-1))^2}{2 \langle \boldsymbol{x} \oplus \boldsymbol{i}^\star(\hat{\boldsymbol{\mu}}(t-1)), \hat{\boldsymbol{\omega}}(t-1)^{-1}\rangle} \geq \frac{\min_{k \in [K]} \hat{\omega}_k(t-1)}{4D(t-1)},$$

where the inequality results from (37) that $\triangle_{\min}(\hat{\boldsymbol{\mu}}(t-1)) \geq \frac{1}{\sqrt{t-1}}$, $\|\boldsymbol{x} \oplus \boldsymbol{x}'\|_1 \leq 2D$ for any $\boldsymbol{x}, \boldsymbol{x}' \in \mathcal{X}$, and $\langle \boldsymbol{y}, \boldsymbol{z}\rangle \leq \|\boldsymbol{y}\|_1 \|\boldsymbol{z}\|_\infty$ for any $\boldsymbol{y}, \boldsymbol{z} \in \mathbb{R}^K$. Further, combining with Lemma 14 (which states $\min_{k \in [K]} \hat{\omega}_k(t-1) \geq \frac{1}{2\sqrt{(t-1)|\mathcal{X}_0|}}$) in Appendix F yields

$$F_{\hat{\boldsymbol{\mu}}(t-1)}(\hat{\boldsymbol{\omega}}(t-1)) \geq \frac{1}{8D\sqrt{|\mathcal{X}_0|}(t-1)^{1.5}}. \tag{39}$$

---

[6]For any $\boldsymbol{\pi} \in \Lambda$, $\triangle_{\min}(\boldsymbol{\pi})$ requires to compute $\boldsymbol{i}^\star(\boldsymbol{\pi})$ and and solve $\max_{\boldsymbol{x} \neq \boldsymbol{i}^\star(\boldsymbol{\pi})} \langle \boldsymbol{\pi}, \boldsymbol{x}\rangle$, where the latter requires at most $D$ calls to the LM Oracle by Lemma 2 in § 2.2.

From (39), $\|\hat{\boldsymbol{\mu}}(t-1)\|_\infty \leq \sqrt{t-1}$, Lemma 14, and substituting the parameters (10) into (38), we know that the number of LM Oracle calls performed at any round $t \geq 4|\mathcal{X}_0| + 1$ is at most

$$\mathcal{O}\left(t^6 |\mathcal{X}_0|^2 K^3 D^7 \ln K \left( \frac{\ln(t^2 \delta^{-1})}{\epsilon_t^2} + \frac{n_t \ln \theta_t^{-1}}{\rho_t^2} \right) \right)$$
$$= \mathcal{O}\left( t^6 |\mathcal{X}_0|^2 K^3 D^7 \ln K \left( t^{\frac{2}{9}} \ln\left(\frac{t}{\delta}\right) + t^{2.75} D^4 |\mathcal{X}_0|^2 \right) \right)$$
$$= \mathcal{O}\left( t^{8.75} \ln\left(\frac{t}{\delta}\right) |\mathcal{X}_0|^4 D^{11} K^3 \ln K \right). \tag{40}$$

Overall complexity: Invoking Theorem 4 in D.3 with $\tilde{\epsilon} = 0.1$ and $\epsilon = \frac{1}{12T^\star(\boldsymbol{\mu})}$ results in

$$\mathbb{E}_{\boldsymbol{\mu}}[\tau] = \mathcal{O}\left( T^\star(\boldsymbol{\mu}) \ln\left( \frac{T^\star(\boldsymbol{\mu})}{\delta} \right) + \frac{1}{\triangle_{\min}^{\frac{18}{7}}} + K^{16} D^{30} \|\boldsymbol{\mu}\|_\infty^{20} T^\star(\boldsymbol{\mu})^{10} \right)$$

which after using $T^\star(\boldsymbol{\mu}) \leq 4KD/\triangle_{\min}^2$ (Lemma 1 in §2.1) becomes

$$\mathbb{E}_{\boldsymbol{\mu}}[\tau] = \mathcal{O}\left( \frac{KD}{\triangle_{\min}^2} \ln\left( \frac{KD}{\delta \triangle_{\min}^2} \right) + \frac{K^{26} D^{40} \|\boldsymbol{\mu}\|_\infty^{20}}{\triangle_{\min}^{20}} \right). \tag{41}$$

Hence, by a summation of (40) over $t = 4|\mathcal{X}_0| + 1$ to $\mathbb{E}_{\boldsymbol{\mu}}[\tau]$, the expected total number of the LM Oracle calls is upper bounded by

$$\mathcal{O}\left( \mathbb{E}_{\boldsymbol{\mu}}[\tau]^{9.75} \ln\left( \frac{\mathbb{E}_{\boldsymbol{\mu}}[\tau]}{\delta} \right) |\mathcal{X}_0|^4 D^{11} K^3 \ln K \right), \tag{42}$$

where the inequality uses integral by parts $\int t^{8.75} \ln t \, dt = \mathcal{O}(t^{9.75} \ln t)$. Remind that $\max\{D, |\mathcal{X}_0|\} \leq K$. Thus, we conclude that (42) is bounded by a polynomial function in $\ln \delta^{-1}$, $\|\boldsymbol{\mu}\|_\infty$, $\triangle_{\min}^{-1}$, and $K$ (due to (41), $\mathbb{E}_{\boldsymbol{\mu}}[\tau]$ is bounded by a polynomial function in the same variables). $\square$

# E    Convergence of `P-FWS` under the good events

Throughout this section, we assume that $\boldsymbol{\mu}$ is fixed and drop $\boldsymbol{\mu}$ from the notation, e.g., $F = F_{\boldsymbol{\mu}}$, $\bar{F}_\eta = \bar{F}_{\boldsymbol{\mu},\eta}$, $\tilde{F}_{\eta,t} = \tilde{F}_{\boldsymbol{\mu},\eta,t}$, and $\triangle_{\min} = \triangle_{\min}(\boldsymbol{\mu})$. Also, we will use $\boldsymbol{\omega}^\star \in \operatorname{argmax}_{\boldsymbol{\omega} \in \Sigma} F(\boldsymbol{\omega})$ to denote any optimal allocation and let $\boldsymbol{i}^\star = \boldsymbol{i}^\star(\boldsymbol{\mu})$. Recall that $\underline{h}(T) \geq T^a$ and $\overline{h}(T) \geq T^{a+b}$ is defined in (23) in Appendix D.3.1 for some $a, b \in (0, 1)$.

**Theorem 5.** *Let $\epsilon \in (0, \min\{1, \frac{2D^2 \triangle_{\min}^2}{K}\})$ and $T$ be an integer at least larger than*

$$\max\left\{(4|\mathcal{X}_0|)^{\frac{1}{a}}, \left(\frac{4K^2}{\epsilon^2 D^2 |\mathcal{X}_0|}\right)^{\frac{1}{a}}, \left(\frac{2}{\triangle_{\min}}\right)^{\frac{2}{a}}, \left(\frac{3\|\boldsymbol{\mu}\|_\infty}{2}\right)^{\frac{2}{a}}\right\} + \max\left\{\left(\frac{\ell}{\epsilon}\right)^{\frac{1}{b}}, \left(\frac{5\ell K^2}{\epsilon\sqrt{|\mathcal{X}_0|}}\right)^{\frac{2}{a+b}}\right\}.$$

*Under $\mathcal{E}_{1,\epsilon}(T) \cap \mathcal{E}_{2,\epsilon}(T)$, Algorithm 2 with (10) satisfies that: for any $t = \overline{h}(T), \overline{h}(T) + 1, \cdots, T$,*

*(i)* $F(\boldsymbol{\omega}^\star) - F(\hat{\boldsymbol{\omega}}(t)) \leq 5\epsilon$, *(ii)* $\triangle_{\min}(\hat{\boldsymbol{\mu}}(t)) \geq \dfrac{\triangle_{\min}}{2}$, *and (iii)* $\|\hat{\boldsymbol{\mu}}(t)\|_\infty \leq \dfrac{3\|\boldsymbol{\mu}\|_\infty}{2}$.

**Proof**    Fix arbitrary $\epsilon$ and $T$ that satisfy the conditions in the statement, and suppose $\mathcal{E}_{1,\epsilon}(T) \cap \mathcal{E}_{2,\epsilon}(T)$ holds. (ii)(iii) directly follows from Lemma 10 (where one can verified that its assumption of Lemma 10 on $T$ is satisfied). With (ii)(iii), the analysis of FW convergence will be greatly simplified as (ii)(iii) ensure that

$$\max\left\{\frac{1}{\triangle_{\min}(\hat{\boldsymbol{\mu}}(t-1))}, \|\hat{\boldsymbol{\mu}}(t-1)\|\right\} \leq \sqrt{t-1}.$$

This means that the forced-exploration procedure will only be invoked by the condition of $\sqrt{t/|\mathcal{X}_0|}$ when $t \geq \underline{h}(T) = T^a$.

Proof of (i) $F(\boldsymbol{\omega}^\star) - F(\hat{\boldsymbol{\omega}}(t)) \leq 5\epsilon$: Fix $t \geq \underline{h}(T)$. As mentioned above, for such $t$, the forced-exploration procedure will be invoked only when $\sqrt{t/|\mathcal{X}_0|} \in \mathbb{N}$. To specify the rounds performing FW udpates, introduce $s(t) = \lfloor \sqrt{t/|\mathcal{X}_0|} \rfloor - 1$ and define

$$p(t) = (s(t)^2 + 1)|\mathcal{X}_0| \quad \text{and} \quad q(t) = (s(t) + 1)^2 |\mathcal{X}_0| - 1.$$

Notice that $p(t)$ and $q(t)$ are respectively the starting (including) and the ending (including) round of a successive FW update rounds with no forced exploration in between. Let $\phi_t = F(\boldsymbol{\omega}^\star) - \bar{F}_{\eta_t}(\hat{\boldsymbol{\omega}}(t))$ be the error. By a careful analysis, we derive a recursive relationship satisfied by $\phi_t$ (Lemma 11):

$$\begin{cases} t\phi_t \leq (t - |\mathcal{X}_0|)\phi_{t-|\mathcal{X}_0|} + 2\ell\sqrt{D}|\mathcal{X}_0|^2 & \text{if } t = p(t) - 1, \\ t\phi_t \leq (t-1)\phi_{t-1} + 3\epsilon + \ell\left(\eta_{t-1} + \frac{K^2}{2t\eta_t}\right) & \text{if } t \in [p(t), q(t)]. \end{cases} \tag{43}$$

The first case (in round $t = p(t) - 1$) is exactly the ending round of a forced-exploration procedure (from $t - |\mathcal{X}_0|, \cdots, t$), and the second case (in round $t \in [p(t), q(t)]$) is a FW-update round. By repeatedly applying (43), we have

$$\overline{h}(T)\phi_{\overline{h}(T)} \leq \underline{h}(T)\phi_{\underline{h}(T)} + 2\ell\sqrt{D}|\mathcal{X}_0|^2 \left(s(\overline{h}(T)) - s(\underline{h}(T))\right) + 3\sum_{t=\underline{h}(T)}^{\overline{h}(T)}\left(\frac{\ell K^2}{\sqrt{t|\mathcal{X}_0|}} + \epsilon\right)$$

$$\leq \underline{h}(T)\ell + \ell\left(2\sqrt{D}|\mathcal{X}_0|^{1.5} + \frac{3K^2}{\sqrt{|\mathcal{X}_0|}}\right)\left(\sqrt{\overline{h}(T)} - \sqrt{\underline{h}(T)}\right) + 3\epsilon(\overline{h}(T) - \underline{h}(T)),$$

where the second inequality follows from $\phi_{\overline{h}(T)} \leq \max_{\boldsymbol{\omega} \in \Sigma} F_{\boldsymbol{\mu}}(\boldsymbol{\omega}) \leq \ell$ (Lemma 22 in Appendix I), $s(\overline{h}(T)) - s(\underline{h}(T)) \leq \frac{\sqrt{\overline{h}(T)} - \sqrt{\underline{h}(T)}}{\sqrt{\mathcal{X}_0}}$ and $\sum_{t=\underline{h}(T)}^{\overline{h}(T)}\frac{1}{\sqrt{t}} \leq \sqrt{\overline{h}(T)} - \sqrt{\underline{h}(T)}$. Substituting $\underline{h}(T)$ and $\overline{h}(T)$ from (23) and simplifying the terms, we get:

$$F(\boldsymbol{\omega}^\star) - F(\hat{\boldsymbol{\omega}}(t)) \leq \phi_{\overline{h}(T)} \leq \ell T^{-b} + \frac{5\ell K^2}{\sqrt{|\mathcal{X}_0|}}T^{-\frac{a+b}{2}} + 3\epsilon(1 - T^{-b}) \leq 5\epsilon$$

when

$$T \geq \max\left\{(4|\mathcal{X}_0|)^{\frac{1}{a}}, \left(\frac{4K^2}{\epsilon^2 D^2 |\mathcal{X}_0|}\right)^{\frac{1}{a}}, \left(\frac{2}{\triangle_{\min}}\right)^{\frac{2}{a}}, \left(\frac{3\|\boldsymbol{\mu}\|_\infty}{2}\right)^{\frac{2}{a}}\right\} + \max\left\{\left(\frac{\ell}{\epsilon}\right)^{\frac{1}{b}}, \left(\frac{5\ell K^2}{\epsilon\sqrt{|\mathcal{X}_0|}}\right)^{\frac{2}{a+b}}\right\},$$

where the first inequality is due to $F(\hat{\boldsymbol{\omega}}(t)) \geq \bar{F}_{\eta_t}(\hat{\boldsymbol{\omega}}(t))$ by Proposition 2 (i) in §4.1. $\qquad\square$

**Lemma 10.** *Let* $\epsilon \in (0, \min\{1, 2D^2\triangle_{\min}^2/K\})$ *and* $T$ *be a positive integer s.t.*

$$T \geq \max\left\{(4|\mathcal{X}_0|)^{\frac{1}{a}}, \left(\frac{4K^2}{\epsilon^2 D^2 |\mathcal{X}_0|}\right)^{\frac{1}{a}}, \left(\frac{2}{\triangle_{\min}}\right)^{\frac{2}{a}}, \left(\frac{3\|\boldsymbol{\mu}\|_\infty}{2}\right)^{\frac{2}{a}}\right\}. \tag{44}$$

*Suppose* $\mathcal{E}_{1,\epsilon}(T) \cap \mathcal{E}_{2,\epsilon}(T)$ *holds. Then, for any* $t \geq \underline{h}(T)$,

$$\triangle_{\min}(\hat{\boldsymbol{\mu}}(t)) \geq \frac{\triangle_{\min}}{2} \quad and \quad \|\hat{\boldsymbol{\mu}}(t)\|_\infty \leq \frac{3\|\boldsymbol{\mu}\|_\infty}{2}. \tag{45}$$

**Proof** Fix any $T$ satisfying (45) and suppose $\mathcal{E}_{1,\epsilon}(T) \cap \mathcal{E}_{2,\epsilon}(T)$ holds. Consider any $t \geq T^a$ and hence $t \geq 4|\mathcal{X}_0|$. To show the first inequality of (45), from $\mathcal{E}_{2,\epsilon}(T)$ and $\epsilon < 2D^2\triangle_{\min}^2/K$, we have

$$\triangle_{\min}(\hat{\boldsymbol{\mu}}(t-1)) \geq \triangle_{\min} - \frac{2D\epsilon}{24D^3\|\boldsymbol{\mu}\|_\infty} > \triangle_{\min} - \frac{\triangle_{\min}^2}{6K\|\boldsymbol{\mu}\|_\infty} > \frac{\triangle_{\min}}{2},$$

where the last inequality is because $\triangle_{\min} \leq 2D\|\boldsymbol{\mu}\|_\infty$ and $D \leq K$. To show the second inequality of (45), observe that

$$\|\hat{\boldsymbol{\mu}}(t)\|_\infty \leq \|\boldsymbol{\mu}\|_\infty + \frac{\epsilon}{24D^3\|\boldsymbol{\mu}\|_\infty} < \|\boldsymbol{\mu}\|_\infty + \frac{\triangle_{\min}^2}{12KD\|\boldsymbol{\mu}\|_\infty} < \frac{3\|\boldsymbol{\mu}\|_\infty}{2},$$

where the first inequality is because of $\mathcal{E}_{2,\epsilon}(T)$, the second is due to $\epsilon < 2D^2\triangle_{\min}^2/K$, and the last uses $\triangle_{\min} \leq 2D\|\boldsymbol{\mu}\|_\infty$. $\qquad\square$

**Lemma 11.** *Let* $\epsilon > 0$ *and* $t \in \mathbb{N}$ *be such that* (45) *holds. Then, under the event* $\mathcal{E}_{1,\epsilon}^{(t)} \cap \mathcal{E}_{2,\epsilon}^{(t)}$,

$$\begin{cases} t\phi_t \leq (t - |\mathcal{X}_0|)\phi_{t-|\mathcal{X}_0|} + 2\ell\sqrt{D}|\mathcal{X}_0|^2 & \text{if } t = p(t) - 1 \\ t\phi_t \leq (t-1)\phi_{t-1} + 3\epsilon + \ell\left(\eta_{t-1} + \frac{K^2}{2t\eta_t}\right) & \text{if } t \in [p(t), q(t)] \end{cases}, \tag{43}$$

*where* $p(t) = (s(t)^2 + 1)|\mathcal{X}_0|$, $q(t) = (s(t) + 1)^2|\mathcal{X}_0| - 1$, *and* $s(t) = \lfloor\sqrt{t/|\mathcal{X}_0|}\rfloor - 1$.

**Proof** The first case basically follows from the Lipschitzness of $F$ (Appendix I), whereas the second relies on results on stochastic smoothing (Appendix H).

Case $t = p(t) - 1$: In this case, round $t$ is exactly the end (including) round of a forced-exploration procedure. By $\ell$-Lipschitzness of $F$ (Lemma 21 in Appendix I),

$$F(\hat{\boldsymbol{\omega}}(t)) - F(\hat{\boldsymbol{\omega}}(t - |\mathcal{X}_0|)) \geq -\ell\|\hat{\boldsymbol{\omega}}(t) - \hat{\boldsymbol{\omega}}(t - |\mathcal{X}_0|)\|_2 \geq -\frac{\ell\sqrt{D}|\mathcal{X}_0|^2}{t},$$

where the second inequality stems from $\hat{\boldsymbol{\omega}}(t) = \frac{t-|\mathcal{X}_0|}{t}\hat{\boldsymbol{\omega}}(t - |\mathcal{X}_0|) + \frac{|\mathcal{X}_0|}{t}\sum_{\boldsymbol{x}\in\mathcal{X}_0}\boldsymbol{x}$ after performing the forced exploration. It then follows that $\|\hat{\boldsymbol{\omega}}(t) - \hat{\boldsymbol{\omega}}(t - |\mathcal{X}_0|)\|_2 \leq \frac{\sqrt{D}|\mathcal{X}_0|^2}{t}$. By $\max_{\boldsymbol{\omega}\in\Sigma}F(\boldsymbol{\omega}) \leq \ell$ (Lemma 22 in Appendix I) and a rearrangement of the above yields

$$t\phi_t \leq t\phi_{t-|\mathcal{X}_0|} + \ell\sqrt{D}|\mathcal{X}_0|^2 \leq (t - |\mathcal{X}_0|)\phi_{t-|\mathcal{X}_0|} + \ell|\mathcal{X}_0|(\sqrt{D}|\mathcal{X}_0| + 1).$$

The proof is completed after simplifying the terms.

Case: $t \in [p(t), q(t)]$: In this case, round $t$ performs a FW update. For brevity, let $\boldsymbol{z} = \hat{\boldsymbol{\omega}}(t)$ and $\boldsymbol{y} = \hat{\boldsymbol{\omega}}(t-1)$. By $\frac{\ell K}{\eta_t}$-smoothness of $\bar{F}_{\eta_t}$ (Proposition 2 (iii) in §4.1) and $\boldsymbol{z} - \boldsymbol{y} = \frac{1}{t}(\boldsymbol{x}(t) - \boldsymbol{y})$,

$$\bar{F}_{\eta_t}(\boldsymbol{z}) \geq (*) - \frac{\ell K}{2\eta_t}\|\boldsymbol{z} - \boldsymbol{y}\|_2^2 \geq (*) - \frac{\ell K}{2t^2\eta_t}\|\boldsymbol{x}(t) - \boldsymbol{y}\|_2^2 \geq (*) - \frac{\ell K^2}{2t^2\eta_t},$$

where $(*) = \bar{F}_{\eta_t}(\boldsymbol{y}) + \langle \nabla \bar{F}_{\eta_t}(\boldsymbol{y}), \boldsymbol{z} - \boldsymbol{y} \rangle = \bar{F}_{\eta_t}(\boldsymbol{y}) + \frac{1}{t}\langle \nabla \bar{F}_{\eta_t}(\boldsymbol{y}), \boldsymbol{x}(t) - \boldsymbol{y}\rangle$. It follows from $\mathcal{E}_{1,\epsilon}^{(t)} \cap \mathcal{E}_{2,\epsilon}^{(t)}$ and the continuity argument (Lemma 7 in Appendix G.1) that

$$\langle \nabla \bar{F}_{\eta_t}(\boldsymbol{y}), \boldsymbol{x}(t) - \boldsymbol{y}\rangle \geq \langle \nabla \bar{F}_{\hat{\boldsymbol{\mu}}(t-1),\eta_t}(\boldsymbol{y}), \boldsymbol{x}(t) - \boldsymbol{y}\rangle - \epsilon$$
$$\geq \max_{\boldsymbol{x}\in\mathcal{X}} \langle \nabla \bar{F}_{\hat{\boldsymbol{\mu}}(t-1),\eta_t}(\boldsymbol{y}), \boldsymbol{x} - \boldsymbol{y}\rangle - 2\epsilon \geq \max_{\boldsymbol{x}\in\mathcal{X}} \langle \nabla \bar{F}_{\eta_t}(\boldsymbol{y}), \boldsymbol{x} - \boldsymbol{y}\rangle - 3\epsilon.$$

Then, the duality gap [Jag13] and the $\ell$-Lipschitzness of $F$ (Lemma 21 in Appendix I) yield

$$\max_{\boldsymbol{x}\in\mathcal{X}} \langle \nabla \bar{F}_{\eta_t}(\boldsymbol{y}), \boldsymbol{x} - \boldsymbol{y}\rangle \geq \max_{\boldsymbol{\omega}\in\Sigma} \bar{F}_{\eta_t}(\boldsymbol{\omega}) - \bar{F}_{\eta_t}(\boldsymbol{y})$$
$$\geq (F(\boldsymbol{\omega}^\star) - \eta_t\ell) - (\bar{F}_{\eta_{t-1}}(\boldsymbol{y}) + \ell(\eta_{t-1} - \eta_t)) = \phi_{t-1} - \ell\eta_{t-1}.$$

Therefore, $\bar{F}_{\eta_t}(\boldsymbol{z}) \geq \bar{F}_{\eta_t}(\boldsymbol{y}) + \frac{\phi_{t-1}-\ell\eta_{t-1}-3\epsilon}{t} - \frac{\ell K^2}{2t^2\eta_t}$ and subtracting $F(\boldsymbol{\omega}^\star)$ on both sides,

$$\phi_t = F(\boldsymbol{\omega}^\star) - \bar{F}_{\eta_t}(\boldsymbol{z})$$
$$\leq (F(\boldsymbol{\omega}^\star) - \bar{F}_{\eta_t}(\boldsymbol{y})) + \frac{-\phi_{t-1} + \ell\eta_{t-1} + 3\epsilon}{t} + \frac{\ell K^2}{2t^2\eta_t}$$
$$= \frac{t-1}{t}\phi_{t-1} + \frac{1}{t}\left(3\epsilon + \ell\left(\eta_{t-1} + \frac{K^2}{2t\eta_t}\right)\right),$$

which completes the proof. $\qquad\square$

# F Upper bound of $\sum_{T=M+1}^{\infty} \mathbb{P}_{\boldsymbol{\mu}}[(\mathcal{E}_{1,\epsilon}(T) \cap \mathcal{E}_{2,\epsilon}(T))^c]$ under `P-FWS`

Recall from (24) that $\mathcal{E}_{1,\epsilon}(T) = \cap_{t=\underline{h}(T)}^{T} \mathcal{E}_{1,\epsilon}^{(t)}$ and $\mathcal{E}_{2,\epsilon}(T) = \cap_{t=\underline{h}(T)}^{T} \mathcal{E}_{2,\epsilon}^{(t)}$, where

$$\mathcal{E}_{1,\epsilon}^{(t)} = \left\{ \left\langle \nabla \bar{F}_{\hat{\boldsymbol{\mu}}(t-1),\eta_t}(\hat{\boldsymbol{\omega}}(t-1)), \boldsymbol{x}(t) \right\rangle \geq \max_{\boldsymbol{x} \in \mathcal{X}} \left\langle \nabla \bar{F}_{\hat{\boldsymbol{\mu}}(t-1),\eta_t}(\hat{\boldsymbol{\omega}}(t-1)), \boldsymbol{x} \right\rangle - \epsilon \right\},$$

$$\mathcal{E}_{2,\epsilon}^{(t)} = \left\{ \|\hat{\boldsymbol{\mu}}(t-1) - \boldsymbol{\mu}\|_{\infty} < \frac{\epsilon}{24D^3 \|\boldsymbol{\mu}\|_{\infty}} \right\},$$

$T \geq M$ and $M$ is defined in (25). Also, recall $\boldsymbol{x}(t) \in \operatorname{argmax}_{\boldsymbol{x} \in \mathcal{X}} \left\langle \nabla \tilde{F}_{\hat{\boldsymbol{\mu}}(t-1),\eta_t,n_t}(\hat{\boldsymbol{\omega}}(t-1)), \boldsymbol{x} \right\rangle$, where $\nabla \tilde{F}_{\hat{\boldsymbol{\mu}}(t-1),\eta_t,n_t}(\hat{\boldsymbol{\omega}}(t-1))$ is computed by $(\rho_t, \theta_t)$-`MCP` algorithm with

$$(\eta_t, n_t, \rho_t, \theta_t) = \left( \frac{1}{4\sqrt{t|\mathcal{X}_0|}}, \left\lceil t^{\frac{1}{4}} \right\rceil, \frac{1}{16tD^2\mathcal{X}_0}, \frac{1}{t^{\frac{1}{4}} e^{\sqrt{t}}} \right). \tag{10}$$

Our main result Lemma 8 is built by bounding $\mathbb{P}_{\boldsymbol{\mu}}[\mathcal{E}_{1,\epsilon}(T)]$ and $\mathbb{P}_{\boldsymbol{\mu}}[\mathcal{E}_{2,\epsilon}(T)]$ separately with Lemma 12 in F.1 and Lemma 14 in F.2.

**Lemma 8.** *Let* $\epsilon \in (0, 2D^2 \triangle_{\min}^2 / K)$ *and* $M$ *be defined as in* (25) *Then,*

$$\sum_{T=M+1}^{\infty} \mathbb{P}_{\boldsymbol{\mu}}[(\mathcal{E}_{1,\epsilon}(T) \cap \mathcal{E}_{2,\epsilon}(T))^c] < 2K \left( \frac{3}{A_1(\epsilon)^{2+\frac{2}{a}}} + \frac{2}{A_2(\epsilon)^{2+\frac{2}{a}}} \right) \Gamma \left( 2 + \frac{2}{a} \right),$$

*where* $A_1(\epsilon) = \min\{1, \frac{\epsilon^2}{8\ell^2 K^3 D^2}\}$, $A_2(\epsilon) = \frac{\epsilon^2}{2304D^6 \|\boldsymbol{\mu}\|_{\infty}^2 \sqrt{|\mathcal{X}_0|}}$, *and* $\Gamma$ *denotes the gamma function* $\Gamma(z) = \int_0^{\infty} t^{z-1} e^{-t} dt$ *for any* $z > 0$.

**Proof** Fix $\epsilon > 0$. For all $T \geq M$, we have

$$\mathbb{P}_{\boldsymbol{\mu}}[(\mathcal{E}_{1,\epsilon}(T) \cap \mathcal{E}_{2,\epsilon}(T))^c] \leq \mathbb{P}_{\boldsymbol{\mu}}[\mathcal{E}_{1,\epsilon}(T)^c] + \mathbb{P}_{\boldsymbol{\mu}}[\mathcal{E}_{2,\epsilon}(T)^c].$$

Bounding $\mathbb{P}_{\boldsymbol{\mu}}[\mathcal{E}_{1,\epsilon}(T)^c]$: This is done by using Lemma 12 with $v = \hat{\boldsymbol{\omega}}(t)$ and $(\eta, n) = (\eta_t, n_t)$. Before applying Lemma 12, we verify that our chosen $\rho_t$ in (10) satisfies the assumption of Lemma 12:

$$\frac{(\min_{k \in [K]} \hat{\omega}_k(t) - \eta_t)^2}{D^2} \geq \frac{1}{D^2} \left( \frac{1}{2\sqrt{t|\mathcal{X}_0|}} - \frac{1}{4\sqrt{t|\mathcal{X}_0|}} \right)^2 = \frac{1}{16tD^2|\mathcal{X}_0|} = \rho_t,$$

where the inequality is because of $\min_{k \in [K]} \hat{\omega}_k(t) \geq \frac{1}{2\sqrt{t|\mathcal{X}_0|}}$ (Lemma 14) and that $\eta_t = \frac{1}{4\sqrt{t|\mathcal{X}_0|}}$ in (10). Then, applying Lemma 12 with $v = \hat{\boldsymbol{\omega}}(t)$, $(v, \eta, n) = (\frac{1}{2\sqrt{t|\mathcal{X}_0|}}, \frac{1}{4\sqrt{t|\mathcal{X}_0|}}, \lceil t^{\frac{1}{4}} \rceil)$ yields that:

$$\mathbb{P}_{\boldsymbol{\mu}}[\mathcal{E}_{1,\epsilon}(T)^c] \leq \sum_{t=\underline{h}(T)}^{T} K \left( 2 \exp \left( -\frac{\epsilon^2 \sqrt{t}}{8\ell^2 K^3 D^2} \right) + \exp \left( -\sqrt{t} \right) \right) \leq \sum_{t=\underline{h}(T)}^{T} 3K \exp \left( -\sqrt{t} A_1(\epsilon) \right),$$

where $A_1(\epsilon) = \min\{1, \frac{\epsilon^2}{8\ell^2 K^3 D^2}\}$.

Bounding $\mathbb{P}_{\boldsymbol{\mu}}[\mathcal{E}_{2,\epsilon}(T)^c]$: As Lemma 14 provides a lower bound on the number of pulls, $\min_{k \in [K]} N_k(t) \geq \frac{1}{2} \sqrt{\frac{t}{|\mathcal{X}_0|}}$, for all arms, using this lower bound of $N_k(t)$ as the number of i.i.d. samples in the application of Chernoff bound leads to:

$$\mathbb{P}_{\boldsymbol{\mu}} \left[ |\hat{\mu}_k(t) - \mu_k| \geq \frac{\epsilon}{24D^3 \|\boldsymbol{\mu}\|_{\infty}} \right] \leq 2 \exp \left( -\sqrt{t} A_2(\epsilon) \right).$$

Hence, $\mathbb{P}_{\boldsymbol{\mu}}[\mathcal{E}_{2,\epsilon}(T)^c] \leq 2K \sum_{t=\underline{h}(T)}^{T} \exp \left( -\sqrt{t} A_2(\epsilon) \right)$. Then, we have

$$\sum_{T=M+1}^{\infty} \mathbb{P}_{\boldsymbol{\mu}}[(\mathcal{E}_{1,\epsilon}(T) \cap \mathcal{E}_{2,\epsilon}(T))^c] \leq \int_{M+1}^{\infty} \int_{T^a}^{\infty} \left( 3K e^{-\sqrt{t} A_1(\epsilon)} + 2K e^{-\sqrt{t} A_2(\epsilon)} \right) dt \, dT$$

$$\leq 2K \left( \frac{3}{A_1(\epsilon)^{2+\frac{2}{a}}} + \frac{2}{A_2(\epsilon)^{2+\frac{2}{a}}} \right) \Gamma \left( 2 + \frac{2}{a} \right),$$

where the second inequality uses Lemma 15. $\qquad \square$

## F.1 Lemmas for bounding $\mathbb{P}_{\boldsymbol{\mu}}[\mathcal{E}_{1,\epsilon}(T)^c]$

The following lemma is a result of two concentration inequalities, one bounds how much the empirical average deviates from the expectation (Proposition 3), and the other bounds the error incurred by MCP (Lemma 13).

**Lemma 12.** *Let* $(\boldsymbol{\pi}, \boldsymbol{\omega}, \theta) \in \Lambda \times \Sigma_+ \times (0,1)$, $v \in (0, \min_{k \in [K]} \omega_k)$, *and* $\eta \in (0, v)$. *Then,* $\forall \epsilon \in (0, 4K(v-\eta)/D)$,

$$\mathbb{P}\left[ \left\langle \nabla \bar{F}_{\boldsymbol{\pi},\eta}(\boldsymbol{\omega}), \tilde{\boldsymbol{x}}^\star - \boldsymbol{\omega} \right\rangle \geq \max_{\boldsymbol{x} \in \mathcal{X}} \left\langle \nabla \bar{F}_{\boldsymbol{\pi},\eta}(\boldsymbol{\omega}), \boldsymbol{x} - \boldsymbol{\omega} \right\rangle - \epsilon \right] \geq 1 - K\left( 2\exp\left(-\frac{\epsilon^2 n^2}{8\ell^2 K^3 D^2}\right) + n\theta \right),$$

*where* $\nabla \tilde{F}_{\boldsymbol{\pi},\eta,n}(\boldsymbol{\omega})$ *is computed by* $\left(\frac{(v-\eta)^2}{D^2}, \theta\right)$*-MCP, and* $\tilde{\boldsymbol{x}}^\star \in \operatorname{argmax}_{\boldsymbol{x} \in \mathcal{X}} \left\langle \nabla \tilde{F}_{\boldsymbol{\pi},\eta,n}(\boldsymbol{\omega}), \boldsymbol{x} \right\rangle$.

**Proof** Let $\boldsymbol{x}^\star \in \operatorname{argmax}_{\boldsymbol{x} \in \mathcal{X}} \left\langle \nabla \bar{F}_{\boldsymbol{\pi},\eta}(\boldsymbol{\omega}), \boldsymbol{x} \right\rangle$. From $\tilde{\boldsymbol{x}}^\star \in \operatorname{argmax}_{\boldsymbol{x} \in \mathcal{X}} \left\langle \nabla \tilde{F}_{\boldsymbol{\pi},\eta,n}(\boldsymbol{\omega}), \boldsymbol{x} \right\rangle$,

$$\left\langle \nabla \bar{F}_{\boldsymbol{\pi},\eta}(\boldsymbol{\omega}), \boldsymbol{x}^\star - \tilde{\boldsymbol{x}}^\star \right\rangle \leq \left\langle \nabla \bar{F}_{\boldsymbol{\pi},\eta}(\boldsymbol{\omega}), \boldsymbol{x}^\star - \tilde{\boldsymbol{x}}^\star \right\rangle + \left\langle \nabla \tilde{F}_{\boldsymbol{\pi},\eta}(\boldsymbol{\omega}), \tilde{\boldsymbol{x}}^\star - \boldsymbol{x}^\star \right\rangle$$

$$= \left\langle \nabla \bar{F}_{\boldsymbol{\pi},\eta}(\boldsymbol{\omega}) - \nabla \tilde{F}_{\boldsymbol{\pi},\eta,n}(\boldsymbol{\omega}), \boldsymbol{x}^\star - \tilde{\boldsymbol{x}}^\star \right\rangle.$$

Fix $\epsilon > 0$. Recall that $\nabla \tilde{F}_{\boldsymbol{\pi},\eta,n}(\boldsymbol{\omega}) = \frac{1}{n} \sum_{m=1}^n \nabla_{\boldsymbol{\omega}} f_{\hat{\boldsymbol{x}}_m}(\boldsymbol{\omega} + \eta \boldsymbol{Z}_m, \boldsymbol{\pi})$ where each $\hat{\boldsymbol{x}}_m$ is computed by $\left(\frac{(v-\eta)^2}{D^2}, \theta\right)$-MCP$(\boldsymbol{\omega} + \eta \boldsymbol{Z}_m, \boldsymbol{\pi})$, and each $\boldsymbol{Z}_m$ is independently sampled from Uniform$(B_2)$. Now, consider any fixed $\boldsymbol{x} = \boldsymbol{e}_k$ for any $k \in [K]$. Invoking Proposition 3 with $\epsilon = \frac{\epsilon}{4K}$ and $\boldsymbol{x} = \boldsymbol{e}_k$, we get:

$$\mathbb{P}\left[ \left| \left\langle \nabla \bar{F}_{\boldsymbol{\pi},\eta}(\boldsymbol{\omega}) - \frac{1}{n} \sum_{m=1}^n \nabla F_{\boldsymbol{\pi}}(\boldsymbol{\omega} + \eta \boldsymbol{Z}_m), \boldsymbol{e}_k \right\rangle \right| \geq \frac{\epsilon}{4K} \right] \leq 2\exp\left(-\frac{\epsilon^2 n^2}{8\ell^2 K^3 D^2}\right).$$

Also, for $\nabla \tilde{F}_{\boldsymbol{\pi},\eta,n}(\boldsymbol{\omega})$ computed by the $((v-\eta)^2/D^2, \theta)$-MCP algorithm, Lemma 13 with $\boldsymbol{x} = \boldsymbol{e}_k$, and $\theta = \theta$ and the assumption that $\epsilon \in (0, 4K(v-\eta)/D)$ implies that:

$$\mathbb{P}\left[ \left| \left\langle \frac{1}{n} \sum_{m=1}^n \nabla F_{\boldsymbol{\pi}}(\boldsymbol{\omega} + \eta \boldsymbol{Z}_m) - \nabla \tilde{F}_{\boldsymbol{\pi},\eta,n}(\boldsymbol{\omega}), \boldsymbol{e}_k \right\rangle \right| \geq \frac{\epsilon}{4K} \right] \leq n\theta.$$

Combining the two inequalities leads to:

$$\mathbb{P}\left[ \left| \left\langle \nabla \bar{F}_{\boldsymbol{\pi},\eta}(\boldsymbol{\omega}) - \nabla \tilde{F}_{\boldsymbol{\pi},\eta,n}(\boldsymbol{\omega}), \boldsymbol{e}_k \right\rangle \right| \leq \frac{\epsilon}{2K} \right] \geq 1 - \left( 2\exp\left(-\frac{\epsilon^2 n^2}{8\ell^2 K^3 D^2}\right) + n\theta \right).$$

Then, an application of a union bound over all $\{\boldsymbol{e}_k\}_{k \in [K]}$ gives

$$\mathbb{P}\left[ \left\langle \nabla \bar{F}_{\boldsymbol{\pi},\eta}(\boldsymbol{\omega}) - \nabla \tilde{F}_{\boldsymbol{\pi},\eta,n}(\boldsymbol{\omega}), \boldsymbol{x}^\star - \tilde{\boldsymbol{x}}^\star \right\rangle \leq \epsilon \right] \geq 1 - K\left( 2\exp\left(-\frac{\epsilon^2 n^2}{8\ell^2 K^3 D^2}\right) + n\theta \right). \quad (46)$$

Observe $\left\langle -\nabla \tilde{F}_{\boldsymbol{\pi},\eta,n}(\boldsymbol{\omega}), \boldsymbol{x}^\star - \tilde{\boldsymbol{x}}^\star \right\rangle \geq 0$ implies

$$\left\{ \left\langle \nabla \bar{F}_{\boldsymbol{\pi},\eta}(\boldsymbol{\omega}) - \nabla \tilde{F}_{\boldsymbol{\pi},\eta,n}(\boldsymbol{\omega}), \boldsymbol{x}^\star - \tilde{\boldsymbol{x}}^\star \right\rangle \leq \epsilon \right\} \subseteq \left\{ \left\langle \nabla \bar{F}_{\boldsymbol{\pi},\eta}(\boldsymbol{\omega}), \boldsymbol{x}^\star - \tilde{\boldsymbol{x}}^\star \right\rangle \leq \epsilon \right\}. \quad (47)$$

From (46)-(47), we conclude that the r.h.s. of (47) happens with probability at least $1 - K\left(2\exp\left(-\frac{\epsilon^2 n^2}{8\ell^2 K^3 D^2}\right) + n\theta\right)$. The proof is completed by simply rearranging the r.h.s. of (47). $\square$

**Lemma 13.** *Let* $(\boldsymbol{\pi}, \boldsymbol{\omega}, \boldsymbol{x}, \theta) \in \Lambda \times \Sigma_+ \times \{0,1\}^K \times (0,1)$ *with* $\|\boldsymbol{x}\|_1 \leq D$ *and* $v \in (0, \min_{k \in [K]} \omega_k)$.

$$\forall (\eta, \boldsymbol{z}) \in (0, v) \times B_2, \quad \mathbb{P}\left[ |\langle \nabla_{\boldsymbol{\omega}} f_{\boldsymbol{x}_\star}(\boldsymbol{\omega} + \eta \boldsymbol{z}) - \nabla_{\boldsymbol{\omega}} f_{\hat{\boldsymbol{x}}}(\boldsymbol{\omega} + \eta \boldsymbol{z}), \boldsymbol{x}\rangle| \leq \frac{v-\eta}{D} \right] \geq 1 - \theta,$$

*where* $\boldsymbol{x}_\star$ *is some action satisfying* $f_{\boldsymbol{x}_\star}(\boldsymbol{\omega} + \eta \boldsymbol{z}) = F_{\boldsymbol{\pi}}(\boldsymbol{\omega} + \eta \boldsymbol{z})$, *and* $\hat{\boldsymbol{x}}$ *is the returned action of* $((v-\eta)^2/D^2, \theta)$*-MCP*$(\boldsymbol{\omega} + \eta \boldsymbol{z}, \boldsymbol{\pi})$.

**Proof** This basically follows from a direct calculation. Let $\epsilon > 0$ and fix any $(\boldsymbol{\pi}, \boldsymbol{\omega}, \boldsymbol{x}) \in \Lambda \times \Sigma_+ \times \{0,1\}^K$, $\|\boldsymbol{x}\|_1 \leq D$, and any $(\eta, \boldsymbol{z}) \in (0, v) \times B_2$. Then, for $\hat{\boldsymbol{x}}$ computed by $(\rho, \theta)$-MCP$(\boldsymbol{\omega} + \eta \boldsymbol{z}, \boldsymbol{\pi})$ with $\rho = (v - \eta)^2 / D^2$, we have with probability at least $1 - \theta$

$$\rho \geq |\langle \nabla_{\boldsymbol{\omega}} f_{\boldsymbol{x}_\star}(\boldsymbol{\omega} + \eta \boldsymbol{z}) - \nabla_{\boldsymbol{\omega}} f_{\hat{\boldsymbol{x}}}(\boldsymbol{\omega} + \eta \boldsymbol{z}), \boldsymbol{\omega} + \eta \boldsymbol{z} \rangle|$$
$$\geq \min_{k \in [K]} (\boldsymbol{\omega} + \eta \boldsymbol{z})_k \|\nabla_{\boldsymbol{\omega}} f_{\boldsymbol{x}_\star}(\boldsymbol{\omega} + \eta \boldsymbol{z}) - \nabla_{\boldsymbol{\omega}} f_{\hat{\boldsymbol{x}}}(\boldsymbol{\omega} + \eta \boldsymbol{z})\|_\infty.$$

Hence, remarking that $\min_{k \in [K]} (\boldsymbol{\omega} + \eta \boldsymbol{z})_k \geq v - \eta > 0$, we get: with probability at least $1 - \theta$,

$$|\langle \nabla_{\boldsymbol{\omega}} f_{\boldsymbol{x}_\star}(\boldsymbol{\omega} + \eta \boldsymbol{z}) - \nabla_{\boldsymbol{\omega}} f_{\hat{\boldsymbol{x}}}(\boldsymbol{\omega} + \eta \boldsymbol{z}), \boldsymbol{x} \rangle| \leq \frac{\rho D}{v - \eta} = \frac{v - \eta}{D},$$

where we used the fact that $\|\boldsymbol{x}\|_1 \leq D$ and Hölder's inequality. $\qquad\square$

**Proposition 3.** *Let $(\boldsymbol{\pi}, \boldsymbol{\omega}, \boldsymbol{x}) \in \Lambda \times \Sigma_+ \times \{0,1\}^K$, $\eta \in (0, \min_{k \in [K]} \omega_k)$, and $\|\boldsymbol{x}\|_1 \leq D$. Then,*

$$\forall \epsilon > 0, \quad \mathbb{P}\left[\left|\left\langle \nabla \bar{F}_{\boldsymbol{\pi}, \eta}(\boldsymbol{\omega}) - \frac{1}{n} \sum_{m=1}^n \nabla F_{\boldsymbol{\pi}}(\boldsymbol{\omega} + \eta \boldsymbol{\mathcal{Z}}_m), \boldsymbol{x} \right\rangle\right| \geq \epsilon\right] \leq 2 \exp\left(-\frac{2\epsilon^2 n^2}{\ell^2 K D^2}\right),$$

*where $\boldsymbol{\mathcal{Z}}_1, \cdots, \boldsymbol{\mathcal{Z}}_n$ are independently sampled from Uniform$(B_2)$.*

**Proof** Fix $(\boldsymbol{\pi}, \boldsymbol{\omega}, \boldsymbol{x}) \in \Lambda \times \Sigma_+ \times \{0,1\}^K$ where $\|\boldsymbol{x}\|_1 \leq D$, and fix $\eta \in (0, \min_{k \in [K]} \omega_k)$. Define

$$\phi(\boldsymbol{z}_1, \cdots, \boldsymbol{z}_t) = \left\langle \nabla \bar{F}_{\boldsymbol{\pi}, \eta}(\boldsymbol{\omega}) - \frac{1}{n} \sum_{m=1}^n \nabla F_{\boldsymbol{\pi}}(\boldsymbol{\omega} + \eta \boldsymbol{z}_m), \boldsymbol{x} \right\rangle.$$

Note that $\mathbb{E}_{\boldsymbol{\mathcal{Z}}_1, \cdots, \boldsymbol{\mathcal{Z}}_n}[\phi(\boldsymbol{\mathcal{Z}}_1, \cdots, \boldsymbol{\mathcal{Z}}_n)] = 0$ by definition. Now we also observe that:

$$\max_{\boldsymbol{z}_1, \cdots, \boldsymbol{z}_n, \boldsymbol{z}' \in B_2, m \in [n]} |\phi(\boldsymbol{z}_1, \cdots, \boldsymbol{z}_n) - \phi(\boldsymbol{z}_1, \cdots, \boldsymbol{z}_{m-1}, \boldsymbol{z}', \boldsymbol{z}_{m+1}, \cdots, \boldsymbol{z}_n)| \leq \frac{\ell D}{n}$$

due to the $\ell$-Lipschitzness of $F_{\hat{\boldsymbol{\mu}}}$ (Lemma 21 in Appendix I) and $\max_{\boldsymbol{x} \in \mathcal{X}} \|\boldsymbol{x}\|_1 \leq D$. Hence it follows from McDiarmid's inequality (Lemma 16 in F.3) that

$$\forall \epsilon > 0, \quad \mathbb{P}[|\phi(\boldsymbol{\mathcal{Z}}_1, \cdots, \boldsymbol{\mathcal{Z}}_n)| \geq \epsilon] \leq 2 \exp\left(-\frac{2\epsilon^2}{K(\frac{\ell D}{n})^2}\right) = 2 \exp\left(-\frac{2\epsilon^2 n^2}{\ell^2 K D^2}\right).$$

$\qquad\square$

### F.2 Lemmas for bounding $\mathbb{P}_{\boldsymbol{\mu}}[\mathcal{E}_{2,\epsilon}(T)^c]$

**Lemma 14** (forced exploration)**.** *Let $\mathcal{X}_0 \subseteq \mathcal{X}$ be any set covering all arms $[K]$ and $t \geq 4|\mathcal{X}_0|$. Any algorithm with forced-exploration procedure satisfies*

$$\hat{\boldsymbol{\omega}}(t) \in \Sigma_{\sqrt{\frac{1}{t|\mathcal{X}_0|}} - \frac{1}{t}} \subset \Sigma_{\frac{1}{2}\sqrt{\frac{1}{t|\mathcal{X}_0|}}}, \quad \forall t \geq 4|\mathcal{X}_0|.$$

**Proof** Fix any $k \in [K]$. By merely counting the rounds before $t$ performing forced exploration,

$$N_k(t) \geq \sum_{s \in [t] : \lfloor \sqrt{\frac{s}{|\mathcal{X}_0|}} \rfloor \in \mathbb{N}} \sum_{\boldsymbol{x} \in \mathcal{X}_0} x_k \geq \sqrt{\frac{t}{|\mathcal{X}_0|}} - 1 \geq \frac{1}{2} \sqrt{\frac{t}{|\mathcal{X}_0|}},$$

where the last inequality holds for any $t \geq 4|\mathcal{X}_0|$. $\qquad\square$

### F.3 Technical lemmas

**Lemma 15** ([WTP21])**.** *Let $\alpha \in (0,1)$ and $A, \beta > 0$. Then,*

$$\int_0^\infty \left(\int_{T^\alpha}^\infty e^{-At^\beta} dt\right) dT = \frac{\Gamma(\frac{1}{\alpha\beta} + \frac{1}{\beta})}{\beta A^{\frac{1}{\alpha\beta} + \frac{1}{\beta}}}.$$

**Proof** The result of Lemma 5 [WTP21] is stated for $\alpha, \beta \in (0,1)$ but it actually applies for the case of $\beta > 0$ as well. Here we provide a proof for completeness.

$$\int_0^\infty \left( \int_{T^\alpha}^\infty e^{-At^\beta} dt \right) dT = \int_0^\infty \alpha T^\alpha e^{-AT^{\alpha\beta}} dT = \frac{1}{\beta} \int_0^\infty x^{\frac{1}{\alpha\beta} + \frac{1}{\beta} - 1} e^{-Ax} dx = \frac{\Gamma(\frac{1}{\alpha\beta} + \frac{1}{\beta})}{\beta A^{\frac{1}{\alpha\beta} + \frac{1}{\beta}}}.$$

$\square$

The below Lemma 16, also known as bounded different inequality, can be found in many textbooks, e.g., Theorem 6.2 in [BLM13].

**Lemma 16** (McDiarmid's inequality). *Let $\boldsymbol{\mathcal{Z}} = (\mathcal{Z}_1, \cdots, \mathcal{Z}_n)$ be independent random variables, and $\phi : \mathbb{R}^n \mapsto \mathbb{R}$ be a measurable function. Suppose $\phi(\mathbf{z})$ changes by at most $c_i > 0$ under an arbitrary change of the $i$-th coordinate. Then,*

$$\forall \epsilon > 0, \quad \mathbb{P}[\phi(\boldsymbol{\mathcal{Z}}) - \mathbb{E}[\phi(\boldsymbol{\mathcal{Z}})] \geq \epsilon] \leq \exp\left( -\frac{2\epsilon^2}{\sum_{i=1}^n c_i^2} \right).$$

# G Continuity arguments

In this section, we establish the continuity of $F_{\boldsymbol{\pi}}(\boldsymbol{\omega})$ and $\nabla \bar{F}_{\boldsymbol{\pi},\eta}(\boldsymbol{\omega})$ in $\boldsymbol{\pi}$ for any fixed $\boldsymbol{\omega} \in \Sigma_+$, where $\nabla \bar{F}_{\boldsymbol{\pi},\eta}(\boldsymbol{\omega})$ denotes the gradient $\nabla_{\boldsymbol{\omega}} \bar{F}_{\boldsymbol{\pi},\eta}(\boldsymbol{\omega})$ taken w.r.t. the input $\boldsymbol{\omega}$. As the consequence of the continuity of $F_{\boldsymbol{\pi}}$ and $\nabla \bar{F}_{\boldsymbol{\pi},\eta}$ in $\boldsymbol{\pi}$, we can show the point-wise convergence of $F_{\hat{\boldsymbol{\mu}}(t)} \to F_{\boldsymbol{\mu}}$ and $\nabla \bar{F}_{\hat{\boldsymbol{\mu}}(t),\eta} \to \nabla \bar{F}_{\boldsymbol{\mu},\eta}$ given that $\hat{\boldsymbol{\mu}}(t) \to \boldsymbol{\mu}$ almost surely.

**Notation.** Throughout this section, we define $\nabla \bar{F}_{\boldsymbol{\pi},\eta}(\boldsymbol{\omega}) = \mathbf{0}_K$ if $\eta \geq \min_{k \in [K]} \omega_k$ for any $(\boldsymbol{\pi}, \boldsymbol{\omega}) \in \Lambda \times \Sigma_+$. Moreover, for any $(\boldsymbol{v}, \boldsymbol{\omega}) \in \mathbb{R}^K \times \Sigma_+$, we will use $\nabla_{\boldsymbol{\pi}} F_{\boldsymbol{v}}(\boldsymbol{\omega})$ (resp. $\nabla_{\boldsymbol{\pi}} (\frac{\partial \bar{F}_{\boldsymbol{v},\eta}(\boldsymbol{\omega})}{\partial \omega_k})$) to denote the gradient of the function $\boldsymbol{\pi} \mapsto F_{\boldsymbol{\pi}}(\boldsymbol{\omega})$ (resp. $\boldsymbol{\pi} \mapsto \frac{\partial \bar{F}_{\boldsymbol{\pi},\eta}(\boldsymbol{\omega})}{\partial \omega_k}$) evaluated at the point $\boldsymbol{v}$.

The main result in this section, Lemma 7, is derived based on Lemma 17 in Appendix G.1 (which asserts the continuity of the function $\psi_{\boldsymbol{\omega},\boldsymbol{x},\eta}(\boldsymbol{\pi}) = \langle \nabla \bar{F}_{\boldsymbol{\pi},\eta}(\boldsymbol{\omega}), \boldsymbol{x} - \boldsymbol{\omega} \rangle$ on $\mathbb{R}^K$) and Proposition 4 in Appendix G.2 (which upper bounds the length of $\nabla f_{\boldsymbol{x}}(\boldsymbol{\omega}, \boldsymbol{\mu})$).

**Lemma 7.** *Let* $\boldsymbol{\mu} \in \Lambda$ *and* $\epsilon \in (0, \frac{2D^2 \triangle_{\min}(\boldsymbol{\mu})^2}{K})$. *Then, any* $\boldsymbol{\pi} \in \mathbb{R}^K$ *with* $\|\boldsymbol{\pi} - \boldsymbol{\mu}\|_\infty < \frac{\epsilon}{24D^3 \|\boldsymbol{\mu}\|_\infty}$ *satisfies the following:*

$$|F_{\boldsymbol{\mu}}(\boldsymbol{\omega}) - F_{\boldsymbol{\pi}}(\boldsymbol{\omega})| < \epsilon, \quad \forall \boldsymbol{\omega} \in \Sigma_+ \tag{35}$$

$$|\langle \nabla \bar{F}_{\boldsymbol{\pi},\eta}(\boldsymbol{\omega}) - \nabla \bar{F}_{\boldsymbol{\mu},\eta}(\boldsymbol{\omega}), \boldsymbol{x} - \boldsymbol{\omega} \rangle| < \epsilon, \quad \forall (\boldsymbol{\omega}, \boldsymbol{x}) \in \Sigma_+ \times \mathcal{X}, \forall \eta \in (0, \min_{k \in [K]} \omega_k). \tag{36}$$

**Proof** Inspired by Lemma 14 in [WTP21], we prove this lemma using Proposition 4 and applying the mean-value theorem to $\psi_{\boldsymbol{\omega},\boldsymbol{x},\eta}$.

Fix $(\boldsymbol{\omega}, \boldsymbol{\mu}) \in \Sigma_+ \times \Lambda$, and let $\boldsymbol{i}^\star = \boldsymbol{i}^\star(\boldsymbol{\mu})$ and $\triangle_{\boldsymbol{x}} = \triangle_{\boldsymbol{x}}(\boldsymbol{\mu})$ for any $\boldsymbol{x} \in \mathcal{X} \backslash \{\boldsymbol{i}^\star\}$. Fix $\epsilon \in (0, \frac{2D^2 \triangle_{\min}^2}{K})$ and $\boldsymbol{\pi} \in \mathbb{R}^K$ such that $\|\boldsymbol{\pi} - \boldsymbol{\mu}\|_\infty < \frac{\epsilon}{24D^3 \|\boldsymbol{\mu}\|_\infty}$. One may check that this $\boldsymbol{\pi}$ satisfies the assumption of Proposition 4 as

$$\|\boldsymbol{\pi} - \boldsymbol{\mu}\|_\infty < \frac{\epsilon}{24D^3 \|\boldsymbol{\mu}\|_\infty} < \frac{2D^2 \triangle_{\min}^2}{24KD^3 \|\boldsymbol{\mu}\|_\infty} = \frac{\triangle_{\min}^2}{12KD \|\boldsymbol{\mu}\|_\infty} \leq \frac{\triangle_{\min}}{6K} < \frac{\triangle_{\min}}{\sqrt{2KD}},$$

where the second inequality stems from the choice of $\epsilon$ and the second last is because $\triangle_{\min} \leq 2D \|\boldsymbol{\mu}\|_\infty$. In what follows, we will be applying the mean-value theorem to $\psi_{\boldsymbol{\omega},\boldsymbol{x},\eta}$ (whose continuity is stated in Lemma 17). For convenience, introduce the function $\boldsymbol{r}(\beta) = (1 - \beta)\boldsymbol{\mu} + \beta \boldsymbol{\pi}$ for any $\beta \in (0, 1)$.

Proof of (35): For any $\boldsymbol{x} \in \mathcal{X} \backslash \{\boldsymbol{i}^\star\}$, by the mean-value theorem, there exists a $\beta \in (0, 1)$ such that

$$|f_{\boldsymbol{x}}(\boldsymbol{\omega}, \boldsymbol{\pi}) - f_{\boldsymbol{x}}(\boldsymbol{\omega}, \boldsymbol{\mu})| = |\langle \nabla_{\boldsymbol{\pi}} f_{\boldsymbol{x}}(\boldsymbol{\omega}, \boldsymbol{r}(\beta)), \boldsymbol{\pi} - \boldsymbol{\mu} \rangle|$$

$$= \left| \sum_{k \in [K]} \omega_k \left\langle \nabla_{\boldsymbol{\pi}} \left( \frac{\partial f_{\boldsymbol{x}}(\boldsymbol{\omega}, \boldsymbol{r}(\beta))}{\partial \omega_k} \right), \boldsymbol{\pi} - \boldsymbol{\mu} \right\rangle \right|$$

$$\leq \sum_{k \in [K]} \omega_k \left\| \nabla_{\boldsymbol{\pi}} \left( \frac{\partial f_{\boldsymbol{x}}(\boldsymbol{\omega}, \boldsymbol{r}(\beta))}{\partial \omega_k} \right) \right\|_1 \|\boldsymbol{\pi} - \boldsymbol{\mu}\|_\infty < \epsilon, \tag{48}$$

where the last inequality uses $\boldsymbol{\omega} \in \Sigma_+$, $\|\boldsymbol{\pi} - \boldsymbol{\mu}\|_\infty < \frac{\epsilon}{24D^3 \|\boldsymbol{\mu}\|_\infty}$, $\left\| \nabla_{\boldsymbol{\pi}} \left( \frac{\partial f_{\boldsymbol{x}}(\boldsymbol{\omega}, \boldsymbol{r}(\beta))}{\partial \omega_k} \right) \right\|_1 \leq 12D^2 \|\boldsymbol{\mu}\|_\infty$ (Proposition 4). Hence, from a substitution of $\boldsymbol{x}$ in (48) with $\boldsymbol{x}_e \in \arg\min_{\boldsymbol{x} \neq \boldsymbol{i}^\star} f_{\boldsymbol{x}}(\boldsymbol{\omega}, \boldsymbol{\mu})$ and the fact that $F_{\boldsymbol{\pi}}(\boldsymbol{\omega}) \leq f_{\boldsymbol{x}_e}(\boldsymbol{\omega}, \boldsymbol{\pi})$, we derive

$$F_{\boldsymbol{\pi}}(\boldsymbol{\omega}) - F_{\boldsymbol{\mu}}(\boldsymbol{\omega}) \leq f_{\boldsymbol{x}_e}(\boldsymbol{\omega}, \boldsymbol{\pi}) - f_{\boldsymbol{x}_e}(\boldsymbol{\omega}, \boldsymbol{\mu}) < \epsilon.$$

The other inequality of $F_{\boldsymbol{\mu}}(\boldsymbol{\omega}) - F_{\boldsymbol{\pi}}(\boldsymbol{\omega}) < \epsilon$ can be derived similarly. This proves (35).

Proof of (36): Recall that $\psi_{\boldsymbol{\omega},\boldsymbol{x},\eta}(\boldsymbol{\pi}) = \langle \nabla \bar{F}_{\boldsymbol{\pi},\eta}(\boldsymbol{\omega}), \boldsymbol{x} - \boldsymbol{\omega} \rangle$ is continuous on $\mathbb{R}^K$ (Lemma 17). By the mean-value theorem, there exists $\beta \in (0, 1)$ such that

$$|\psi_{\boldsymbol{\omega},\boldsymbol{x},\eta}(\boldsymbol{\pi}) - \psi_{\boldsymbol{\omega},\boldsymbol{x},\eta}(\boldsymbol{\mu})| = |\langle \nabla_{\boldsymbol{\pi}} \psi_{\boldsymbol{\omega},\boldsymbol{x},\eta}(\boldsymbol{r}(\beta)), \boldsymbol{\pi} - \boldsymbol{\mu} \rangle|$$

$$\leq \|\nabla_{\boldsymbol{\pi}} \psi_{\boldsymbol{\omega},\boldsymbol{x},\eta}(\boldsymbol{r}(\beta))\|_1 \|\boldsymbol{\pi} - \boldsymbol{\mu}\|_\infty. \tag{49}$$

To bound $\|\nabla_{\boldsymbol{\pi}}\psi_{\boldsymbol{\omega},\boldsymbol{x},\eta}(\boldsymbol{r}(\beta))\|_1$, we write

$$\nabla_{\boldsymbol{\pi}}\psi_{\boldsymbol{\omega},\boldsymbol{x},\eta}(\boldsymbol{r}(\beta)) = \sum_{k\in[K]} \nabla_{\boldsymbol{\pi}}\left(\frac{\partial\bar{F}_{\boldsymbol{r}(\beta),\eta}(\boldsymbol{\omega})}{\partial\omega_k}\right)(x_k-\omega_k).$$

Then it follows from the fundamental theorem of calculus that: the gradient $\nabla_{\boldsymbol{\pi}}$ and the expectation operators are exchangeable, i.e.,

$$\forall k\in[K],\quad \nabla_{\boldsymbol{\pi}}\left(\frac{\partial\bar{F}_{\boldsymbol{r}(\beta),\eta}(\boldsymbol{\omega})}{\partial\omega_k}\right) = \mathbb{E}_{\boldsymbol{\mathcal{Z}}\sim\mathrm{Uniform}(B_2)}\left[\nabla_{\boldsymbol{\pi}}\left(\frac{\partial F_{\boldsymbol{r}(\beta)}(\boldsymbol{\omega}+\eta\boldsymbol{\mathcal{Z}})}{\partial\omega_k}\right)\right].$$

As shown in Appendix H, $\frac{\partial F_{\boldsymbol{r}(\beta)}(\boldsymbol{\omega}+\eta\boldsymbol{\mathcal{Z}})}{\partial\omega_k}$ exists almost surely. When such gradient exists, Proposition 4 bounds its 1-norm length by

$$\left\|\nabla_{\boldsymbol{\pi}}\left(\frac{\partial F_{\boldsymbol{r}(\beta)}(\boldsymbol{\omega}+\eta\boldsymbol{\mathcal{Z}})}{\partial\omega_k}\right)\right\|_1 \leq 12D^2\|\boldsymbol{\mu}\|_\infty,$$

so it follows that $\left\|\nabla_{\boldsymbol{\pi}}\left(\frac{\partial\bar{F}_{\boldsymbol{r}(\beta),\eta}(\boldsymbol{\omega})}{\partial\omega_k}\right)\right\|_1 \leq 12D^2\|\boldsymbol{\mu}\|_\infty$ as well. Hence, substituting the above back to $\nabla_{\boldsymbol{\pi}}\psi_{\boldsymbol{\omega},\boldsymbol{x},\eta}(\boldsymbol{r}(\beta))$ yields:

$$\|\nabla_{\boldsymbol{\pi}}\psi_{\boldsymbol{\omega},\boldsymbol{x},\eta}(\boldsymbol{r}(\beta))\|_1 \leq \max_{k\in[K]}\left\|\nabla_{\boldsymbol{\pi}}\left(\frac{\partial\bar{F}_{\boldsymbol{r}(\beta),\eta}(\boldsymbol{\omega})}{\partial\omega_k}\right)\right\|_1 \|\boldsymbol{x}-\boldsymbol{\omega}\|_1 \leq 24D^3\|\boldsymbol{\mu}\|_\infty,$$

where the first inequality use Hölder's inequality. Finally, plugging the above into (49) and recalling that $\|\boldsymbol{\pi}-\boldsymbol{\mu}\|_\infty < \frac{\epsilon}{24D^3\|\boldsymbol{\mu}\|_\infty}$, we have

$$|\psi_{\boldsymbol{\omega},\boldsymbol{x},\eta}(\boldsymbol{\pi}) - \psi_{\boldsymbol{\omega},\boldsymbol{x},\eta}(\boldsymbol{\mu})| < \epsilon.$$

This concludes the proof. $\qquad\square$

### G.1 An application of the maximum theorem

Recall that $\psi_{\boldsymbol{\omega},\boldsymbol{x},\eta}(\boldsymbol{\pi}) = \langle\nabla\bar{F}_{\boldsymbol{\pi},\eta}(\boldsymbol{\omega}),\boldsymbol{x}-\boldsymbol{\omega}\rangle$.

**Lemma 17.** *For any $\epsilon > 0$, there exists a constant $\xi_\epsilon > 0$ such that if $\|\boldsymbol{\pi}-\boldsymbol{\mu}\|_\infty < \xi_\epsilon$, then*

$$|\psi_{\boldsymbol{\omega},\boldsymbol{x},\eta}(\boldsymbol{\pi}) - \psi_{\boldsymbol{\omega},\boldsymbol{x},\eta}(\boldsymbol{\mu})| < \epsilon,\quad \forall(\boldsymbol{\omega},\boldsymbol{x})\in\Sigma_+\times\mathcal{X},\ \forall\eta\in(0,\min_{k\in[K]}\omega_k). \tag{50}$$

The proof of Lemma 17 replies on the celebrated maximum theorem [FKV14], which is introduced below. After that, we then show its proof.

**Maximum Theorem:** Here we briefly introduce the maximum theorem and Lemma 17 will be proved at the end of this section. The definitions and results are taken from [FKV14] (see also Appendix K.1 of [WTP21]).

**Definition 1.** *Let $U \neq \emptyset$ be a subset of a topological space and $h: U \mapsto \mathbb{R}$ be a function. Define the level sets of $h$ for $y \in \mathbb{R}$ as*

$$L_h(y,U) = \{x\in U: h(x)\leq y\} \quad and \quad L_h^<(y,U) = \{x\in U: h(x)<y\}.$$

*The function $h$ is said to be lower semi-continuous (resp. upper semi-continuous) on $U$ if $L_h(y,U)$ are closed (resp. $L_h^<(y,U)$ are open) for all $y\in\mathbb{R}$; $h$ is said to be inf-compact on $U$ if $L_h(y,U)$ and $L_h^<(y,U)$ are compact for all $y\in\mathbb{R}$.*

**Definition 2.** *Let $\mathbb{X}$ and $\mathbb{Y}$ be Hausdorff topological spaces and $\Phi: \mathbb{X}\rightrightarrows\mathbb{S}(\mathbb{Y})$ be a set-valued function, where $\mathbb{S}(\mathbb{Y})$ is the set of non-empty subsets of $\mathbb{Y}$. Define*

$$Gr_U(\Phi) = \{(x,y)\in U\times\mathbb{Y}: y\in\Phi(x)\}$$

*as the graph of $\Phi$ restricted to $U$. The function $u: \mathbb{X}\times\mathbb{Y}\mapsto\mathbb{R}$ is said to be $\mathbb{K}$-inf-compact on $Gr_{\mathbb{X}}(\Phi)$ if for all non-empty compact subset $C$ of $\mathbb{X}$, $u$ is inf-compact on $Gr_C(\Phi)$.*

**Theorem 6** (Maximum theorem). *Suppose $\mathbb{X}$ is compactly generated, $\Phi: \mathbb{X}\rightrightarrows\mathbb{S}(\mathbb{Y})$ is lower hemicontinuous, and $u: \mathbb{X}\times\mathbb{Y}\mapsto\mathbb{R}$ is $\mathbb{K}$-inf-compact and upper semi-continuous on $Gr_{\mathbb{X}}(\Phi)$. Then, the function $v(x) = \inf_{y\in\Phi(x)}u(x,y)$ is continuous and the set of its optimal solutions $\Phi^\star(x) = \{y\in\Phi(x): u(x,y)=v(x)\}$ is upper hemicontinuous and compact-valued.*

**Proof of Lemma 17:** Fix any $\boldsymbol{\mu} \in \Lambda$ and let $\boldsymbol{i}^\star = \boldsymbol{i}^\star(\boldsymbol{\mu})$. The goal is to show that for any $\epsilon > 0$, there exists a constant $\xi_\epsilon > 0$ such that if $\|\boldsymbol{\pi} - \boldsymbol{\mu}\|_\infty < \xi_\epsilon$, then

$$|\psi_{\boldsymbol{\omega},\boldsymbol{x},\eta}(\boldsymbol{\pi}) - \psi_{\boldsymbol{\omega},\boldsymbol{x},\eta}(\boldsymbol{\mu})| < \epsilon, \quad \forall (\boldsymbol{\omega},\boldsymbol{x}) \in \Sigma_+ \times \mathcal{X}, \ \forall \eta \in (0, \min_{k \in [K]} \omega_k), \tag{50}$$

where $\psi_{\boldsymbol{\omega},\boldsymbol{x},\eta}(\boldsymbol{\pi}) = \langle \nabla \bar{F}_{\boldsymbol{\pi},\eta}(\boldsymbol{\omega}), \boldsymbol{x} - \boldsymbol{\omega} \rangle$. In what follows, we will use $p$ to denote the probability distribution of Uniform$(B_2)$. We will first show that $\psi_{\boldsymbol{\omega},\boldsymbol{x},\eta}$ is continuous for each fixed $(\boldsymbol{\omega},\boldsymbol{x},\eta) \in \Sigma_+ \times \mathcal{X} \times (0,1)$, and then use Theorem 6 to show (50).

Continuity of $\psi_{\boldsymbol{\omega},\boldsymbol{x},\eta}$: Fix $(\boldsymbol{\omega},\boldsymbol{x},\eta) \in \Sigma_+ \times \mathcal{X} \times (0,1)$. Let $U_\eta = \{\boldsymbol{z} \in B_2 : |\partial F_{\boldsymbol{\mu}}(\boldsymbol{\omega} + \eta\boldsymbol{z})| > 1\}$ which is a measure-zero set under $p$ (Lemma 20 in Appendix H). For its complement set $B_2 \backslash U_\eta$, we split $B_2 \backslash U_\eta = \cup_{\boldsymbol{y} \neq \boldsymbol{i}^\star} B_\eta(\boldsymbol{y})$ into possibly overlapping sets $B_\eta(\boldsymbol{y}) = \{\boldsymbol{z} \in B_2 \backslash U_\eta : \nabla_{\boldsymbol{\omega}} F_{\boldsymbol{\pi},\eta}(\boldsymbol{\omega} + \eta\boldsymbol{z}) = \nabla_{\boldsymbol{\omega}} f_{\boldsymbol{y}}(\boldsymbol{\omega} + \eta\boldsymbol{z}, \boldsymbol{\pi})\}$, and define $\psi_{\boldsymbol{\omega},\boldsymbol{y},\eta}(\boldsymbol{y}, \cdot) = \int_{\boldsymbol{z} \in B_\eta(\boldsymbol{y})} \langle \nabla_{\boldsymbol{\omega}} f_{\boldsymbol{y}}(\boldsymbol{\omega} + \eta\boldsymbol{z}, \cdot), \boldsymbol{x} - \boldsymbol{\omega} \rangle \, dp(\boldsymbol{z})$ on each of these sets $B_\eta(\boldsymbol{y})$. Observe that for any $\boldsymbol{\pi} \in \mathbb{R}^K$, we have

$$\psi_{\boldsymbol{\omega},\boldsymbol{x},\eta}(\boldsymbol{\pi}) = \int_{\boldsymbol{z} \in B_2 \backslash U_\eta} \langle \nabla F_{\boldsymbol{\pi},\eta}(\boldsymbol{\omega} + \eta\boldsymbol{z}), \boldsymbol{x} - \boldsymbol{\omega} \rangle \, dp(\boldsymbol{z}) = \sum_{\boldsymbol{y} \neq \boldsymbol{i}^\star} \psi_{\boldsymbol{\omega},\boldsymbol{y},\eta}(\boldsymbol{y}, \boldsymbol{\pi}).$$

To show the continuity of $\psi_{\boldsymbol{\omega},\boldsymbol{x},\eta}(\boldsymbol{\pi})$, it suffices to show that each $\psi_{\boldsymbol{\omega},\boldsymbol{x},\eta}(\boldsymbol{y}, \cdot)$ is continuous. Fix $\boldsymbol{y} \in \mathcal{X} \backslash \{\boldsymbol{i}^\star\}$ and any sequence $\{\boldsymbol{\pi}_n\}_{n=1}^\infty$ converging to $\boldsymbol{\mu}$. Then, for any $\forall \boldsymbol{z} \in B_2$, we have

(i) $|\langle \nabla_{\boldsymbol{\omega}} f_{\boldsymbol{y}}(\boldsymbol{\omega} + \eta\boldsymbol{z}, \boldsymbol{\pi}_n), \boldsymbol{x} - \boldsymbol{\omega} \rangle| \leq \|\nabla_{\boldsymbol{\omega}} f_{\boldsymbol{y}}(\boldsymbol{\omega} + \eta\boldsymbol{z}, \boldsymbol{\pi}_n)\|_\infty \|\boldsymbol{x} - \boldsymbol{\omega}\|_1 \leq 2D\ell$

(ii) $\lim_{n \to \infty} \langle \nabla_{\boldsymbol{\omega}} f_{\boldsymbol{y}}(\boldsymbol{\omega} + \eta\boldsymbol{z}, \boldsymbol{\pi}_n), \boldsymbol{x} - \boldsymbol{\omega} \rangle = \langle \nabla_{\boldsymbol{\omega}} f_{\boldsymbol{y}}(\boldsymbol{\omega} + \eta\boldsymbol{z}, \boldsymbol{\mu}), \boldsymbol{x} - \boldsymbol{\omega} \rangle$. This is because $\nabla_{\boldsymbol{\omega}} f_{\boldsymbol{y}}(\boldsymbol{\omega} + \eta\boldsymbol{z}, \cdot) = \frac{\langle \boldsymbol{i}^\star - \boldsymbol{y}, \cdot \rangle^2 (\boldsymbol{i}^\star \oplus \boldsymbol{y}) \odot (\boldsymbol{\omega} + \eta\boldsymbol{z})^{-2}}{2 \langle \boldsymbol{i}^\star \oplus \boldsymbol{y}, (\boldsymbol{\omega} + \eta\boldsymbol{z})^{-1} \rangle^2}$ by Lemma 19 and Proposition 1 (Appendix C.1) is obviously continuous and that function composition preserves continuity.

From (i) and (ii), the dominated convergence theorem implies that

$$\psi_{\boldsymbol{\omega},\boldsymbol{x},\eta}(\boldsymbol{y}, \boldsymbol{\mu}) = \lim_{n \to \infty} \int_{\boldsymbol{z} \in B_\eta(\boldsymbol{y})} \langle \nabla_{\boldsymbol{\omega}} f_{\boldsymbol{y}}(\boldsymbol{\omega} + \eta\boldsymbol{z}, \boldsymbol{\pi}_n), \boldsymbol{x} - \boldsymbol{\omega} \rangle \, dp(\boldsymbol{z}).$$

This shows the continuity of $\psi_{\boldsymbol{\omega},\boldsymbol{x},\eta}(\boldsymbol{y}, \cdot)$ for each $\boldsymbol{y} \neq \boldsymbol{i}^\star$, and thus $\psi_{\boldsymbol{\omega},\boldsymbol{y},\eta}$ is continuous.

Application of the maximum theorem (Theorem 6): For this part, we take the approach similar to Lemma 6 in [WTP21]. Define

$$\phi(\boldsymbol{\pi}) = \min \left\{ -|\psi_{\boldsymbol{\omega},\boldsymbol{x},\eta}(\boldsymbol{\pi}) - \psi_{\boldsymbol{\omega},\boldsymbol{x},\eta}(\boldsymbol{\mu})| : (\boldsymbol{\omega},\boldsymbol{x},\eta) \in \Sigma_+ \times \mathcal{X} \times (0,1) \right\}.$$

We prove the continuity of $\phi$ on $\mathcal{S} = \mathbb{R}^K \backslash \mathsf{cl}\,(\mathsf{Alt}(\boldsymbol{\mu}))$ by invoking Theorem 6 with the following substitutions:

- $\mathbb{X} = \mathcal{S}$,
- $\mathbb{Y} = \Sigma_+ \times \mathcal{X} \times (0,1)$,
- $\Phi = \Sigma_+ \times \mathcal{X} \times (0,1)$,
- $u(\boldsymbol{\pi}, \boldsymbol{\omega}, \boldsymbol{x}, \eta) = -|\psi_{\boldsymbol{\omega},\boldsymbol{x},\eta}(\boldsymbol{\pi}) - \psi_{\boldsymbol{\omega},\boldsymbol{x},\eta}(\boldsymbol{\mu})|$.

Here we verify that the assumptions of Theorem 6 are satisfied. $\mathbb{X}$ is compactly generated as $\mathcal{S}$ is a metric space; $\Phi$ is continuous as it is a constant map; $u$ is continuous due to the continuity of $\psi_{\boldsymbol{\omega},\boldsymbol{x},\eta}$. To show that $u$ is $\mathbb{K}$-inf compact, consider any compact set $C \subset \mathcal{S}$ and any $y \in \mathbb{R}$. We see that $L_u(y, C \times \Sigma_+ \times \mathcal{X} \times (0,1))$ is compact because it is bounded (as $\Sigma_+ \times \mathcal{X} \times (0,1)$ is bounded and $C$ is compact) and closed (as $u$ is continuous and the preimage of $[0, y]$ is closed). Hence, $\phi$ is continuous on $\mathcal{S}$ by Theorem 6. Finally, by $\phi(\boldsymbol{\mu}) = 0$ and the continuity of $\phi$, there exists $\xi_\epsilon > 0$ such that $\phi(\boldsymbol{\pi}) > -\epsilon$ for any $\|\boldsymbol{\pi} - \boldsymbol{\mu}\|_\infty < \xi_\epsilon$. This completes the proof of (50). □

### G.2 The length of gradients

Throughout this subsection, we fix $\boldsymbol{\mu} \in \Lambda$ and denote $\boldsymbol{i}^\star = \boldsymbol{i}^\star(\boldsymbol{\mu})$, $\triangle_{\boldsymbol{x}} = \triangle_{\boldsymbol{x}}(\boldsymbol{\mu})$, and $\triangle_{\min}(\boldsymbol{\mu}) = \triangle_{\min}$ for short. Here we aim to present Proposition 4, in which (i) quantifies how close an estimate $\boldsymbol{\pi}$ of $\boldsymbol{\mu}$ should be such that $\boldsymbol{i}^\star(\boldsymbol{\pi}) = \boldsymbol{i}^\star$, and (ii) asserts the continuity of any component of $\nabla_{\boldsymbol{\omega}} f_{\boldsymbol{x}}(\boldsymbol{\omega}, \boldsymbol{\pi})$ in $\boldsymbol{\pi}$, and that its gradient with respective to $\boldsymbol{\pi}$ is bounded.

**Proposition 4.** *Any $\boldsymbol{\pi} \in \mathbb{R}^K$ such that $\|\boldsymbol{\pi} - \boldsymbol{\mu}\|_\infty < \frac{\triangle_{\min}}{\sqrt{2KD}}$ satisfies*

(i) $i^\star(\pi) = i^\star$,

(ii) $\forall x \in \mathcal{X}\backslash\{i^\star\}$ and all $k \in [K]$, $\frac{\partial f_x(\omega, \pi)}{\partial \omega_k}$ is continuous in $\pi$ and

$$\left\|\nabla_\pi \left(\frac{\partial f_x(\omega, \pi)}{\partial \omega_k}\right)\right\|_1 \leq 12D^2 \|\mu\|_\infty.$$

**Proof** Proof of (i): Lemma 18 is equivalent to that: any $\pi \in \mathbb{R}^K$ satisfying $\|\mu - \pi\|_\infty < \frac{\triangle_{\min}}{\sqrt{2KD}}$ implies that $\pi \notin \mathrm{cl}\,(\mathrm{Alt}(\mu))$. As closure of finite union equals union of closures,

$$\mathbb{R}^K \backslash \mathrm{cl}\,(\mathrm{Alt}(\mu)) = \mathbb{R}^K \backslash \left(\cup_{x \neq i^\star} \mathrm{cl}\,\left(\{\lambda \in \mathbb{R}^K : \langle i^\star - x, \lambda\rangle < 0\}\right)\right)$$
$$= \mathbb{R}^K \backslash \left(\cup_{x \neq i^\star}\{\lambda \in \mathbb{R}^K : \langle i^\star - x, \lambda\rangle \leq 0\}\right)$$
$$= \{\lambda \in \mathbb{R}^K : i^\star(\lambda) = i^\star\}.$$

Thus, $\pi \notin \mathrm{cl}\,(\mathrm{Alt}(\mu))$ is equivalent to $i^\star(\pi) = i^\star$. This concludes the proof of (i).

Proof of (ii): Fix any $\pi \in \mathbb{R}^K$ satisfying $\|\mu - \pi\|_\infty < \frac{\triangle_{\min}}{\sqrt{2KD}}$. By Lemma 19 and $i^\star(\pi) = i^\star$,

$$\forall k \in [K], \quad \frac{\partial f_x(\omega, \pi)}{\partial \omega_k} = \frac{\langle i^\star - x, \pi\rangle^2 (x_k \oplus i_k^\star)}{2 \langle x \oplus i^\star, \omega^{-1}\rangle^2 \omega_k^2}.$$

Fix $k \in [K]$. Note that the function $\pi \mapsto \frac{\partial f_x(\omega, \pi)}{\partial \omega_k}$ is continuous and differentiable since it consists of inner products, element-wise products, and since its denumerator is always positive. For its derivative,

$$\left\|\nabla_\pi \left(\frac{\partial f_x(\omega, \pi)}{\partial \omega_k}\right)\right\|_1 = \left\|\frac{(i^\star - x)\langle i^\star - x, \pi\rangle (x_k \oplus i_k^\star)}{\langle x \oplus i^\star, \omega^{-1}\rangle^2 \omega_k^2}\right\|_1$$
$$\leq \|(i^\star - x)\langle i^\star - x, \pi\rangle (x_k \oplus i_k^\star)\|_1$$
$$\leq \|i^\star - x\|_1 |\langle i^\star - x, \pi\rangle| \leq 4D^2 \|\pi\|_\infty \leq 12D^2 \|\mu\|_\infty,$$

where the first inequality is because $\langle x \oplus i^\star, \omega^{-1}\rangle \omega_k \geq 1$ if $(x_k \oplus i_k^\star) = 1$; the second is because $x_k \oplus i_k^\star \leq 1$; the third uses $\|i^\star - x\|_1 \leq 2D$ and $|\langle i^\star - x, \pi\rangle| \leq \|i^\star - x\|_1 \|\pi\|_\infty$; the last uses the triangle inequality:

$$\|\pi\|_\infty \leq \|\mu\|_\infty + \|\mu - \pi\|_\infty \leq \|\mu\|_\infty + \frac{\triangle_{\min}}{\sqrt{2KD}} \leq 3 \|\mu\|_\infty,$$

where the last inequality is due to an application of Hölder's inequality to

$$\triangle_{\min} \leq \min_{x \neq i^\star} \|i^\star - x\|_1 \|\mu\|_\infty \leq 2D \|\mu\|_\infty.$$

$\square$

**Lemma 18.** $\inf_{\lambda \in \mathrm{Alt}(\mu)} \|\mu - \lambda\|_\infty \geq \frac{\triangle_{\min}}{\sqrt{2KD}}.$

**Proof** We claim that

$$\inf_{\lambda \in \Lambda : \langle \lambda, i^\star - x\rangle < 0} \|\mu - \lambda\|_2^2 = \frac{\triangle_x}{\|i^\star \oplus x\|_2}, \quad \forall x \neq i^\star. \tag{51}$$

Observe that the proof immediately follows from (51) because the facts that $\|y\|_2 \leq \sqrt{K}\|y\|_\infty$ for any $y \in \mathbb{R}^K$, $\mathrm{Alt}(\mu) = \cup_{x \neq i^\star}\{\lambda \in \Lambda : \langle \lambda, i^\star - x\rangle < 0\}$, and $\|i^\star \oplus x\|_2 \leq \sqrt{2D}$.

Proof of (51): By solving the stationary conditions, i.e., $\nabla_\lambda \mathcal{L}_x(\lambda_x^\star, \alpha^\star) = 2(\mu - \lambda_x^\star) + \alpha^\star(i^\star - x) = 0$ and $\nabla_\alpha \mathcal{L}_x(\lambda_x^\star, \alpha^\star) = \langle \lambda_x^\star, i^\star - x\rangle = 0$, we find

$$\lambda_x^\star = \mu - \frac{\triangle_x(\mu) \odot (i^\star - x)}{\|i^\star \oplus x\|_2^2}$$

is a minimizer for $\inf_{\lambda \in \Lambda : \langle \lambda, i^\star - x\rangle < 0} \|\mu - \lambda\|_2$. (51) follows by plugging $\lambda_x^\star$ into $\|\mu - \lambda\|_2^2$. $\square$

Remind that $\nabla_\omega f_x$ can be evaluated by the following Lemma 19.

**Lemma 19** (Envelope theorem). *Let* $(\boldsymbol{\omega}, \boldsymbol{\mu}) \in \Sigma_+ \times \Lambda$ *and* $\boldsymbol{x} \in \mathcal{X} \setminus \{\boldsymbol{i}^\star\}$. *Define* $\boldsymbol{\lambda}^\star_{\boldsymbol{\omega}, \boldsymbol{\mu}}(\boldsymbol{x}) \in$ $\operatorname{argmin}_{\boldsymbol{\lambda} \in \mathsf{cl}(\mathcal{C}_{\boldsymbol{x}})} \left\langle \boldsymbol{\omega}, \frac{(\boldsymbol{\mu} - \boldsymbol{\lambda})^2}{2} \right\rangle$. *Then,*

$$\nabla_{\boldsymbol{\omega}} f_{\boldsymbol{x}}(\boldsymbol{\omega}, \boldsymbol{\mu}) = \frac{(\boldsymbol{\mu} - \boldsymbol{\lambda}^\star_{\boldsymbol{\omega}, \boldsymbol{\mu}}(\boldsymbol{x}))^2}{2} = \frac{\triangle_{\boldsymbol{x}}(\boldsymbol{\mu})^2 (\boldsymbol{x} \oplus \boldsymbol{i}^\star) \odot \boldsymbol{\omega}^{-2}}{2 \left\langle \boldsymbol{x} \oplus \boldsymbol{i}^\star, \boldsymbol{\omega}^{-1} \right\rangle^2}.$$

**Proof** The first equality is an application of Lemma 6 and Proposition 1 of [WTP21] with $\mathcal{I} = \mathcal{X}$, $\mathcal{J}_{\boldsymbol{x}} = \{\boldsymbol{x}\}$, $\Sigma = \Sigma_K$, $\mathcal{S}_{\boldsymbol{x}} = \{\boldsymbol{\lambda} \in \Lambda : \boldsymbol{i}^\star(\boldsymbol{\lambda}) = \boldsymbol{x}\}$ (see Appendix K.2 and Appendix K.4 in [WTP21] for more details). The second equality substitutes $\boldsymbol{\lambda}^\star_{\boldsymbol{\omega}, \boldsymbol{\mu}}(\boldsymbol{x}) = \boldsymbol{\mu} + \frac{\triangle_{\boldsymbol{x}}(\boldsymbol{\mu})(\boldsymbol{x} - \boldsymbol{i}^\star) \odot \boldsymbol{\omega}^{-1}}{\langle \boldsymbol{x} \oplus \boldsymbol{i}^\star, \boldsymbol{\omega}^{-1} \rangle}$ by using (11)-(14). $\qquad\square$

# H   Stochastic smoothing

This section is devoted to present Proposition 2 and verify the assumptions required for applying Proposition 2 to our objective $F_{\boldsymbol{\mu}}$.

Stochastic smoothing [FKM05, DBW12] is a well-studied technique and has been widely applied to online convex nonsmooth optimization [HK12, H$^+$16]. Proposition 2 is a restatement of existing results. In particular, Proposition 2 (i), (ii) and (iii) directly follow from Lemma E.2 in [DBW12] with $(L_0, u) = (\ell, \eta)$, $f = -\Phi$ and $f_u = -\bar{\Phi}_\eta(\cdot)$, and Proposition 2 (iv) can be established by Jensen's inequality as done in the proof of Theorem 2.1 [DBW12].

**Proposition 2.** *Assume that* $\Phi : \mathbb{R}_{>0}^K \mapsto \mathbb{R}$ *is concave,* $\ell$-*Lipschitz, and differentiable almost everywhere. Let* $B_2 = \{\boldsymbol{v} \in \mathbb{R}^K : \|\boldsymbol{v}\|_2 \leq 1\}$*. For any* $\boldsymbol{\omega} \in \Sigma_+$ *and* $\eta \in (0, \min_{k \in [K]} \omega_k)$*, define*

$$\bar{\Phi}_\eta(\boldsymbol{\omega}) = \mathbb{E}_{\boldsymbol{Z} \sim Uniform(B_2)}[\Phi(\boldsymbol{\omega} + \eta \boldsymbol{Z})]. \tag{6}$$

*Then,* $\bar{\Phi}_\eta(\boldsymbol{\omega})$ *satisfies that:*

    *(i)* $\Phi(\boldsymbol{\omega}) - \eta\ell \leq \bar{\Phi}_\eta(\boldsymbol{\omega}) \leq \Phi(\boldsymbol{\omega})$

    *(ii)* $\nabla\bar{\Phi}_{\boldsymbol{\mu},\eta}(\boldsymbol{\omega}) = \mathbb{E}_{\boldsymbol{Z} \sim Uniform(B_2)}[\nabla\Phi_{\boldsymbol{\mu}}(\boldsymbol{\omega} + \eta \boldsymbol{Z})]$

    *(iii)* $\bar{\Phi}_\eta$ *is* $\frac{\ell K}{\eta}$-*smooth*

    *(iv) if* $\eta > \eta' > 0$*, then* $\bar{\Phi}_{\eta'}(\boldsymbol{\omega}) \geq \bar{\Phi}_\eta(\boldsymbol{\omega})$

Now, we validate assumptions of Proposition 2 on $F_{\boldsymbol{\mu}}$. The concavity of $F_{\boldsymbol{\mu}}$, which is shown by [WTP21], follows from the facts that each $f_{\boldsymbol{x}}(\cdot, \boldsymbol{\mu})$ is concave and that $F_{\boldsymbol{\mu}}$ is a minimum of these functions $f_{\boldsymbol{x}}(\cdot, \boldsymbol{\mu})$ over all possible $\boldsymbol{x}$. The Lipschitzness of $F_{\boldsymbol{\mu}}$ is shown in Lemma 21 in Appendix I). Hence, it remains to show the almost-everywhere differentiability of $F_{\boldsymbol{\mu}}$. To show that the set of non-differentiable points of $F_{\boldsymbol{\mu}}$, i.e.,

$$\bigcup_{\boldsymbol{x}, \boldsymbol{x}' \in \mathcal{X} \setminus \{\boldsymbol{i}^\star(\boldsymbol{\mu})\}, \, \boldsymbol{x} \neq \boldsymbol{x}'} \{\boldsymbol{z} \in B_2 : f_{\boldsymbol{x}}((\boldsymbol{\omega} + \eta\boldsymbol{z}, \boldsymbol{\mu}) = f_{\boldsymbol{x}'}(\boldsymbol{\omega} + \eta\boldsymbol{z}, \boldsymbol{\mu})\},$$

is measure-zero under Uniform($B_2$), it suffices to show the following lemma.

**Lemma 20.** *Let* $\boldsymbol{\mu} \in \Lambda$ *and* $\boldsymbol{x}_1, \boldsymbol{x}_2$ *be distinct actions in* $\mathcal{X} \setminus \{\boldsymbol{i}^\star(\boldsymbol{\mu})\}$*. Then under the probability measure of Uniform($B_2$),*

$$\{\boldsymbol{z} \in B_2 : f_{\boldsymbol{x}_1}(\boldsymbol{\omega} + \eta\boldsymbol{z}, \boldsymbol{\mu}) = f_{\boldsymbol{x}_2}(\boldsymbol{\omega} + \eta\boldsymbol{z}, \boldsymbol{\mu})\}$$

*is a measure-zero set.*

**Proof**   To simplify the notation, let $\boldsymbol{i}^\star = \boldsymbol{i}^\star(\boldsymbol{\mu})$ and $\triangle_{\boldsymbol{x}} = \triangle_{\boldsymbol{x}}(\boldsymbol{\mu})$. Thanks to the close-form expressions of $f_{\boldsymbol{x}_1}$ and $f_{\boldsymbol{x}_2}$, $\boldsymbol{z} \in B_2$ such that $f_{\boldsymbol{x}_1}(\boldsymbol{\omega} + \eta\boldsymbol{z}, \boldsymbol{\mu}) = f_{\boldsymbol{x}_2}(\boldsymbol{\omega} + \eta\boldsymbol{z}, \boldsymbol{\mu})$ are the points satisfying that:

$$\frac{\triangle_{\boldsymbol{x}_1}^2}{2 \langle \boldsymbol{x}_1 \oplus \boldsymbol{i}^\star, (\boldsymbol{\omega} + \eta\boldsymbol{z})^{-1} \rangle} = \frac{\triangle_{\boldsymbol{x}_2}^2}{2 \langle \boldsymbol{x}_2 \oplus \boldsymbol{i}^\star, (\boldsymbol{\omega} + \eta\boldsymbol{z})^{-1} \rangle}.$$

In other words, the set of interests is

$$\left\{ \boldsymbol{z} \in B_2 : \sum_{k=1}^K a_k \prod_{k' \neq k} (\omega_{k'} + \eta z_{k'}) = 0 \right\}, \tag{52}$$

where $a_k = (\boldsymbol{x}_2 \oplus \boldsymbol{i}^\star)_k \triangle_{\boldsymbol{x}_1}^2 - (\boldsymbol{x}_1 \oplus \boldsymbol{i}^\star)_k \triangle_{\boldsymbol{x}_2}^2$ for all $k \in [K]$. We claim that $\boldsymbol{a}$ is a non-zero vector. Otherwise, $a_k = 0, \forall k \in [K]$, which together with the fact that $\triangle_{\boldsymbol{x}_1}^2, \triangle_{\boldsymbol{x}_2}^2 > 0$ directly imply $(\boldsymbol{x}_2 \oplus \boldsymbol{i}^\star)_k = 0$ if and only if $(\boldsymbol{x}_1 \oplus \boldsymbol{i}^\star)_k = 0$. That means $\boldsymbol{x}_1 = \boldsymbol{x}_2$, but this becomes a contradiction. Therefore, the set in (52) are the roots of a non-zero polynomial inside $B_2$, and hence it is a measure-zero set (see e.g. Lemma in [Oka73]). $\qquad\square$

# I   Lischitzness of $F_{\boldsymbol{\mu}}$ and boundness of $F_{\boldsymbol{\mu}}$ on $\Sigma_K \cap \mathbb{R}_{>0}^K$

In this section, we show the Lipschitzness of $F_{\boldsymbol{\mu}}(\boldsymbol{v}) = \min_{\boldsymbol{x} \neq \boldsymbol{i}^\star} f_{\boldsymbol{x}}(\boldsymbol{v}, \boldsymbol{\mu})$ for $\boldsymbol{v} \in \mathbb{R}_{>0}^K$. Let $\boldsymbol{x}_e$ be an equilibrium action such that $F_{\boldsymbol{\mu}}(\boldsymbol{v}) = f_{\boldsymbol{x}_e}(\boldsymbol{v}, \boldsymbol{\mu})$. We will use the envelope theorem (Lemma 19 in Appendix G) to evaluate $\nabla_{\boldsymbol{\omega}} f_{\boldsymbol{x}_e}(\boldsymbol{v}, \boldsymbol{\mu})$ in closed-form, and then bound its length. We will also derive an upper bound of $F_{\boldsymbol{\mu}}(\boldsymbol{v})$ valid for any positive vector $\boldsymbol{v}$ in the $(K-1)$-dimensional simplex $\Sigma_K$. In what below, we denote $\boldsymbol{i}^\star = \boldsymbol{i}^\star(\boldsymbol{\mu})$ and $\triangle_{\boldsymbol{x}} = \triangle_{\boldsymbol{x}}(\boldsymbol{\mu})$ for any $\boldsymbol{x} \neq \boldsymbol{i}^\star$ for short.

**Lemma 21.** *Let $\boldsymbol{\mu} \in \Lambda$ and $\ell = 2D^2 \|\boldsymbol{\mu}\|_\infty^2$. Then, $F_{\boldsymbol{\mu}}$ is $\ell$-Lipschitz with respect to $\|\cdot\|_\infty$ on $\mathbb{R}_{>0}^K$,*

**Proof**   Let $\boldsymbol{v} \in \mathbb{R}_{>0}^K$. Recall that $F_{\boldsymbol{\mu}}(\boldsymbol{v}) = \min_{\boldsymbol{x} \neq \boldsymbol{i}^\star} f_{\boldsymbol{x}}(\boldsymbol{v}, \boldsymbol{\mu})$, and each $f_{\boldsymbol{x}}(\boldsymbol{v}, \boldsymbol{\mu})$ is differentiable (proven in Lemma 19 in Appendix G.2). Hence if $\boldsymbol{x}$ is the action such that $F_{\boldsymbol{\mu}}(\boldsymbol{v}) = f_{\boldsymbol{x}}(\boldsymbol{v}, \boldsymbol{\mu})$, the concavity of $F_{\boldsymbol{\mu}}(\boldsymbol{v})$ and the fact that $\nabla_{\boldsymbol{\omega}} f_{\boldsymbol{x}}(\boldsymbol{v}, \boldsymbol{\mu})$ is the subdifferential of $F_{\boldsymbol{\mu}}$ on $\boldsymbol{v}$ yield that

$$\forall \boldsymbol{v}' \in \mathbb{R}_{>0}^K, |F_{\boldsymbol{\mu}}(\boldsymbol{v}) - F_{\boldsymbol{\mu}}(\boldsymbol{v}')| \leq |\langle \nabla_{\boldsymbol{\omega}} f_{\boldsymbol{x}}(\boldsymbol{v}, \boldsymbol{\mu}), \boldsymbol{v} - \boldsymbol{v}' \rangle| \leq \|\nabla_{\boldsymbol{\omega}} f_{\boldsymbol{x}}(\boldsymbol{v}, \boldsymbol{\mu})\|_1 \|\boldsymbol{v} - \boldsymbol{v}'\|_\infty,$$

where the last inequality stems from Hölder's inequality. From the above, the $\ell$-Lipschitz can be derived by upper bounding $\|\nabla_{\boldsymbol{\omega}} f_{\boldsymbol{x}}(\boldsymbol{v}, \boldsymbol{\mu})\|_1$ by $\ell$. Now applying Lemma 19 in Appendix G.2 yields

$$\|\nabla_{\boldsymbol{\omega}} f_{\boldsymbol{x}}(\boldsymbol{v}, \boldsymbol{\mu})\|_1 = \left\| \frac{(\boldsymbol{\mu} - \boldsymbol{\lambda}_{\boldsymbol{v},\boldsymbol{\mu}}^\star(\boldsymbol{x}, \alpha_{\boldsymbol{x}}^\star))^2}{2} \right\|_1 = \frac{\|\boldsymbol{v}^{-2} \odot (\boldsymbol{x} \oplus \boldsymbol{i}^\star)\|_1 \triangle_{\boldsymbol{x}}^2}{2 \langle \boldsymbol{x} \oplus \boldsymbol{i}^\star, \boldsymbol{v}^{-1} \rangle^2}. \tag{53}$$

To simplify the above, we observe that

$$\langle \boldsymbol{x} \oplus \boldsymbol{i}^\star, \boldsymbol{v}^{-1} \rangle^2 = \left( \sum_{k=1}^K v_k^{-1} \mathbb{1}\{x_k \neq i_k^\star\} \right)^2 \geq \sum_{k=1}^K v_k^{-2} \mathbb{1}\{x_k \neq i_k^\star\} = \|\boldsymbol{v}^{-2} \odot (\boldsymbol{x} \oplus \boldsymbol{i}^\star)\|_1, \tag{54}$$

where the inequality uses the fact that $v_k > 0$ for all $k \in [K]$. Also,

$$\triangle_{\boldsymbol{x}} = \langle \boldsymbol{i}^\star - \boldsymbol{x}, \boldsymbol{\mu} \rangle \leq \|\boldsymbol{i}^\star - \boldsymbol{x}\|_1 \|\boldsymbol{\mu}\|_\infty \leq 2D \|\boldsymbol{\mu}\|_\infty. \tag{55}$$

Thus, (53)-(54)-(55) yields that $\|\nabla_{\boldsymbol{\omega}} f_{\boldsymbol{x}}(\boldsymbol{v}, \boldsymbol{\mu})\|_1 \leq 2D^2 \|\boldsymbol{\mu}\|_\infty^2$.   $\square$

**Lemma 22.** *Let $\boldsymbol{\mu} \in \Lambda$ and $\ell = 2D^2 \|\boldsymbol{\mu}\|_\infty^2$. Then, $\max_{\boldsymbol{\omega} \in \Sigma_K \cap \mathbb{R}_{>0}^K} F_{\boldsymbol{\mu}}(\boldsymbol{\omega}) \leq \ell$.*

**Proof**   Observe that $f_{\boldsymbol{x}}(\boldsymbol{v}, \boldsymbol{\mu}) = \langle \boldsymbol{\omega}, \nabla_{\boldsymbol{\omega}} f_{\boldsymbol{x}}(\boldsymbol{v}, \boldsymbol{\mu}) \rangle$ for any $\boldsymbol{x} \neq \boldsymbol{i}^\star$. Combining this observation with the fact that $\triangle_{\boldsymbol{x}} \leq 2D \|\boldsymbol{\mu}\|_\infty$ (as argued in (54) in proof of Lemma 21) implies:

$$F_{\boldsymbol{\mu}}(\boldsymbol{v}) = \min_{\boldsymbol{x} \neq \boldsymbol{i}^\star} \frac{\triangle_{\boldsymbol{x}}^2}{2 \langle \boldsymbol{x} \oplus \boldsymbol{i}^\star, \boldsymbol{v}^{-1} \rangle} \leq \frac{(2D \|\boldsymbol{\mu}\|_\infty)^2}{2} = \ell,$$

where the first inequality is because $\langle \boldsymbol{x} \oplus \boldsymbol{i}^\star, \boldsymbol{v}^{-1} \rangle \geq \min_{k \in [K]} v_k^{-1} \geq 1$ (as $\boldsymbol{v} \in \Sigma_K$ and $v_k > 0$ for all $k \in [K]$).The proof is completed since $\boldsymbol{v}$ is taken arbitrarily.   $\square$

## J Proofs related to combinatorial sets

**Assumption 1.** *(i) There exists a polynomial-time algorithm identifying $i^\star(v)$ for any $v \in \mathbb{R}^K$; (ii) $\mathcal{X}$ is inclusion-wise maximal, i.e., there is no $x, x' \in \mathcal{X}$ s.t. $x < x'$; (iii) for each $k \in [K]$, there exists $x \in \mathcal{X}$ such that $x_k = 1$; (iv) $|\mathcal{X}| \geq 2$.*

As claimed in §2.2, Assumption 1 holds for the following combinatorial sets:

- *m-sets*: $\mathcal{X} = \{x \in \{0,1\}^K : \|x\|_1 = m\}$
- *spanning forests*: $\mathcal{X}$ is a set of all spanning forests in a given graph
- *bipartite matchings*: $\mathcal{X}$ is a set of all maximal matchings in a given bipartite graph
- *s-t paths*: $\mathcal{X}$ is the set of all source-destination paths in a directed acyclic graph

In what below, we present a simple proof for the above examples.

**Proof** Suppose (iii) (iv) hold (as we can always achieve (iii) by removing arms not covered by $\mathcal{X}$ and (iv) holds for non-trivial sets). For (i), it is well-known that a polynomial-time LM Oracle, i.e., $i^\star(\cdot)$, exists for each of the discussed combinatorial structures. For example, see Chapter 39 in [S$^+$03] for the greedy algorithm for matroids (applicable to $m$-set and spanning forests), Chapter 41 in [S$^+$03] for the augmentation-based algorithm for 2-matroid intersection (applicable to bipartite matchings), and algorithms such as Dijkstra's algorithm for $s$-$t$ paths.

It remains to verify (ii) the inclusion-wise maximal property of $\mathcal{X}$. For $\mathcal{X}$ as $m$-sets, the inclusion-wise maximal property clearly holds because any binary vector $x' > x$ (resp. $x' < x$) for some $x \in \mathcal{X}$ must have $\sum_{k \in [K]} x'_k > m$ (resp. $< m$) and thus $x' \notin \mathcal{X}$. The case is similar for $\mathcal{X}$ as spanning forests since the number of edges of any spanning forests of a graph is the same. For $\mathcal{X}$ as maximal matchings in which the term 'maximal' exactly refers to being inclusion-wise maximal, (ii) directly follows from the definition. For $\mathcal{X}$ as the set of all source-destination paths in an acyclic graph, if there exists any source-destination path $x' > x$ for some $x \in \mathcal{X}$ then $x'$ must contain a cycle, so inclusion-wise maximal property holds. $\qquad\square$

**Lemma 2.** *Let $v \in \mathbb{R}^K$ and $x \in \mathcal{X}$. Under Assumption 1, there exists an algorithm that solves $\max_{x' \in \mathcal{X}: x' \neq x} \langle v, x' \rangle$ by only making at most $D$ queries to the LM Oracle.*

**Proof** Fix $x \in \mathcal{X}$. Assume $v \neq 0_K$ (as otherwise, any $x' \neq x$ is a second-best action). Inspired by Lawler-Murty's $m$-best algorithm [Law72], we will prove this lemma by considering the algorithm described as follows. It first computes $i^\star(v)$ by the LM Oracle, and returns it as the output if $i^\star(v) \neq x$. Otherwise, we identify the second-best action by the program below:

$$\max_{k \in [K]: x_k = 1} \left\langle v, i^\star\left(v^{(k)}\right) \right\rangle, \quad \text{where} \quad v_i^{(k)} = \begin{cases} -3 \|v\|_1 & \text{if } i = k \\ v_i & \text{otherwise.} \end{cases} \tag{56}$$

Intuitively, for each arm $k$ of $x$, the action $i^\star(v^{(k)})$ represents the best one among all actions without $k$ (we have a strong negative weight on the $k$-th component of $v^{(k)}$). In the following, we will show that at least one of $\{i^\star(v^{(k)}) : k \in [K], x_k = 1\}$ is the second-best action.

More precisely, we will show that for any maximizer $a \in [K]$ to (56), $i^\star(v^{(a)})$ is a second-best action. Consider if $(i^\star(v^{(a)}))_a = 0$, then the claim follows from the fact that $i^\star(v^{(a)})$ is the best among all actions without $a$ and also the best in $\{i^\star(v^{(k)}) : k \in [K], x_k = 1\}$. It suffices to show that $(i^\star(v^{(a)}))_a = 1$ cannot happen. If $(i^\star(v^{(a)}))_a = 1$, then it follows from Assumption 1 (iv) $|\mathcal{X}| \geq 2$ and (ii) the inclusion-wise maximality of $\mathcal{X}$ that there is another action $x'$ such that $x'_k = 0$ but $x_k = 1$ for some $k \in [K]$. So, by $i^\star(v^{(a)})_a = 1$, $v \neq 0_K$ and the definition of $v^{(a)}$, we get

$$\left\langle v, i^\star\left(v^{(a)}\right) \right\rangle = \sum_{j \in [K]: i^\star(v^{(a)})_j = 1, j \neq a} v_j - 3\|v\|_1 \leq -2\|v\|_1 < \langle v, x' \rangle \leq \left\langle v, i^\star\left(v^{(k)}\right) \right\rangle,$$

which contradicts the optimality of $a$ (as it would imply that $i^\star(v^{(k)})$ is better).

Finally, as $\|\boldsymbol{x}\|_1 \leq D$, the number of LM Oracle calls required for solving (56) is at most $D$. □

Finally, we present the property of $\mathcal{X}_0$ briefly argued in § 4.2.

**Lemma 23.** *Let $\boldsymbol{e}_k$ is the $k$-th column of an identity matrix. Under Assumption 1, $\mathcal{X}_0$ is a $[K]$-covering set and $|\mathcal{X}_0| \geq 2$.*

**Proof** Showing that $\mathcal{X}_0$ covers $[K]$: Assumption 1 (iii) ensures $\{\boldsymbol{x} \in \mathcal{X} : x_k = 1\} \neq \emptyset$, and it follows that $\max_{\boldsymbol{x} \in \mathcal{X}} \langle \boldsymbol{x}, \boldsymbol{e}_k \rangle = 1$, i.e., $(\boldsymbol{i}^\star(\boldsymbol{e}_k))_k = 1$. As $(\boldsymbol{i}^\star(\boldsymbol{e}_k))_k = 1$ holds for all $k$, the proof is completed.

Showing that $|\mathcal{X}_0| \geq 2$: Suppose on the contrary, $|\mathcal{X}_0| = 1$. Thanks to Assumption 1 (iv) $|\mathcal{X}| \geq 2$, there exists $\boldsymbol{x} \in \mathcal{X}$ such that $x_k = \langle \boldsymbol{e}_k, x \rangle \geq \langle \boldsymbol{e}_k, x' \rangle = x'_k$ for all $k \in [K]$, $x' \neq x$. Together with Assumption 1 (iii), one can easily deduce that $x_k = 1$ for all $k \in [K]$. However, this implies $\boldsymbol{x}' < \boldsymbol{x}$ for any $\boldsymbol{x}' \neq \boldsymbol{x}$ and hence contradicts to Assumption 1 (ii) that $\mathcal{X}$ is inclusion-wise maximal. □

# K   Sample complexity lower bound

In this section, we assume $\boldsymbol{\mu} \in \Lambda$ and $\delta \in (0,1)$ is fixed, and show Theorem 7 by adapting Lemma 19 in [KCG16].

**Lemma 24** ([KCG16]).  *Any $\delta$-PAC algorithm satisfies*

$$\forall \boldsymbol{\lambda} \in \mathsf{Alt}(\boldsymbol{\mu}), \quad \sum_{k \in [K]} \sum_{\boldsymbol{x} \in \mathcal{X}: x_k = 1} \mathbb{E}_{\boldsymbol{\mu}}[N_{\boldsymbol{x}}(\tau)] \frac{(\mu_k - \lambda_k)^2}{2} \geq \mathsf{kl}(\delta, 1 - \delta). \tag{57}$$

**Theorem 7.**  *Any $\delta$-PAC strategy satisfies*

$$\mathbb{E}_{\boldsymbol{\mu}}[\tau] \geq T^{\star}(\boldsymbol{\mu}) \mathsf{kl}(\delta, 1 - \delta) \quad with \quad T^{\star}(\boldsymbol{\mu})^{-1} = \sup_{\boldsymbol{\omega} \in \Sigma} \inf_{\boldsymbol{\lambda} \in \mathsf{Alt}(\boldsymbol{\mu})} \left\langle \boldsymbol{\omega}, \frac{(\boldsymbol{\mu} - \boldsymbol{\lambda})^2}{2} \right\rangle, \tag{1}$$

*where $\Sigma = \{\sum_{\boldsymbol{x} \in \mathcal{X}} w_{\boldsymbol{x}} : \boldsymbol{w} \in \Sigma_{|\mathcal{X}|}\}$ and $\mathsf{Alt}(\boldsymbol{\mu}) = \{\boldsymbol{\lambda} \in \Lambda : \boldsymbol{i}^{\star}(\boldsymbol{\lambda}) \neq \boldsymbol{i}^{\star}(\boldsymbol{\mu})\}$.*

**Proof**   We have: under any algorithm,

$$\sup_{\boldsymbol{\omega} \in \Sigma} \inf_{\boldsymbol{\lambda} \in \mathsf{Alt}(\boldsymbol{\mu})} \sum_{k \in [K]} \omega_k \frac{(\mu_k - \lambda_k)^2}{2} \geq \inf_{\boldsymbol{\lambda} \in \mathsf{Alt}(\boldsymbol{\mu})} \sum_{k \in [K]} \sum_{\boldsymbol{x} \in \mathcal{X}: x_k = 1} \frac{\mathbb{E}_{\boldsymbol{\mu}}[N_{\boldsymbol{x}}(\tau)]}{\mathbb{E}_{\boldsymbol{\mu}}[\tau]} \frac{(\mu_k - \lambda_k)^2}{2},$$

Hence if the algorithm is $\delta$-PAC, by Lemma 24,

$$\mathbb{E}_{\boldsymbol{\mu}}[\tau] \sup_{\boldsymbol{\omega} \in \Sigma} \inf_{\boldsymbol{\lambda} \in \mathsf{Alt}(\boldsymbol{\mu})} \sum_{k \in [K]} \omega_k \frac{(\mu_k - \lambda_k)^2}{2} \geq \inf_{\boldsymbol{\lambda} \in \mathsf{Alt}(\boldsymbol{\mu})} \sum_{k \in [K]} \sum_{\boldsymbol{x} \in \mathcal{X}: x_k = 1} \mathbb{E}_{\boldsymbol{\mu}}[N_{\boldsymbol{x}}(\tau)] \frac{(\mu_k - \lambda_k)^2}{2}$$

$$\geq \mathsf{kl}(\delta, 1 - \delta).$$

$\square$

**Lemma 1.**  *For any $\boldsymbol{\mu} \in \Lambda$, $T^{\star}(\boldsymbol{\mu}) \leq 4KD\triangle_{\min}(\boldsymbol{\mu})^{-2}$.*

**Proof**   Take $\boldsymbol{\omega}_0 = \sum_{\boldsymbol{x} \in \mathcal{X}_0} \boldsymbol{x}/|\mathcal{X}_0| \in \Sigma$, where $\mathcal{X}_0 = \{\boldsymbol{i}^{\star}(\boldsymbol{e}_k) : k \in [K]\}$. Observe that $\boldsymbol{\omega}_0 \geq \mathbf{1}_K/K$ by Lemma 23 (which leads to $\sum_{\boldsymbol{x} \in \mathcal{X}_0} \boldsymbol{x} \geq \mathbf{1}_K$ and $1/|\mathcal{X}_0| \geq 1/K$). Thus,

$$F_{\boldsymbol{\mu}}(\boldsymbol{\omega}_0) = \min_{\boldsymbol{x} \neq \boldsymbol{i}^{\star}(\boldsymbol{\mu})} \frac{\triangle_{\boldsymbol{x}}(\boldsymbol{\mu})^2}{2 \left\langle \boldsymbol{x} \oplus \boldsymbol{i}^{\star}(\boldsymbol{\mu}), \boldsymbol{\omega}_0^{-1} \right\rangle} \geq \frac{\triangle_{\min}(\boldsymbol{\mu})^2}{4KD},$$

where we used Proposition 1 in §3.1 to obtain the equality, and the last inequality is because

$$\left\langle \boldsymbol{x} \oplus \boldsymbol{i}^{\star}(\boldsymbol{\mu}), \boldsymbol{\omega}_0^{-1} \right\rangle \leq \|\boldsymbol{x} \oplus \boldsymbol{i}^{\star}(\boldsymbol{\mu})\|_1 \|\boldsymbol{\omega}_0^{-1}\|_\infty \leq \frac{2D}{\min_{k \in [K]}(\boldsymbol{\omega}_0)_k} \leq 2KD.$$

As $T^{\star}(\boldsymbol{\mu})^{-1} = \max_{\boldsymbol{\omega} \in \Sigma} F_{\boldsymbol{\mu}}(\boldsymbol{\omega}) \geq F_{\boldsymbol{\mu}}(\boldsymbol{\omega}_0)$, we then have $T^{\star}(\boldsymbol{\mu}) \leq \frac{4KD}{\triangle_{\min}(\boldsymbol{\mu})^2}$.   $\square$

## L Extension to the transductive setting

In this section, we extend our results to the transductive combinatorial semi-bandits. In transductive best-arm identification with fixed confidence with semi-bandit feedback [JMKK21], the decision maker is given an exploration set $\mathcal{A} \subseteq \{0,1\}^K$ and a decision set $\mathcal{X} \subseteq \{0,1\}^K$ ($\mathcal{A}$ might differ from $\mathcal{X}$), and at each round, she selects an action in $\mathcal{A}$ to receive a semi-bandit feedback. Her goal is to identify the best action in $\mathcal{X}$ using as few samples as possible.

**Notation.** Let $\mathcal{M} \subseteq \{0,1\}^K$ be any set of actions. We use $\boldsymbol{i}^\star_{\mathcal{M}}(\boldsymbol{\mu})$ to denote any maximizer in $\mathcal{M}$ of the linear maximization $\max_{\boldsymbol{x} \in \mathcal{M}} \langle \boldsymbol{x}, \boldsymbol{\mu} \rangle$. We also use $\Sigma_{\mathcal{M}} = \{\sum_{\boldsymbol{x} \in \mathcal{M}} w_{\boldsymbol{x}} : \boldsymbol{w} \in \Sigma_{|\mathcal{M}|}\}$.

**Sample complexity lower bound.** The generalization of Theorem 7 to the transductive setting has been made in [JMKK21]: any $\delta$-PAC algorithm satisfies

$$\mathbb{E}_{\boldsymbol{\mu}}[\tau] \geq T^\star(\boldsymbol{\mu}) \mathsf{kl}(\delta, 1 - \delta) \quad \text{with} \quad T^\star(\boldsymbol{\mu})^{-1} = \sup_{\boldsymbol{\omega} \in \Sigma_{\mathcal{A}}} \inf_{\boldsymbol{\lambda} \in \mathsf{Alt}(\boldsymbol{\mu})} \left\langle \boldsymbol{\omega}, \frac{(\boldsymbol{\mu} - \boldsymbol{\lambda})^2}{2} \right\rangle. \tag{58}$$

The inner optimization is still with respect to $\mathcal{X}$ while the outer optimization is with respect to the exploration set $\mathcal{A}$. Refer to Appendix C in [JMKK21] for the proof.

**Transductive** `P-FWS` **algorithm.** Assumption 1 has to be extended. It now needs to ensure that $\boldsymbol{i}^\star_{\mathcal{A}}(\boldsymbol{v})$ for any $\boldsymbol{v} \in \mathbb{R}^K$ can be computed in polynomial-time. The `P-FWS` algorithm also needs to be adapted to the transductive setting. This is done by the following two modifications:

- $[K]$-covering set: $\mathcal{X}_0 \leftarrow \{\boldsymbol{i}^\star_{\mathcal{A}}(\boldsymbol{e}_k) : k \in [K]\}$
- FW update: $\boldsymbol{x}(t) \leftarrow \boldsymbol{i}^\star_{\mathcal{A}}\left(\nabla \tilde{F}_{\hat{\boldsymbol{\mu}}(t-1), \eta_t, n_t}(\hat{\boldsymbol{\omega}}(t-1))\right)$

**Analysis of** `P-FWS`**.** Let $D_{\mathcal{A}} = \max_{\boldsymbol{x} \in \mathcal{A}} \|\boldsymbol{x}\|_1$. The analysis is easily extended by replacing $(D, \mathcal{X})$ with $(D_{\mathcal{A}}, \mathcal{A})$ in Appendix D, Appendix E, Appendix F and Appendix G whenever the context is subject to the exploration set rather than the decision set.