# OpenReview forum: "Closing the Computational-Statistical Gap in Best Arm Identification for Combinatorial Semi-bandits"
_NeurIPS.cc/2023/Conference — NeurIPS 2023 poster_

### Official Review · Reviewer_b447 · 2023-07-04

**Soundness:** 4 excellent
**Presentation:** 3 good
**Contribution:** 3 good
**Rating:** 7
**Confidence:** 3

**Summary:**

The paper presents Perturbed Frank-Wolfe Sampling (P-FWS), an algorithm for the best arm identification problem in combinatorial semi-bandits in the fixed confidence setting. The algorithm achieves instance-specific minimal sample complexity in the high confidence regime and polynomial sample complexity guarantees in the moderate confidence regime. The authors show that P-FWS closes the computational-statistical gap in best arm identification in combinatorial semi-bandits. They describe the design of P-FWS, which starts from the optimization problem defining the information-theoretical and instance-specific sample complexity lower bound. P-FWS solves this problem in an online manner using the Frank-Wolfe algorithm with computationally efficient successive updates.

**Strengths:**

P-FWS is a new algorithm that addresses the best arm identification problem in combinatorial semi-bandits and achieves optimal sample complexity in both high and moderate confidence regimes. This makes it a significant contribution to the field. The paper highlights the closure of the computational-statistical gap in best arm identification in combinatorial semi-bandits. This is an important result, as it demonstrates the algorithm's efficiency in terms of both computational complexity and statistical performance.

**Weaknesses:**

The paper primarily focuses on theoretical analysis and does not include empirical experiments or evaluations on real-world datasets. While the theoretical guarantees are valuable, empirical results could provide further insights into the algorithm's practical performance and generalizability.

**Questions:**

How do you think your work could be extended beyond the LM Oracle ? For instance, in https://arxiv.org/abs/2302.11182, there are several combinatorial problems that do not fit your setting. I think it should be good to add this reference (and perhaps other) and mention (as future work or limitation) that more general combinatorial problems exist, and that your setting is restricted.

**Limitations:**

The paper assumes the availability of a computationally efficient Linear Maximization (LM) Oracle, which can identify the best action given knowledge of the parameter µ. While this assumption may hold for some combinatorial sets of actions, it may not be applicable in all scenarios.

---

> ### Author Rebuttal · Authors · 2023-08-09
>
> Many thanks for your careful review and positive feedback.
>
> About experiments. In this paper, we wanted to highlight its strong theoretical contributions: for combinatorial bandits, our algorithm P-FWS is the first polynomial-time algorithm that is asymptotically optimal in the high confidence regime. It also has a sample complexity polynomial in $K$ in the moderate confidence regime. We implemented our algorithm P-FWS. But we found it difficult to compare its performance to that of other algorithms, for these algorithms are computationally too challenging (it takes too much time to run). Here is an illustrative example. We compare P-FWS to a simplified version of CombGame [36]. The experiments are performed on a Macbook (Apple M1) with 8 cores and 16GB memory. We made 100 runs to compute the expected sample complexity of the algorithms.
>  * On a graph with $K=20$ edges and a decision set consisting of $|X|=21025$ spanning trees and confidence $\delta=0.1$, P-FWS has an expected sample complexity $\tau$ of 1139 samples while CombGame with OFW has an expected sample complexity of 1325 samples but already takes 21 minutes for each run.
>  * Now on a graph of $K=25$ edges and $|X|=0.3$ million spanning trees and confidence $\delta=0.1$, P-FWS expected sample complexity is 2000 samples (and in average, each run takes 17 hours) while any run for CombGame fails to finish within 2 days.
>
> Note that the original CombGame algorithm could not be implemented – it takes way too much time to finish. We had to simplify it. More precisely, we use our MCP subroutine to perform two components of CombGame, namely the best-response player and the stopping rule. Even with this simplification, CombGame rapidly becomes computationally intractable as the size of the decision set increases.
>
> About your comment on problems where the LM oracle cannot be implemented in polynomial time. Thanks for this remark and the reference. We will add and discuss it! This issue remains beyond the scope of our paper. It would be interesting to investigate whether the techniques of (Yang, Feidiao, et al.) could be used. We will mention this in the conclusion.
>
> Reference:
>  * (Yang, Feidiao, et al.) "Follow the perturbed approximate leader for solving semi-bandit combinatorial optimization." Frontiers of Computer Science 15 (2021)

---

> > ### Comment · Reviewer_b447 · 2023-08-13
> >
> > I have read the other reviews and the rebuttal. I am satisfied with the answer provided by the authors. I will not change my score.

---

### Official Review · Reviewer_d388 · 2023-07-07

**Soundness:** 3 good
**Presentation:** 2 fair
**Contribution:** 3 good
**Rating:** 6
**Confidence:** 3

**Summary:**


This manuscript studies the asymptotically optimal sample complexity for the best arm identification in stochastic combinatorial semi-bandits, under the fixed confidence setting. The main contribution is the introduction of a computationally efficient algorithm which achieves the asymptotically optimal sample complexity in the high-confidence regime, confirming the conjecture of [Jourdan et al. 2021] on the nonexistence of a statistical-computational gap. The proposed algorithm relies on the change-of-measure lower bound (Eqn. (1)) in [Garivier and Kaufmann, 2016], with two main ingredients:

1. To solve the inner minimization problem, using the Lagrangian form the authors formulate it as a two-player zero-sum game. The minimizing player (choosing a super arm) uses a FTPL algorithm which is oracle-efficient based on a linear maximization (LM) oracle; the maximizing player (choosing the Lagrangian parameter) plays the best response. The standard result of online learning shows that the average plays converge to the minimax solution. Significance: medium.

2. To solve the outer maximization problem, the authors use a perturbed Frank-Wolfe sampling algorithm. First, the objective function is smoothed by a Gaussian kernel to overcome the issue of multiple gradients. Second, the learner computes the estimated gradient for the smoothed objective and plays the best response to the estimated gradient. A stopping rule, as well as forced exploration rules, is also applied in the algorithm. Significance: medium.

**Strengths:**

This manuscript proved the nonexistence of the statistical-computational gap in stochastic combinatorial semi-bandits and resolved a conjecture of [Jourdan et al. 2021]. The components in the algorithm design, i.e. the two-player games and the perturbed Frank-Wolfe sampling, are both interesting.

**Weaknesses:**

Overall I like this paper and lean towards acceptance. However, I do have some concerns on the significance of the results:

1. The problem in study, i.e. a computationally efficient algorithm for best arm identification in stochastic combinatorial semi-bandits under the fixed confidence setting, is very specific and a bit narrow in scope. The proposed algorithm is very tailored to the current setting, and the results are asymptotic. The motivation of this manuscript seems to be mainly technical curiosity, where the proposed algorithm is too complicated and very unlikely to be used by practitioners.

2. The nonexistence of statistical-computational gap is not very surprising, which slightly lowers the significance of the result. Also, the attained sample complexity is asymptotically optimal only in the high confidence regime and also involves a parameter polynomial in 1/Delta_min.

3. The two-player zero-sum game approach for an oracle-efficient algorithm is nowadays pretty standard, especially when one player is playing FTPL and the other is using best response. So the significance of the first component is medium.

4. The Frank-Wolfe sampling approach, albeit different from [Wang et al. 2021], is also pretty natural. The same also applies to the smoothing technique. The complicated notations and many seemingly arbitrary rules in the algorithm (more on the writing below) make it considerably harder to appreciate the main idea, and also degrade the significance.

In addition, I would like to point out that the writing quality of this manuscript is pretty poor. Just list several examples:

1. The LM oracle is not even formally defined in the main text.

2. Algorithm 2 is poorly explained. Why forced exploration when \sqrt(t/|X_0|) \in \bN? Why does the best response to the estimated gradient correspond to a good decision? The definition of \nabla F_{\mu, \eta, n} is deferred to a very later place. What is the intuition behind the stopping rule? What is the beta function in the stopping rule? Many components require explanations.

3. Small places: "would" in Line 220, "in" in Line 257.

**Questions:**

The remainder term in Theorem 2 depends on Delta_min, which is pretty unfortunate, for this quantity could be small in combinatorial bandits. I am wondering if one can argue that this term is always negligible compared with the main term? More specifically, when Delta_min is small, then the optimal sample complexity T^* would also be large; I'm wondering if T^* could always dominate the poly(1/Delta_min) term in the remainder.

---

> ### Author Rebuttal · Authors · 2023-08-09
>
> Many thanks for your careful review and feedback.
>
> About the paper scope. We believe that combinatorial bandits with semi-bandit feedback constitute one of the classical problems in the bandit literature. They find numerous applications, see [11, 12, 14, 16, 23, 36, 43, 50] (references are those from the supplementary material). Refer to the answer to reviewer Vkf2 for details. We will cite a few of these applications in the final version of the paper.
>
> About:
> > the results are asymptotic.
>
> Our primary objective was to derive an algorithm that is statistically optimal in the high confidence regime (when $\delta$ goes to 0 – so indeed with asymptotic guarantees) and that runs in polynomial time. We achieved this objective, but our algorithm also exhibits non-asymptotic guarantees. It has a sample complexity polynomial in $K$ in the moderate confidence regime, see Theorem 4 (when $\delta$ does not necessarily tend to 0). Note that no algorithm with both (i) minimal sample complexity in the high confidence regime and (ii) polynomial sample complexity in the moderate confidence regime was known even in the vanilla MAB problem (in the recent paper [6], the authors did not manage to get these guarantees). P-FWS enjoys these guarantees in a setting that is far more challenging than what [6] (or [40]) considers.
>
> About:
> > The nonexistence of statistical-computational gap is not very surprising.
>
> We would like to emphasize that our result is new even for the vanilla MAB problem: as mentioned above, there is no existing algorithm whose sample complexity is both asymptotically optimal and is polynomial in $K$ in moderate regime. In addition, for combinatorial semi-bandits, we are the first to show the optimization problem leading to the problem-specific sample complexity lower bound can be solved in polynomial time. The level of complexity of this problem was so far unknown. Based on this result, we can devise an algorithm, P-FWS, that tracks the lower bound in a computationally efficient way.
>
> About:
> > The proposed algorithm is too complicated.
>
> Based on existing Julia implementations [21, 57], we implemented the sampling rule of P-FWS within 50 lines of code (this amount of code is comparable to what other sampling rules take), and used 100 lines of code to implement the MCP algorithm. We did not present this implementation in the paper because we found it difficult to compare the performance of our algorithm to that of other algorithms. The latter algorithms are computationally too challenging (it takes too much time to run). Here is an illustrative example. We compare P-FWS to a simplified version of CombGame [36]. The experiments are performed on a Macbook (Apple M1) with 8 cores and 16GB memory. We made $100$ runs to compute the expected sample complexity of the algorithms
>  * On a graph with $K=20$ edges and a decision set consisting of $|X|=21025$ spanning trees and confidence $\delta=0.1$, P-FWS has an expected sample complexity $\tau$ of 1139 samples while CombGame with OFW has an expected sample complexity of 1325 samples but already takes 21 minutes for each run.
>  * Now on a graph of $K=25$ edges and $|X|=0.3$ million spanning trees and confidence $\delta=0.1$, P-FWS expected sample complexity is 2000 samples (and in average, each run takes 17 hours) while any run for CombGame fails to finish within 2 days.
>
> Note that the original CombGame algorithm had to be simplified. More precisely, we use our MCP subroutine to perform two components of CombGame, namely the best-response player and the stopping rule. Even with this simplification, CombGame rapidly becomes computationally intractable as the size of the decision set increases.
>
> About:
> > the attained sample also involves a parameter polynomial in $1/\Delta_{\min}$
>
> The dependence of sample complexity in $1/\Delta_{\min}$ is unavoidable: it is easy to derive from Proposition 1 (c) that $T^*(\mu)$ is at least $1/\Delta_{\min}^2$.
>
> About:
> > The two-player zero-sum game approach for an oracle-efficient algorithm is nowadays pretty standard.
>
> Our contributions consist in (i) establishing the property that allows one to relate Eq (3) to a two-player zero-sum game and (ii) designing an algorithm that not only converges to the equilibrium but also returns an equilibrium action, even in a setting where one of the domains (the action set of one player) is not convex.
>
> About (i), prior works [12, 57] apply Lagrangian multiplier method to derive the closed-form of $f_x(\omega,\mu)$, but as far as we know, no one noticed the linear-concave property of the Lagrangian dual function as we showed in Proposition 1. This property opens the opportunity to relate Eq (3) to a two-player zero-sum game.
>
> Now achieving (ii) is non-trivial. In the standard setting [2, 19, 52], the domains for both players are convex, and the objective is just to converge to the equilibrium with low cumulative regret. In our case, for MCP, one of the domains (that of the $x$-player) is combinatorial. Moreover, we wish to return the equilibrium action (which is more difficult that just ensuring low cumulative regret) – this type of convergence is referred to as last-iterate convergence in the literature, see e.g. [20] and other references cited in Section 3.2.
>
> About:
> > The Frank-Wolfe sampling approach is also pretty natural.
>
> The Frank-Wolfe approach of [57] cannot be applied here because it would lead to a computationally infeasible algorithm as we explain in the paper. Hence, we had to use and analyze a different smoothing technique.
>
> About:
> > What is the intuition behind the stopping rule? What is the beta function in the stopping rule?
>
> The stopping rule basically compares $F_{\hat{\mu}(t-1)}(\omega(t))$ (which represents the distance of the current estimation to the closest bandit model with a different the best action) to a threshold function $\beta$. Please refer to [29, 42] for details. See line 291-292 for possible choices of threshold function $\beta$.

---

> > ### Comment · Reviewer_d388 · 2023-08-16
> >
> > Thank for for your detailed comment. Some follow-up questions:
> >
> > 1. $1/\Delta_{\min}$ dependence: I know that $T^\star(\mu)$ is at least $1/\Delta_{\min}^2$; what I was asking in my question is that can you show the $\text{poly}(1/\Delta_{\min})$ you derived is always no larger than $T^\star(\mu)$. I mean, if $T^\star(\mu)$ is only larger than $1/\Delta_{\min}^2$ but your $\text{poly}(1/\Delta_{\min})$ term is $1/\Delta_{\min}^{10}$, that's still very unfortunate.
> >
> > 2. Regarding your point (ii) in the two-player zero-sum approach: please specify if your proof of why your $x_e$ works only requires a low cumulative regret (and possibly thanks to the special problem structure in your Lagrangian), or you need to prove the last-iterate convergence. Right now only the cumulative regret is shown in Lemma 3, and I don't know if your proved a hard last-iterate convergence in Theorem 3, or you essentially applied a smart one-line trick to bypass it.

---

> > > ### Author Response · Authors · 2023-08-17
> > >
> > > Thanks for your questions!
> > >
> > > **About $1/\Delta_{\min}$ dependence.** Looking at (32) in the appendix, the dependence of the term $\Psi(\epsilon,\tilde{\epsilon})$ involved in the moderate confidence regime scales well with $\Delta_{\min}$ (i.e., $\Delta_{\min}^{-2.5}$). But in (32), the trade-off between the term for the high confidence regime (roughly $T^\star(\mu)^{-1}\log(1/\delta)$ and the term $\Psi(\epsilon,\tilde{\epsilon})$ involves selecting $\epsilon$ w.r.t. $\Delta_{\min}$. Hence, at the end, the dependence of $\Psi(\epsilon,\tilde{\epsilon})$ in $1/\Delta_{\min}$ could be a polynomial with relatively high order.
> > >
> > > We can probably achieve a better dependence in $\Delta_{\min}$ in the moderate confidence regime. We haven’t tried and we let this question for future work. Indeed, our main goal in the paper was to devise an algorithm that is optimal in the high confidence regime and that runs in polynomial time. We achieved this goal and our guarantees in the moderate confidence regime came as a “bonus”. For the moderate confidence regime, we wanted to focus on the dependence in $K$ (the number of basic arms), and we managed to achieve a sample complexity in this regime only polynomial in $K$ (as explained in Appendix B, line 571-572, naïve methods, e.g. applying the algorithm of [57], would lead to a sample complexity exponentially growing with $K$).
> > >
> > > We would like to finally note that the expected sample complexity in moderate confidence regime and its dependence in $\Delta_{\min}$ is not known, even actually in the vanilla MAB problem. In our setting, an additional difficulty stems from the fact that we need a computationally efficient algorithm; refer to Appendix B for a detailed discussion.
> > >
> > >
> > > **About “two-player zero-sum approach”**. Our approach only requires a low cumulative regret, as explained in line 208-212. Our approximate, $\hat{F}$, converges to the desired minimax value $F_{\mu}(\omega)$ thanks to the following three inequalities:
> > >  1.  (Lemma 3): $\frac{1}{N}\sum_{n=1}^N g_{\omega,\mu}(x^{(n)},\alpha^{(n)})-\frac{1}{N}\min_{x\neq i^\star}\sum_{n=1}^Ng_{\omega,\mu}(x,\alpha^{(n)}) \le \frac{c_\theta}{\sqrt{N}}$ with probability at least $1-\theta$.
> > >  2. (line 208): $\frac{1}{N}\sum_{n=1}^N g_{\omega,\mu}(x^{(n)},\alpha^{(n)})\ge \hat{F}$.
> > >  3. (line 209): $\frac{1}{N}\min_{x\neq i^\star}\sum_{n=1}^Ng_{\omega,\mu}(x,\alpha^{(n)}) \le F_{\mu}(\omega)$.
> > >
> > > As a result, $\hat{F}- F_{\mu}(\omega)\le \frac{c_\theta}{\sqrt{N}}$ with probability at least $1-\theta$. We would like to emphasize that the above approach from low cumulative regret to the returned iterate is specific to the requirement of our problem setting, where the approximate of equilibrium point, $\hat{x}$, has to be an *action* rather than an *average of actions*. This also makes our two-player zero-sum approach not as standard as in other works.

---

> > > > ### Comment · Reviewer_d388 · 2023-08-17
> > > >
> > > > Thank you for the detailed response - I really appreciate it. I'll happily increase my rating from 5 to 6, and I hope the authors could add the above discussions (the precise polynomial dependence on $\Delta_\min$, and the special structure to bypass the last-iterate convergence) to the final version.

---

### Official Review · Reviewer_26At · 2023-07-07

**Soundness:** 3 good
**Presentation:** 3 good
**Contribution:** 3 good
**Rating:** 6
**Confidence:** 4

**Summary:**

The paper is about best arm identification with fixed confidence, in combinatorial semi-bandits. The state of the art in that setting is that we have algorithms that have optimal asymptotic sample complexity, but their computational complexity is very large. The authors provide a new algorithm which remediates this problem. The algorithm is asymptotically optimal and has polynomial computational complexity (in the number of arms and parameters of the problem like the minimal gap).
The main contribution is a method to compute efficiently the closest alternative to an instance of the combinatorial semi-bandit problem, which is a sub-routine is several BAI algorithms. That method is incorporated into a BAI algorithm based on Frank-Wolfe.

**Strengths:**

The algorithm presented is the first to have polynomial computational complexity in combinatorial BAI. This is a significant achievement towards having practical algorithms for that important problem. The sample complexity is also asymptotically optimal.

The new method MCP is the result of an interesting reformulation of the problem and a thorough analysis of its properties. Its integration in a Frank-Wolfe algorithm also uses innovative methods compared to the existing FW algorithm for BAI.

The algorithms and the results are clearly explained.

**Weaknesses:**

The dependence in K, D and other constants is indeed polynomial, but with very large exponents. Some bounds depend on K^15. Since this work presents the first algorithm with polynomial complexity, this is only a mild weakness.

The main weakness is that there is no experimental evaluation.
- Is the method still very inefficient in practice? Can it actually run on a computer?
- Some components of the algorithm seem to be included to make the analysis work, like the comparison of \sqrt{t} with the norm of the empirical mean vector. A natural question is whether they are necessary in practice, or if they are artifacts of the analysis. This could have been studied empirically.
- The paper does not provide only a computationally efficient subroutine for a known algorithm, but also proposes a new BAI algorithm based on Frank-Wolfe (another FW based method already exists, but it is not exactly the same). The sample complexity of the new method should then also be evaluated. Perhaps an innocuous looking modification made it very bad in practice, although the asymptotic guarantees are good?

**Questions:**

Can you provide an empirical analysis of the proposed method and compare it to other algorithms? Both for sample and computational complexity.


**Limitations:**

The limitations of the paper are adequately discussed. No concern about societal impact.

---

> ### Author Rebuttal · Authors · 2023-08-09
>
> Many thanks for your careful review and positive feedback.
>
> About the comment:
> > Some components of the algorithm seem to be included to make the analysis work, like the comparison of \sqrt{t} with the norm of the empirical mean vector.
>
> We agree that naturally, some of the components of our algorithm, like the comparison to $\sqrt{t}$, have been precisely designed so that we obtain the desired performance guarantees. Now it is indeed interesting to see whether in practice, the algorithm would work tuning down these components. We leave this sensitivity analysis for future work.
>
> About the absence of experiments. In this paper, we wanted to highlight its strong theoretical contributions: for combinatorial bandits, our algorithm P-FWS is the first polynomial-time algorithm that is statistically optimal in the high confidence regime. In addition, it has a sample complexity polynomial in $K$ in the moderate confidence regime.
> We implemented P-FWS. But we found it difficult to compare its performance to that of other algorithms, for these algorithms are computationally too challenging (it takes too much time to run). Here is an illustrative example. We compare P-FWS to a simplified version of CombGame [36]. The experiments are performed on a Macbook (Apple M1) with 8 cores and 16GB memory. We made 100 runs to compute the expected sample complexity of the algorithms
>  * On a graph with $K=20$ edges and a decision set consisting of $|X|=21025$ spanning trees and confidence $\delta=0.1$, P-FWS has an expected sample complexity $\tau$ of $1139$ samples while CombGame with OFW has an expected sample complexity of 1325 samples but already takes 21 minutes for each run.
>  * Now on a graph of $K=25$ edges and $|X|=0.3$ million spanning trees and confidence $\delta=0.1$, P-FWS expected sample complexity is 2000 samples (and in average, each run takes 17 hours) while any run for CombGame fails to finish within 2 days.
>
> Note that the original CombGame algorithm could not be implemented – it takes way too much time to finish. We had to simplify it. More precisely, we use our MCP subroutine to perform two components of CombGame, namely the best-response player and the stopping rule. Even with this simplification, CombGame rapidly becomes computationally intractable as the size of the decision set increases.

---

### Official Review · Reviewer_Vkf2 · 2023-07-07

**Soundness:** 4 excellent
**Presentation:** 2 fair
**Contribution:** 4 excellent
**Rating:** 6
**Confidence:** 2

**Summary:**

The paper introduces a computationally efficient algorithm, P-FWS (Perturbed Frank-Wolfe Sampling), designed for best arm identification in Combinatorial semi-bandits. This algorithm operates in polynomial time and offers minimal sample complexity guarantees or polynomial sample complexity, depending on the problem regime. P-FWS is an online estimation algorithm that aims to achieve the information-theoretic lower bound at each stage. It leverages the Frank-Wolfe optimization routine as its core component, relying on linear maximization at each step. Overall, the paper presents an innovative and efficient approach to address the best arm identification problem in Combinatorial semi-bandits.

**Strengths:**

One strength of the paper lies in its exploration of a novel and important problem within the bandit literature. The utilization of Perturbed Frank-Wolfe as the central algorithm for estimating the lower bound is a unique and valuable contribution that has the potential to complement and enhance existing works in the field. The paper showcases strong theoretical rigor by providing detailed proofs for all claims, ensuring the reliability and robustness of the presented findings. Overall, the paper's innovative problem formulation and theoretical foundation make it a valuable addition to the broader bandit literature.

**Weaknesses:**

One weakness of the paper is its challenging readability and navigation. The excessive use of definitions, constants, and interruptions in the flow of the paper makes it difficult to understand and follow. This overload of information can overwhelm the reader and hinder comprehension.

Furthermore, the practical motivation and real-world use cases of the proposed algorithms are discussed in a limited scope. It would be beneficial to expand on the potential applications and provide more examples of how such algorithms can be applied in real-world scenarios.

Additionally, the absence of simulations in the main paper is notable. Although the paper highlights the computational efficiency of the proposed algorithms, there is a lack of performance comparison with other non-efficient methods. Including such comparisons would provide valuable insights into the comparative advantages and disadvantages of the proposed approach.

Overall, addressing these weaknesses would enhance the clarity, applicability, and overall impact of the paper.

**Questions:**

Would this paper have any significance or relevance to the part of the bandit literature that focuses on tracking the lower bound and sampling based on such lower bound estimations, which are often computationally infeasible?

**Limitations:**

No potential negative societal impact discussed.

---

> ### Author Rebuttal · Authors · 2023-08-09
>
> Many thanks for your careful review and feedback.
>
> About the presentation and readability, thanks for your feedback! We had chosen to write a long introduction to summarize all the ideas and contributions of the paper, and to help the reader understand the paper structure. This “unconventional” presentation may have been confusing. We will revert to a more classical and simplified structure.
>
> Regarding the practical motivation, we did not mention many of them because combinatorial bandits have been extensively studied and applied in the past. In particular, applications of combinatorial semi-bandits [11, 12, 14, 16, 23, 36, 43, 50] (references are those from the supplementary material) include:
>  * The online ranking problem [23] of identifying the $m$ most relevant (out of a total of $K$) items can be modeled in our setting with $m$-sets or $m$-permutations as the decision set.
>  * The routing problem [14, 43] of finding the routing tree with the lowest expected latency corresponds to our case with the set of all possible spanning trees in an ISP (Internet Service Provider) network as the decision set.
>  * The loan-assignment problem [43], which aims to identify the matching between the lending institute and the lenders such that the expected paid rate is the highest, can be formulated by setting the decision set as the set of all possible perfect matchings in a bipartite graph.
>  * The path planning problem [36] of finding the path from an origin $s$ to a destination $t$ with minimal expected traveling time can be modeled by using the set of all possible $s$-$t$ paths in the given directed acyclic graph as the decision set.
>
> We will cite a few of these applications in the final version of the paper.
>
> About your question:
> > Would this paper have any significance or relevance to the part of the bandit literature that focuses on tracking the lower bound and sampling based on such lower bound estimations, which are often computationally infeasible?
>
> Indeed, this is exactly what we are addressing in this paper. For the case of combinatorial bandits, the problem specific lower bound solves an intricate optimization problem whose level of complexity was so far unknown. We prove that we can solve this problem in polynomial-time (this is the first result of this kind). As a consequence, we are able to devise an algorithm that tracks the lower bound in a computationally efficient way. Our algorithm is statistically optimal and runs in polynomial time, and this is the first algorithm to combine these two properties!
> We would like to add that our algorithm also exhibits a sample complexity polynomial in $K$ in the moderate confidence regime (when $\delta$ does not necessarily tend to 0). Note that no algorithm with both (i) minimal sample complexity in the high confidence regime and (ii) polynomial sample complexity in the moderate confidence regime was known even in the vanilla MAB problem (in the recent paper [6], the authors did not manage to get these guarantees). P-FWS enjoys these guarantees in a setting that is far more challenging than what [6] (or [40]) considers.
>
> About the absence of experiments. In this paper, we wanted to highlight its strong theoretical contributions – as listed in the previous paragraph. We implemented our algorithm P-FWS. But we found it difficult to compare its performance to that of other algorithms, for these algorithms are computationally too challenging (it takes too much time to run). Here is an illustrative example. We compare P-FWS to a simplified version of CombGame [36]. The experiments are performed on a Macbook (Apple M1) with 8 cores and 16GB memory. We made 100 runs to compute the expected sample complexity of the algorithms
>  * On a graph with $K=20$ edges and a decision set consisting of $|X|=21025$ spanning trees and confidence $\delta=0.1$, P-FWS has an expected sample complexity $\tau$ of 1139 samples while CombGame with OFW has an expected sample complexity of 1325 samples but already takes 21 minutes for each run.
>  * Now on a graph of $K=25$ edges and $|X|=0.3$ million spanning trees and confidence $\delta=0.1$, P-FWS expected sample complexity is 2000 samples (and in average, each run takes 17 hours) while any run for CombGame fails to finish within 2 days.
>
> Note that the original CombGame algorithm could not be implemented – it takes way too much time to finish. We had to simplify it. More precisely, we use our MCP subroutine to perform two components of CombGame, namely the best-response player and the stopping rule. Even with this simplification, CombGame rapidly becomes computationally intractable as the size of the decision set increases.

---

> > ### Author Response · Authors · 2023-08-17
> >
> > Dear reviewer, we were wondering whether you read our rebuttal and whether you find that it clarified our contributions. Please let us know! We would be happy to answer further questions, if any (the discussion phase ends August 21). Thank you. Best wishes.

---

> > > ### Comment · Reviewer_Vkf2 · 2023-08-21
> > >
> > > While I would recommend trying to make the theorems more interpretable and readable, I think it's an individual choice and shouldn't be a major factor in reviewing process.
> > >
> > > After a couple of re-reads and looking at rebuttal responses to other reviewers, I am agreeing with a lot which is already been discussed. I will be changing my score to 6 (taking into account the novelty of the approach and the theoretical backings of the same).

---

### Decision · Program_Chairs · 2023-09-21

**Decision:**

Accept (poster)

**Comment:**

This paper makes a substantial advance on a problem of broad interest to the ML community: combinatorial semi-bandits. Prior to this work, we didn't have approaches that were both computationally and statistically efficient. Their Frank-Wolfe based solution might also inspire similar work on related problems.